# Laboratory and numerical experiments on stem waves due to monochromatic waves along a vertical wall

Sung Bum Yoon[1], Jong-In Lee[2], Young-Take Kim[3] and Choong Hun Shin[1]

[1]Department of Civil and Environmental Engineering, Hanyang University, EIRCA Campus, Ansan, Gyeonggi, 15588, South Korea
[2]Department of Marine and Civil Engineering, Chonnam National University, Yeosu Campus, Yeosu, Jeonnam, 59626, South Korea
[3]River and Coastal Research Division, Korea Institute of Civil Engineering & Building Technology, Goyang, Gyeonggi, 10223, South Korea

*Correspondence to*: Choong Hun Shin (lavici@hanyang.ac.kr)

**Abstract.** In this study, both laboratory and numerical experiments are conducted to investigate stem waves propagating along a vertical wall developed by the incidence of monochromatic waves. The results show the following features: For small amplitude waves, the wave heights along the wall show a slowly varying undulation. Normalized wave heights perpendicular to the wall show a standing wave pattern. The overall wave pattern in the case of small amplitude waves shows a typical diffraction pattern around a semi-infinite thin breakwater. As the amplitude of incident waves increases both the undulation intensity and the asymptotic normalized wave height decrease along the wall. For larger amplitude waves with smaller angle of incidence, the measured data show clearly stem waves. Numerical simulation results are in good agreement with the results of laboratory experiments. The results of present experiments support favorably the existence and the properties of stem waves found by other researchers using numerical simulations. The characteristics of the stem waves generated by the incidence of monochromatic Stokes waves are compared with those of the Mach stem of solitary waves.

## 1 Introduction

Coastal structures have been increasingly constructed in deep water regions as the size of ships becomes larger. In such deep water regions, a vertical-type structure is preferred to save construction costs. In the case of a vertical structure, stem waves occur when waves propagate obliquely against the structure. Thus, there is a need for careful consideration to secure appropriate free board and stability of caisson blocks.

Based on laboratory experiments of the reflection of a solitary wave propagating obliquely against a vertical wall, Perroud (1957) reported the existence of three types of waves when the angle between incident wave ray and a vertical wall is below 45°: incident, reflected, and stem waves. On the one hand, Berger and Kohlhase (1976) conducted laboratory experiments and found that stem waves appeared also in the case of sinusoidal waves, and that the properties of stem waves developed by sinusoidal waves showed similarities to those of solitary waves. On the other hand, according to laboratory experiments by Melville (1980) with solitary waves, the width and height of stem waves were found to be wider and larger, respectively, as

waves propagated along the wall. However, the wave height did not exceed double the height of incident waves. Yue and Mei (1980) analysed stem waves at a constant water depth using parabolic approximation equations for second-order Stokes waves. They found that the influence of reflected waves was removed when the incident angle between the structure and the waves was below 20° and that only incident waves and stem waves appeared. Liu and Yoon (1986) showed that stem waves

occurred also in an area along the line of a depth discontinuity, as in the case of a vertical wall. In addition, Yoon and Liu (1989) introduced a parabolic approximation equation based on the Boussinesq equation and analysed stem waves for the case of cnoidal incident waves. Yoon and Liu (1989) showed the importance of the incident wave nonlinearity. Most previous studies on stem waves focused on the properties of stem waves depending on the incident angle and wave nonlinearity of monochromatic waves.

While the stem waves generated by the sinusoidal waves have drawn less attention in recent years, the Mach stem induced by the interaction between the line solitons in the shallow-waters has continuously attracted the attention of the researchers. Since the pioneering work of Miles (1977a, b) on the obliquely interacting solitary waves, the soliton interactions have been extensively studied. Miles (1977b) developed an analytical solution to predict the amplification of the stem wave along the wall as a function of the interaction parameter, $k_* = \theta_0 / \sqrt{3H_0/h}$, where $H_0$, $h$ and $\theta_0$ are the wave height, the water depth

and the incident angle of solitary wave, respectively. When $k_* = 1$, the amplification of a solitary wave can reach four times of the incident wave. Soomere (2004) investigated the soliton interactions based on the KP equation (Kadomtsev and Petviashvili, 1970). Kodama et al. (2009) and Kodama (2010) proposed the modified interaction parameter, $\kappa_* = \tan \theta_0 / (\sqrt{3H_0/h} \cos \theta_0)$, and developed an exact solution for the KP equation. Li et al. (2011) conducted a precision laboratory experiment to capture the detailed features of Mach reflection using the LIF (laser-induced fluorescent) technique.

The laboratory data of Li et al. (2011) support strongly the theory of Miles (1977b) except the cases where $\kappa_*$ value lies in the neighbourhood of the fourfold amplification. Funakoshi (1980), Tanaka (1993), Li et al. (2011), and Gidel et al. (2017) performed numerical experiments to verify the Miles' fourfold amplification. As summarized by Li et al. (2011) and Gidel et al. (2017) most of the models underestimated the fourfold amplification due to the limitations of the computational resources. The amplification ratio of 3.6 obtained by Gidel et al. (2017) is so far the maximum among the numerical results showing

the full development stage of stem wave.

Even though the existence and the properties of stem waves for sinusoidal waves are well known theoretically via numerical simulations (e.g., Yue and Mei, 1980; Yoon and Liu, 1989), they are not yet fully supported by physical experiments. Berger and Kohlhase (1976) conducted hydraulic experiments to show the existence of stem waves for the cases of sinusoidal waves. Their experimental data, however, failed to produce clear stem waves, possibly due to partial reflection from the beach,

diffraction from the ends of vertical wall, or insufficient space in the wave basin. Lee et al. (2003), Lee and Yoon (2006) and Lee and Kim (2007) performed laboratory experiments to investigate stem waves for sinusoidal waves, and compared the measured waves with the numerical results obtained using a nonlinear parabolic approximation equation model. Their hydraulic experiments demonstrated stem waves for some cases with a relatively large incident wave. However, the stem

waves were not clearly developed because of both the narrowness of the wave basin and the reflected waves from the beach. Only four cases of incident wave conditions were tested in their experiment. Thus, the experimental data were not sufficient to investigate the properties of stem waves. Moreover, the numerical results for the cases of large angle of incidence were not highly accurate because of the small-angle parabolic model employed for their numerical simulations. Thus, there is still need to perform a precisely controlled experiment to investigate the existence and the properties of stem waves.

In this study, precisely-controlled laboratory experiments are conducted to investigate the characteristics of stem waves developed by the incidence of monochromatic waves. The measured data are compared with numerical simulations and analytical solutions. In the following section, the numerical simulation method and the analytical solution employed in this study are summarized. In section 3, the experimental setup and procedure are briefly presented. In section 4, the measured wave heights are compared with numerically simulated results and analytical solutions. In section 5, the characteristics of the stem waves generated by the incidence of monochromatic Stokes waves are compared with those of the Mach stem of solitary waves. In the final section, the major findings from this study are summarized.

## 2 Numerical simulation and analytical solution

In this study, the stem waves that have developed along a vertical wall over a constant water depth are investigated for the cases of monochromatic waves. Fig. 1 shows the definition sketch of the wave field around a vertical wedge. The monochromatic waves are symmetrically incident towards the tip of the wedge. The $x$-axis of the coordinate system is aligned with a side wall of the wedge. The angle of incidence $\theta_0$ is defined as the angle between the $x$-axis and the incident wave ray. The computational domain lies in the region of $0 \leq x$ and $y \leq 0$.

### 2.1 Numerical simulation method

To compare with our experiments, the latest version of REF/DIF, a wide-angle nonlinear parabolic approximation equation model developed by Kirby et al (2002), is employed to simulate stem waves. The REF/DIF model can deal with the refraction-diffraction of Stokes waves of third order nonlinearity over a slowly varying depth and current. Due to the use of parabolic formulation the reflection in the main direction of propagation is forbidden, but not in the transverse direction. In this study, the water depth is uniform, and no ambient current is present. With no current and energy dissipation on a constant water depth and by selecting a (1, 1) Padé approximant in the model, the governing equation of the REF/DIF model is simplified as

$$2ik \frac{\partial A}{\partial x} + \frac{\partial^2 A}{\partial y^2} + \frac{i}{2k} \frac{\partial^3 A}{\partial x \partial y^2} - \frac{\omega k^3}{C_g} D |A|^2 A = 0, \tag{1}$$

where $h$ is the water depth, $i = \sqrt{-1}$, $C_g$ is the wave group velocity, $A$ is the complex wave amplitude, $k$ and $\omega$ are the wave number and the angular frequency, respectively, and satisfy the following linear dispersion relationship:

$$\omega^2 = gk \tanh kh, \tag{2}$$

where $g$ is the gravitational acceleration, and $D$ is given as

$$D = \frac{\cosh 4kh + 8 - 2 \tanh^2 kh}{8 \sinh^4 kh}. \tag{3}$$

The third term of Eq. (1) is the correction term obtained by selecting (1, 1) Padé approximant for the wide angle parabolic approximation. According to Fig. 2 of Kirby (1986) the accuracy of the waves propagating obliquely to the main direction of propagation, i.e., $x$-direction, can be maintained up to $\pm 45°$. In this study the range of the incidence angles of both incident and reflected waves lies from $\pm 10°$ to $\pm 40°$. Thus, the considerable accuracy of the numerical solution is expected.

The conventional parabolic approximation equation, i.e., the nonlinear Schrödinger equation of Yue and Mei (1980) is obtained if this term is neglected. The last term in Eq. (1) represents the nonlinear effect of waves. Fig. 2 shows the coordinate system for the present numerical simulation in comparison with that of Yue and Mei (1980). In the present simulation the incident waves are prescribed obliquely along the $y$-axis as

$$A = a_0 e^{ik_n y \sin \theta_0}, \tag{4}$$

where $a_0$ is the amplitude of the incident wave, and $k_n$ is the nonlinear wave number given as

$$k_n = k \left( 1 - \frac{C}{2C_g} D(k|A|)^2 \right), \tag{5}$$

where $C (= \omega/k)$ is the phase speed of wave. No-flux boundary condition is prescribed along the vertical wall ($y = 0$) given as

$$\frac{\partial A}{\partial y} = 0. \tag{6}$$

If the side boundary opposite to the vertical wall is located far from the wall, no flux boundary condition, Eq. (6), can also be used. However, to save the computational resources the obliquely-incident plane wave condition is prescribed along the side boundary at $y = -y_{\max}$ as

$$A = a_0 e^{i(k_n x \cos \theta_0 - k_n y_{\max} \sin \theta_0)}. \tag{7}$$

Along the down-wave side no boundary condition is necessary, because Eq. (1) is a parabolic type differential equation. The grid size, $\Delta x$ and $\Delta y$, is $L/80$ where $L$ is the wave length of an incident wave. The size of computational domain is $50L$ in the $x$-direction, and $400L$ in the $y$-direction.

For the later use the nonlinear parameter, $K$, proposed by Yue and Mei (1980) is given as:

$$K = \left(\frac{ka_0}{\tan \theta_0}\right)^2 \frac{CD}{C_g}. \tag{8}$$

$K$ is the single parameter representing both the nonlinearity of incident wave and the angle of incidence on the formation of stem waves along the vertical wall. This nonlinear parameter was obtained by Yue and Mei (1980) from the dimensionless form of the small angle version of Eq. (1). The details of the derivation of $K$ can be found in Yue and Mei (1980).

## 2.2 Analytical solution

Chen (1987) developed an analytical solution for the Helmholtz equation in polar coordinates to solve the combined reflection and diffraction of monochromatic waves due to a vertical wedge. The analytical solution is given in a polar coordinate as shown in Fig. 1 as

$$\Phi(r, \theta^*, z, t) = -\frac{iga_0}{\omega} \frac{\cosh\{k(z+h)\}}{\cosh kh} F(r, \theta^*) e^{i\omega t}, \tag{9}$$

where $\Phi(r, \theta^*, z, t)$ is the velocity potential, and $F(r, \theta^*)$ is a diffraction factor (i.e., $A/a_0$) given as:

$$F(r, \theta^*) = \frac{2}{\nu}\left[J_0(kr) + 2\sum_{n=1}^{\infty} e^{in\pi/2\nu} J_{n/\nu}(kr) \cos\frac{n\alpha^*}{\nu} \cos\frac{n\theta^*}{\nu}\right], \tag{10}$$

where $\theta^* = \theta - 2\theta_0$, $\alpha^* = \pi - \theta_0$, $\nu = 2(\pi - \theta_0)/\pi$, and $\theta_0$ is the angle of incidence. $J_0(kr)$ is the Bessel function of the first kind of order 0. The absolute value of the diffraction factor $|F(r, \theta^*)|$ represents the normalized wave amplitude $|A|/a_0$, or the normalized wave height $H/H_0$ where $H_0 (= 2a_0)$ is the wave height of the incident wave. The analytical solution of

Chen (1987) is linear. Thus, this analytical solution does not allow the formation of stem waves. The details of the derivation of the analytical solution can be found in Chen (1987).

## 3 Hydraulic experiments

Hydraulic experiments are carried out in the multidirectional irregular wave generation basin of the Korea Institute of Construction Technology (see Photo 1). The basin used in the laboratory experiments is 42 m long, 36 m wide and 1.05 m high. A snake-type wave generator consisting of 60 wave boards, each with dimensions of 0.5 m in width and 1.1 m in height and driven by an electronic servo piston, is installed along the 36 m long bottom wall of the wave basin. Free surface displacements are measured using 0.6 m long capacitance-type wave gauges with a measuring range of ±0.3 m.

Fig. 3 shows the configuration of the experimental setup and model installation. A 30 m long vertical wall is installed along the left lateral side of the basin in four different orientations. A dissipating gravel beach with a 1/20 slope is arranged on the opposite side of the wave generator to reduce the reflection of waves inside the basin. Another dissipating beach and wave absorber are also set along the lateral sides and at the back of the wave generator. Along the lateral side opposite to the vertical wall a 10 m long wave guide is installed to avoid diffraction from the side wall. Note that $\theta_0$ is the angle between the vertical wall and the incident waves. The origin of the spatial coordinate system of the laboratory experiments (i.e., $x_0$, $y_0$) is set at the tip of the vertical wall which is located 3 m and 5 m away from the lateral side and the wave generator, respectively, as shown in Fig. 3. The width and height of the vertical wall were both 0.6 m. The experiments are carried out at a constant water depth of $h = 0.25$ m. The free board from a still water level to the top of the vertical wall is 0.35 m in order to prevent overtopping of waves.

The incident wave conditions are summarized in Table 1. The title of each test case is composed of three alphabet characters and a numeric digit. The first alphabet M stands for 'monochromatic' waves. The second alphabet S or L represents 'shorter' or 'longer' waves in terms of period, respectively. The third alphabet S, M or L represents 'small', 'medium', or 'large' waves in terms of wave height, respectively. Finally, the numeric digit represents the size of the angle of incidence.

The wave periods of $T = 0.7$ s and 1.1 s are tested. The wave heights are $H_0 = 0.009$ m, 0.027 m, and 0.036 m for 0.7 s waves, and $H_0 = 0.018$ m, 0.054 m, and 0.072 m for 1.1 s waves so that no wave breaking occurs during the experiments. The length of the vertical wall in the laboratory experiments is $40L$ for the case of $T = 0.7$ s and $20L$ for the case of $T = 1.1$ s, where $L$ represents the wavelength of monochromatic waves corresponding to the given period $T$. The incident angles of $\theta_0 = 10°$, $20°$, $30°$, and $40°$ are obtained by adjusting the orientation of the vertical wall. Thus, the incident waves propagate normal to the line of the wave generator. The nonlinearity of the incident waves are presented in two dimensionless parameters, wave steepness $kH_0$ and the nonlinear parameter $K$ given by Eq. (8).

In the real world, we can assume the situation where the swell is incident on a breakwater. Swell waves are the regular longer period waves created by storms far away from the beach. Swell waves tend to have longer periods than wind waves. The wave period of swell lies between 10 s to 15 s. Breakwaters are generally constructed at a depth of about 10 m to 20 m.

If the wave height is 1 m to 3m, the swell wave conditions can be within the range of Stokes wave as shown in Fig. 4. In the figure the empty blue circles represent the swell wave conditions and the red triangles represent the incident waves tested in this study. It can be seen that the incident waves tested in this study belong to the Stokes range. The dispersion effect of the Stokes waves is much stronger than that of the solitary waves. Thus, the characteristics of stem waves in this study should be

much different from those of the solitary waves. In Fig. 4, the $x$-axis represents the relative water depth (ratio of water depth to deep water wave length, i.e., the measure of wave dispersion). On the other hand, the $y$-axis represents the wave steepness (ratio of wave height to deep water wave length, i.e., the measure of wave nonlinearity).

In the experiments, wave heights are measured along both the vertical wall ($x$-direction) and normal to the vertical wall ($y$-direction). Note that wave heights in the $x$-direction are measured 0.05 m away from the front side of the wall, while wave

heights in the $y$-direction are measured along two lines of $x = 6L$ and $15L$. The intervals of the wave height measurement positions are $\Delta x = 0.2$ m and 0.4 m for $T = 0.7$ s and 1.1 s, respectively, along the wall, while $\Delta y = 0.1$ m and 0.2 m for $T = 0.7$ s and 1.1 s, respectively, normal to the wall. Table 2 gives a summary of the wave height measurement positions. The wave heights are extracted from the measured free surface displacements using the zero-upcrossing method. In this method a wave is defined when the surface elevation crosses the zero-line or the mean water level upward and continues

until the next crossing point. This method is a widely accepted method for extracting representative statistics from raw wave data. Photo 2 shows the hexagonal or beehive wave pattern captured during the experiment in front of a vertical wall for the case of $\theta_0 = 30°$. This is typical of the cross-sea generated by the oblique interaction of two or more traveling plane waves (see e.g., Le Mehauté, 1976; Mei, 1983; Nicholls, 2001). Postacchini et al. (2014) studied the dynamics of crossing wave trains on a plane slope in shallow waters. The stem waves can be developed at the intersection of two crest lines of the

crossing waves. When the crossing waves propagating towards a shore, they experience shoaling and break. Postacchini et al. (2014) proposed an analytical theory based on ray convergence to identify the position and the crest length of the breaker. The stem waves in the present study are developed by the oblique nonlinear interaction between the incident and the reflected waves. Thus, the generation mechanism is similar to Postacchini et al's.

Prior to the main experiments the performance of the wave generator is tested. For this test no vertical wall is placed in the

wave basin. After the initiation of wave generation the time histories of free surface displacement are recorded at three incident-wave-measuring points as shown in Fig. 3. The first part of data with a sufficiently long time is discarded in evaluating the wave height to avoid the start-up transients, and the wave height and period are obtained using the zero-upcrossing method. The tests show that the target waves are well generated, and also showed that the bottom friction is negligible within the test area of the wave basin. In particular, three wave gauges aligned in a wave propagation direction

with a specified distance are placed at the incident-wave-measuring point located near the gravel beach with a 1/20 slope to estimate the wave reflection from the beach. The incident and reflected waves are separated using the three-point higher order separation technique. This higher order technique is developed for finite amplitude waves by adding the second and third harmonics to the linear separation scheme proposed by Suh et al. (2001). The reflection coefficient due to the gravel beach is less than maintained 3% for all the waves considered in the experiments.

## 4 Results and discussions

In this study, experiments on the formation of stem waves around a vertical wall are conducted and the measured wave heights are compared with results calculated using both the wide-angle parabolic approximation equation numerical model, REF/DIF, and the analytical solution of Chen (1987). All the figures for the experimental and calculated data are presented in the Appendix to avoid a flourish of figures.

Prior to presenting the experimental and numerical results, the definitions of the stem angle and the stem width are discussed. The definition of stem width is rather controversial. Yue and Mei (1980) defined the stem width as the distance from the wall to the edge of the uniform wave amplitude region. However, it is not an easy task to locate the edge of the flat region. Berger and Kohlhase (1976) defined the stem width for the periodic waves as the distance along the stem crest lines from the wall to the first node line of the standing wave pattern which is easier to identify from the measured data. On the other hand, Soomere (2004) obtained the analytical stem length using the KP equation for the obliquely interacting two solitary waves. As pointed out by Li et al. (2011) the crest lines of the stem wave, the incident and the reflected solitons measured in their experiment are not straight, and they do not meet at a point. In reality, the analytical solutions of the KP equation deviate slightly from the pattern observed in the experiment. Thus, Li et al. (2011) proposed the edge of the Mach stem as the intersection of the linear extensions of the stem and the incident-wave crest lines.

For the periodic waves the wave pattern is more complicated because many wave components are superposed. Thus, the definitions of the stem boundary and the stem angle should be different from the case of solitary waves. As shown in Fig. 2(a) and Fig. 5, when the stem waves are fully developed, the stem boundary is nearly parallel to the first node line. Thus, as suggested by Berger and Kohlhase (1976), the experimental stem angle α is determined in this study as the angle of node line, $\alpha_n$. The node line is roughly determined using the node points from the wave height data measured along two lines of $x = 6L$ and $15L$. When the distances between the first node points and the wall are $\lambda_6$ and $\lambda_{15}$ for two sections of $x = 6L$ and $15L$, respectively (see Figs. A5 and A6), then the angle of the node line, $\alpha_n$, can be determined as

$$\alpha \approx \alpha_n = \tan^{-1}\left(\frac{\lambda_{15} - \lambda_6}{9L}\right). \tag{11}$$

This $\alpha_n$ decreases as the waves propagate along the wall. It reaches an asymptotic value after the waves propagate approximately 30 wave lengths. Thus, the experimental $\alpha_n$ determined by Eq. (11) is slightly overestimated for $x \leq 30L$. In this study the stem angle, α, is defined as the asymptotic angle of node line as shown in Fig. 5. To estimate the asymptotic $\alpha_n$ the numerical calculation is conducted using the domain extended up to $50L$ in the $x$-direction, and the instantaneous free surface displacements are calculated and plotted as shown in Fig. 5. Using two distances between the node points and the wall, $\lambda_{30}$ and $\lambda_{50}$ for two sections of $x = 30L$ and $50L$, respectively, the stem angle α is determined as

$$\alpha = \alpha_n = \tan^{-1}\left(\frac{\lambda_{50} - \lambda_{30}}{20L}\right). \qquad (12)$$

The stem width $\lambda_s$ can be determined using the stem angle $\alpha$ as

$$\lambda_s = x \tan \alpha. \qquad (13)$$

### 4.1 Shorter waves ($T = 0.7$ s)

Fig. A1 shows the comparisons between the measured, numerically simulated, and analytically calculated wave heights, $H/H_0$, along the vertical wall for the cases of $H_0 = 0.009$ m with $T = 0.7$ s (i.e., MSS-series). The amplitude of the incident waves is small as the title of the test cases indicates. The solid circles represent the results of the laboratory experiments. The solid and dashed lines represent the numerical (using REF/DIF) and analytical solution results, respectively. Various incident angles of $\theta_0 = 10°$, $20°$, $30°$, and $40°$ are presented. For the case of small angle of incidence (MSS1, $\theta_0 = 10°$) the measured

wave height along the vertical wall increases monotonically with the distance from the tip of the vertical wall. As the angle of incidence increases, the wave height shows a slowly varying undulation with the average value of $H/H_0 = 2.0$. The maximum value of undulation is approximately $H/H_0 \approx 2.3$, and the location of maximum wave height decreases with increasing angle of incidence. In particular, the overall pattern of wave height distribution does not support the generation of stem waves, which are characterized by uniform wave heights smaller than those obtained from linear diffraction theory

(Yue and Mei, 1980; Yoon and Liu, 1989). The wave heights calculated using the REF/DIF numerical model (Kirby and Dalrymple, 1994) and the analytical solution of Chen (1987) agree well with the measured wave heights. This supports the idea that the effects of nonlinearity of incident waves are too weak to develop stem waves. In the case of $\theta_0 = 10°$, the maximum normalized wave heights does not reach $H/H_0 \approx 2.3$ because the size of the experimental area is insufficient. If the vertical wall is sufficiently long, the same result could apparently be obtained for $\theta_0 = 10°$.

Figs. A2 and A3 show the comparisons of wave heights $H/H_0$ along a line ($x = 6L$, $15L$) perpendicular to the vertical wall. The distribution of wave height shows the typical pattern of standing waves formed by superposition of the reflected waves on the incident waves. Berger and Kohlhase (1976) called these standing waves stem waves as long as they propagated parallel to the wall. If stem waves, however, are defined as waves with a uniform wave height in the direction normal to the wall, then the wave height distributions for these small amplitude waves in MSS-series show no sign of stem waves. The

wave amplitude for this MSS-series is too small to generate stem waves along the wall.

Fig. A4 shows normalized wave heights along the vertical wall for the cases of MSM-series (i.e., $H_0 = 0.027$ m, $T = 0.7$ s) with various angles of incidence. The amplitude of the incident waves is three times larger than the MSS-series waves. Figs. A5 and A6 show normalized wave heights perpendicular to the vertical wall at positions $x = 6L$ and $15L$, respectively. The results shown in Fig. A4 indicate that, when the angle of incidence is small ($\theta_0 = 10°$), the normalized wave height

approaches to a uniform value of $H/H_0 \approx 1.75$ as waves propagated along the vertical wall. At larger incident angles, the maximum normalized wave heights reach up to $H/H_0 \approx 2.25$, and showed a slowly varying undulation.

In the results shown in Figs. A5 and A6 the stem waves of uniform wave height are found under the conditions of $\theta_0 = 10°$, $x = 6L$ and $15L$, albeit the stem widths are small. However, in the cases of other incident angles, stem waves do not appear.

The red lines shown in the figures represent the stem waves. For the stem width $\lambda_s$, the stem angle $\alpha$ is first determined by Eq. (12) using the numerical simulation result with the using extended domain. The stem width $\lambda_s$ is then calculated using Eq. (13) for given $x$.

The results from laboratory experiments are in good agreement with those of the results of REF/DIF model. However, the analytical solutions of Chen (1987) do not agree well with the measured data, probably because of nonlinear interactions

between incident and reflected waves. The discrepancy between the analytical solution of Chen (1987) and the measured data decreases as the angle of incidence increases. This can be attributed to the decrease in the intensity of nonlinear interactions between incident and reflected waves as the angle of incidence increases.

Figs. A7, A8, and A9 show the comparisons of the measured, numerically simulated, and analytically calculated results for the cases of MSL-series ($H_0 = 0.036$ m, $T = 0.7$ s). The amplitude of the incident waves is the largest among the shorter wave

test cases. For the case of small angle of incidence, $\theta_0 = 10°$, the normalized wave height increases monotonically to reach a constant value of $H/H_0 \approx 1.5$, with a strong indication of stem wave development. In the cases of larger angle of incidence the wave heights show a slowly varying undulation. As shown in Figs. A8 and A9, which represent normalized wave heights in the direction normal to the vertical wall, stem waves appear clearly for $\theta_0 = 10°$ along $x = 6L$ and $15L$. It can also be seen that the width of stem waves increases in proportion to the distance from the tip of vertical wall. In the cases of larger

incidence angles, the normalized wave heights tend to show a distribution pattern similar to that of standing waves normal to the wall.

## 4.2 Longer waves ($T = 1.1$ s)

Figs. A10, A11 and A12 show comparisons between the measured, numerically simulated, and analytically calculated wave heights $H/H_0$ along the vertical wall ($y=0$) and normal to the wall ($x = 6L$ and $15L$) for the cases of $H_0 = 0.018$ m with $T = 1.1$

25    s (MLS-series). The solid circles represent the results of laboratory experiments. The solid and dashed lines represent the numerical and analytical solutions, respectively. The results from laboratory experiments are in good agreement with those from the analytical solution and numerical model. The amplitude of the MLS incident waves is chosen to provide the same steepness, $kH_0 = 0.076$, as the MSS waves. Hence, the wave patterns observed in the MSS-series (Fig. A1) are similar to the results of the MLS-series.

Fig. A13 shows normalized wave heights along the vertical wall for the cases of MLM-series ($H_0 = 0.054$ m, $T = 1.1$ s). The incident wave amplitude is twice that of the cases of MSM-series, but the MLM-series have the same wave steepness $kH_0$ as MSM-series. For $\theta_0 = 10°$, the maximum value of the normalized wave height reached the uniform value of $H/H_0 \approx 1.65$, which shows an indication of the development of stem waves. Figs. A14 and A15 show normalized wave heights normal to

the vertical wall at positions along $x = 6L$ and $15L$ for various incident angles. As shown in Figs. A14 and A15, stem waves appear for the cases of $\theta_0 = 10°$. The stem widths increase proportionally with the distance from the tip of the vertical wall. The width of the stem waves is found to decrease as the incident angle increases. The linear analytical solutions for small incident angles show large deviations from the measured results, which is consistent with previous results for the cases of

MSM-series. On the other hand, the simulation results using the REF/DIF model are generally in good agreement with the results from laboratory experiments.

Figs. A16, A17, and A18 show comparisons of the measured, numerically simulated, and analytically calculated results of MLL-series ($H_0 = 0.072$ m, $T = 1.1$ s). In the results from the laboratory experiment, stem waves appear clearly at positions along $x = 6L$ and $15L$ for $\theta_0 = 10°$ and $20°$. Such clearly identifiable stem waves for periodic waves in the physical

experiments are observed for the first time in this study. Berger and Kohlhase (1976) also conducted laboratory experiments to produce stem waves with a vertical wall. The experiments of Berger and Kohlhase (1976) were conducted in a constant water depth of $h = 0.25$ m for the wave length of $L = 1.0$ m with various incoming wave heights of $H_0 = 0.023 \sim 0.053$ m, and incidence angles of $\theta_0 = 10°, 15°, 20°,$ and $25°$. The experimental wave conditions of Berger and Kohlhase (1976) are similar to those of this study. The length of vertical wall (less than $9.8L$) used in the experiments of Berger and Kohlhase

(1976), however, is much shorter than that of this study ($40L$ for the case of $T = 0.7$ s and $20L$ for the case of $T = 1.1$ s). Moreover, both ends of the vertical wall were open in the experiments of Berger and Kohlhase (1976), while a wave guide is installed from the wave generator to the tip of vertical wall in the present experiments, and the other end of the vertical wall is extended to the midst of 1/20 gravel beach. As a result, the wave heights along the wall measured by Berger and Kohlhase (1976) were contaminated by the parasitic waves diffracted by both ends of the wall. Thus, the stem waves developed along

the wall were not clear in the results of Berger and Kohlhase (1976), while the stem waves observed in the present experiments are clearly noticeable.

Figs. 6(a) and 6(b) show the comparison of the three-dimensional plots of normalized wave height for MLS1 and MLL1 cases, respectively, based on the numerical results of REF/DIF. For the nonlinear case, the overall amplitudes are much smaller and the stem waves are developed along the wall as shown in Fig. 6(b). The stem wave height is nearly constant and

the width of the stem waves tended to increase along the wall. Fig. 7(a) and Fig. 7(b) present the comparison of the three-dimensional plots of normalized free surface displacements, $\zeta/a_0 = \text{Re}((A/a_0)e^{ikx})$, for MLS1 and MLL1 cases, respectively. From Fig. 7(b) it can be seen that the stem waves propagate along the wall. Fig. 8 shows the contour plots of the instantaneous normalized free surface for MLS1 and MLL1 cases. The incident waves are reflected from the wall for the linear case. However, for the nonlinear cases, they seem to be both refracted and partially reflected at the edge of stem region

as depicted also in Fig. 2. The rigorous interpretation of these refraction and partial reflection is that the resonant interaction between the incident and reflected waves generates the stem waves propagating along the wall, and also shift the phase of the reflected waves outward from the stem region.

In conclusion, the results of the laboratory experiments are in good agreement with those of the numerical simulations. However, the analytical solution cannot reproduce the stem waves. In addition, given the same incident angle condition, the

stem waves in the cases of MLL-series show the largest stem width. Moreover, the widths of the stem waves tend to increase as the nonlinear property of the incident waves increases. This further demonstrates the effect of nonlinearity of incident waves on the development of stem waves as suggested by Yue and Mei (1980) and Yoon and Liu (1989).

5 **4.3 Effects of nonlinearity**

Yue and Mei (1980) proposed a single parameter, $K$ given by Eq. (8), controlling the properties of stem waves developed along a vertical wedge based on the nonlinear Schrödinger equation. The $K$ parameter represents both the nonlinearity of incident waves and the wedge slope. Yue and Mei (1980) proposed also a theoretical formula to estimate the amplitude squared of stem waves based on a simple shock model as

$$|A_\infty/a_0|^2 = \frac{1}{2K}\left[2K + 1 + \sqrt{8K + 1}\,\right], \tag{14}$$

where $A_\infty$ is the amplitude of stem waves far from the tip of wedge along the vertical wall, $a_0$ is the amplitude of incident waves. Thus, $|A_\infty/a_0|$ represents the amplification ratio of the stem waves. In Fig. 9 the normalized wave height, $H_\infty/H_0$, instead of $A_\infty/a_0$, along the vertical wall is calculated using Eq. (1), and is compared with both the measured value and the

15 theoretical one given by Eq. (14). A black solid line denotes the theoretical prediction by Yue and Mei (1980), red and blue solid lines represent the present numerical values for $\theta_0 = 10°$ and $20°$, respectively. The amplification curves obtained from the numerical calculations for $K \le 0.45$ take a long distance to reach the asymptotic value of 2 as shown in Fig. 10. Thus, this asymptotic value cannot be realized in the laboratory due to the limitation of experimental facility. However, for $K > 0.45$ the stem waves are generated and the amplification ratio increases monotonically to reach the asymptotic value in

a relatively short distance. The theoretical prediction of Yue and Mei (1980) overestimates slightly the stem heights in comparison with the measured values. The results from the present numerical simulation show good agreement with the measured values. Moreover, the present numerical results show a dependence of stem heights on the angle of incidence. This implies that $K$ is not a unique single parameter to control the property of stem waves. It is interesting to note that the maximum amplification of the stem wave is two times of the incident waves for Stokes waves, while that of solitary waves is

fourfold. This indicates that the resonant interaction between the incident and the reflected waves is weaker for the case of the Stokes waves.

It is well-known that the stem waves are generated by the nonlinear interactions between the incident and the reflected waves. When the angle between the incident and the reflected waves is small and the amplitude of two waves is small-but-finite, two waves attract each other and form a new wave with a single crest so-called the stem wave. The amplitude of the stem

wave is larger than the incident wave, and that of reflected wave is smaller. Three waves meet at a point due to both the continuous growth of the crest length of stem wave and the phase-shift of reflected wave. All the mechanisms observed in

the formation of a Mach stem wave for the solitary waves apply also for the monochromatic Stokes waves, but the intensity of nonlinear interaction is weaker than that of solitary waves.

Yue and Mei (1980) proposed the slope ratio $\beta$ of the edge line, i.e., stem boundary, of the stem region denoted by a black dashed line in Fig. 2(b) as a function of $K$ as:

$$\beta = \frac{1}{4}\left[3 + \sqrt{8K + 1}\right]. \tag{15}$$

This slope ratio $\beta$ of Yue and Mei (1980) can be converted to the angle of stem wedge $\alpha$ as:

$$\alpha = \tan^{-1}(\beta\epsilon) - \theta_0, \tag{16}$$

where $\beta\epsilon$ is the slope of the stem boundary as shown in Fig. 2(b). Fig. 11 shows the comparison of the $\alpha$-values evaluated using Eq. (16) of Yue and Mei (1980) and those determined from the numerical simulation using Eq. (12), along with the measured data determined using Eq. (11). The theoretical prediction of Yue and Mei (1980) overestimates generally the stem angle. In particular, the numerical simulations and experiments show no stem wave for the range of small $K$ less than 0.46, while the prediction of Yue and Mei (1980) still gives a nonzero stem angle. The stem angles measured in the present experiment are slightly larger than those of numerical simulation, because the experimental values are obtained in the development stage.

## 5 Comparison with solitary waves

The characteristics of stem waves developed by monochromatic Stokes waves investigated in this study are compared with those of the solitary waves.

For the comparison purposes the amplification ratio, $H_\infty/H_0$, predicted by Miles (1977) for solitary waves is calculated using the interaction parameter, $\kappa_* = \tan\theta_0/(\sqrt{3H_0/h}\cos\theta_0)$, modified by Kodama et al. (2009) as

$$\frac{H_\infty}{H_0} = \begin{cases} \dfrac{4}{1 + \sqrt{1 - \kappa_*^{-2}}}, & \text{for } \kappa_* \geq 1, \\ (1 + \kappa_*)^2, & \text{for } \kappa_* < 1. \end{cases} \tag{17}$$

The interaction parameter $\kappa_*$ is inversely proportional to $\sqrt{H_0/h}$, while the parameter $K$ is proportional to $(kH_0)^2$. To compare properly the nonlinear effects on the generation of stem waves a new parameter $K_*$ for Stokes waves is proposed as

$$K_* = \gamma K^{-1/4} \sim 1/\sqrt{kH_0},\tag{18}$$

where $\gamma$ is an arbitrary constant to adjust the scale of $K_*$. By taking $\gamma = 0.828$ for $\theta_0 = 10°$, and $\gamma = 0.805$ for $\theta_0 = 20°$ the critical condition that divides the regular and Mach reflections locates at $K_* = 1.0$ for Stokes waves. Fig. 12 shows the comparison between the amplification ratios for the present Stokes waves and the solitary waves. A black solid line denotes the amplification ratio calculated using Eq. (17) for solitary waves, while red and blue solid lines represents the amplification ratios obtained from numerical computations for the Stokes waves. The symbols denote the measured amplification ratios. As shown in the figure the amplification ratios for the Stokes waves are much smaller than those of solitary waves. And the maximum amplification ratio for the Stokes waves is 2, while that of solitary waves is 4. This indicates that the intensity of the resonant interaction between the incident and the reflected waves is much weaker than the case of the solitary waves due to strong frequency dispersion.

**6 Conclusions**

In this study, precisely controlled experiments are conducted to investigate the existence and the properties of stem waves developed along a vertical wedge for the cases of monochromatic Stokes waves. Numerical and analytical solutions are also obtained and compared with the measured data. The results obtained from this study are summarized as follows.

1. For small amplitude waves, the wave height along the wall shows slowly varying undulations with the average value of $H/H_0$=2.0. The maximum value of an undulation is approximately $H/H_0 \approx$2.3, and the distance from the tip to the location of maximum wave height decreases with increasing angle of incidence. Normalized wave heights perpendicular to the wall show a standing wave pattern. In particular, the wave height distributions for these small amplitude waves show no sign of stem wave. Both numerical and linear analytical solutions agree reasonably well with measured wave heights.

2. As the amplitude of incident waves increases, the undulation intensity decrease along the wall. For larger amplitude waves with smaller angle of incidence, i.e., larger $K$ values, the measured data show clear stem waves along the wall. Numerical simulation results are in good agreement with the results of laboratory experiments, while the analytical solution gives no stem wave, because it is linear.

3. Stem waves can be developed when the nonlinear parameter $K$ is greater than approximately 0.46. As the nonlinear parameter $K$ increases, the normalized stem height decreases and the stem width increases.

4. The resonant interactions between the incident and reflected waves predicted for solitary waves are not observed for the periodic Stokes waves. The amplification ratios along the wall do not exceed 2 for the case of Stokes waves, while those can reach fourfold for the solitary waves.

5. The existence and the properties of stem waves for sinusoidal waves found theoretically via numerical simulations are favorably supported by the physical experiments conducted in this study. Experimental data obtained in this study can be used as a useful tool to verify nonlinear dispersive wave numerical models.

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

**Table 1 Experimental wave conditions ($h = 0.25$ m).**

| Test case | Wave period $T$ (s) | Wave height $H_0$ (m) | Incident angle $\theta_0$ (deg.) | Nonlinearity | |
|---|---|---|---|---|---|
| | | | | Wave steepness $kH_0$ | Nonlinear parameter $K$ |
| MSS1 | 0.7 | 0.009 | 10 | 0.076 | 0.088 |
| MSS2 | | | 20 | | 0.021 |
| MSS3 | | | 30 | | 0.008 |
| MSS4 | | | 40 | | 0.004 |
| MSM1 | | 0.027 | 10 | 0.229 | 0.793 |
| MSM2 | | | 20 | | 0.186 |
| MSM3 | | | 30 | | 0.074 |
| MSM4 | | | 40 | | 0.035 |
| MSL1 | | 0.036 | 10 | 0.305 | 1.411 |
| MSL2 | | | 20 | | 0.331 |
| MSL3 | | | 30 | | 0.132 |
| MSL4 | | | 40 | | 0.062 |
| MLS1 | 1.1 | 0.018 | 10 | 0.076 | 0.123 |
| MLS2 | | | 20 | | 0.029 |
| MLS3 | | | 30 | | 0.011 |
| MLS4 | | | 40 | | 0.005 |
| MLM1 | | 0.054 | 10 | 0.228 | 1.108 |
| MLM2 | | | 20 | | 0.260 |
| MLM3 | | | 30 | | 0.103 |
| MLM4 | | | 40 | | 0.049 |
| MLL1 | | 0.072 | 10 | 0.304 | 1.969 |
| MLL2 | | | 20 | | 0.462 |
| MLL3 | | | 30 | | 0.184 |
| MLL4 | | | 40 | | 0.087 |

**Table 2** Measuring points in hydraulic experiments.

| Wave period ($T$) | $x$-dir. (along the wall) | $y$-dir. (normal to the wall) | |
| --- | --- | --- | --- |
| | | at $x/L = 6$ | at $x/L = 15$ |
| 0.7 s | $x = 0.0$ m~11.4 m ($\Delta x = 0.2$ m) | $y = 0.1$ m~3.7 m ($\Delta y = 0.1$ m) | |
| 1.1 s | $x = 0.0$ m~22.8 m ($\Delta x = 0.4$ m) | $y = 0.2$ m~7.3 m ($\Delta y = 0.2$ m) | |

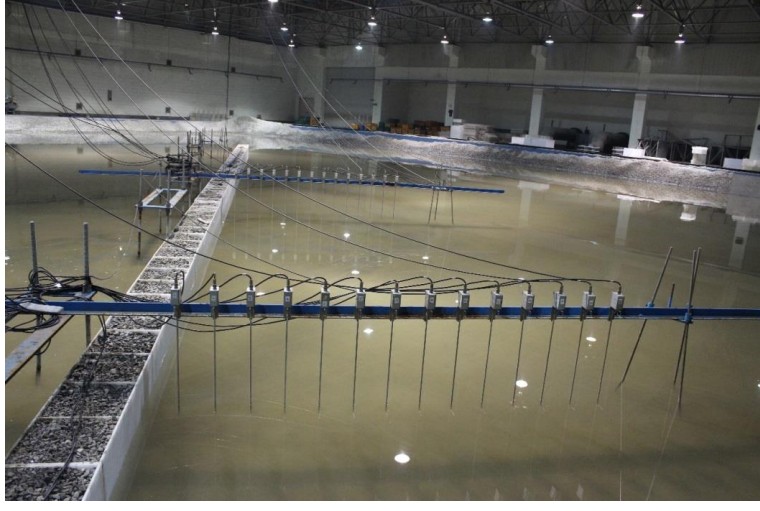

**Photo 1. Experimental facility and wave gauge array.**

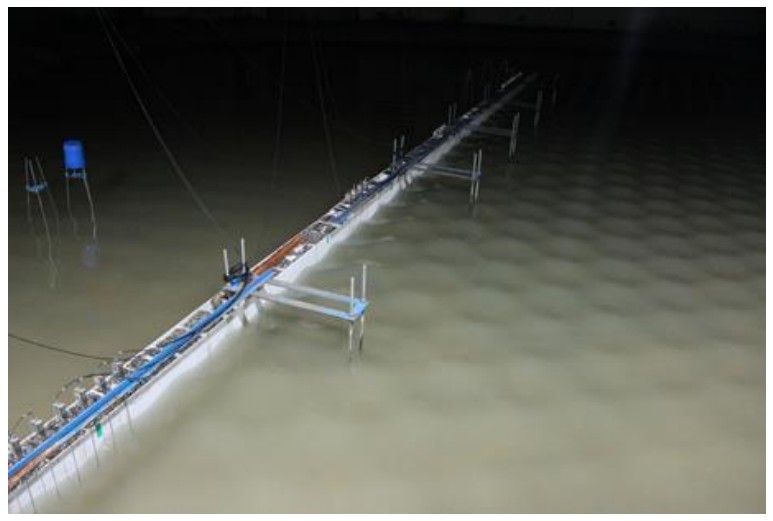

**Photo 2. Wave pattern in front of a vertical wall ($\theta_0 = 30°$).**

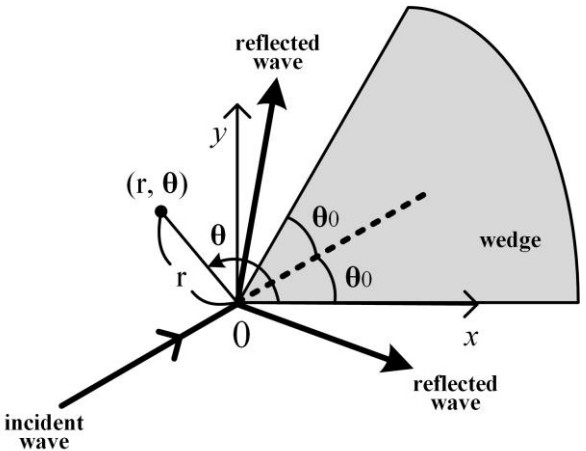

**Figure 1. Definition sketch of wave field around a vertical wedge.**

**Figure 2. Coordinate system for numerical simulations: (a) present, (b) Yue & Mei (1980).**

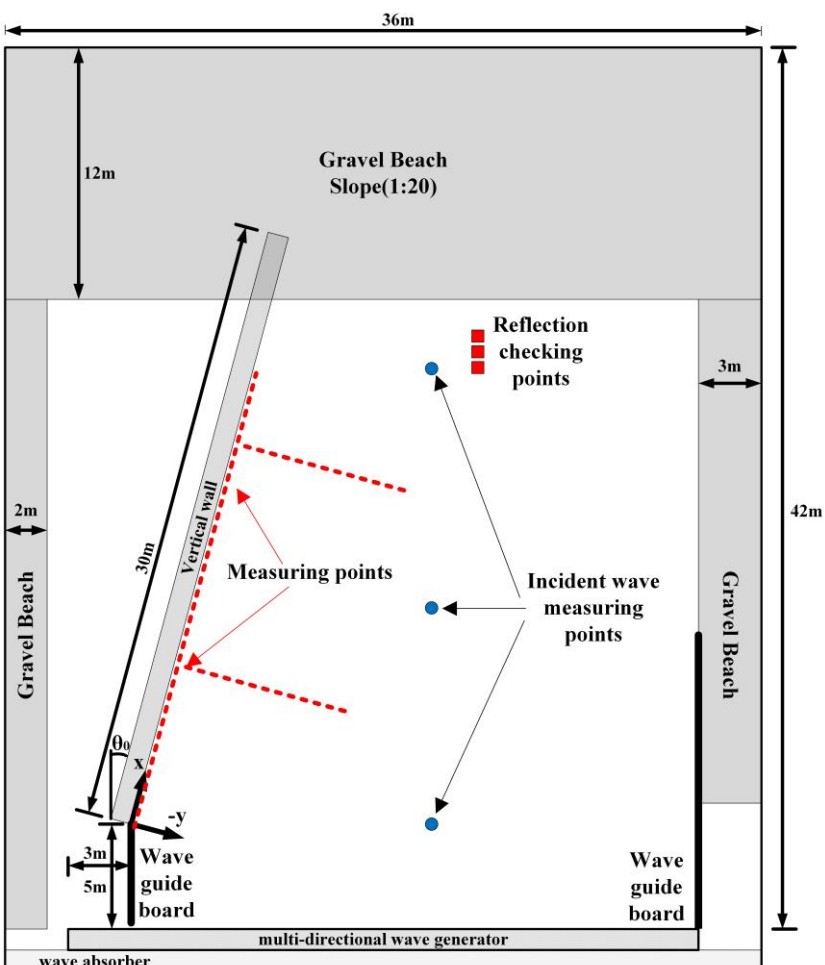

**Figure 3. Definition sketch of the experimental setup.**

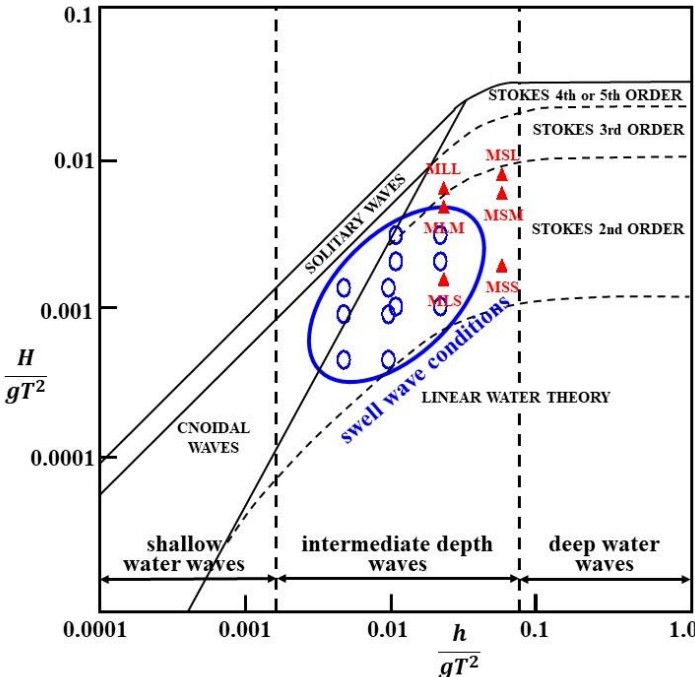

**Figure 4.** The present experiment and wave conditions of the real-world cases (after Le Méhauté, 1976). The solid red triangles represent the incident waves tested in this study and empty blue circles represent the swell wave conditions. The *x*-axis represents the relative water depth (ratio of water depth to deep water wave length, i.e., the measure of wave dispersion). The *y*-axis represents the wave steepness (ratio of wave height to deep water wave length, i.e., the measure of wave nonlinearity).

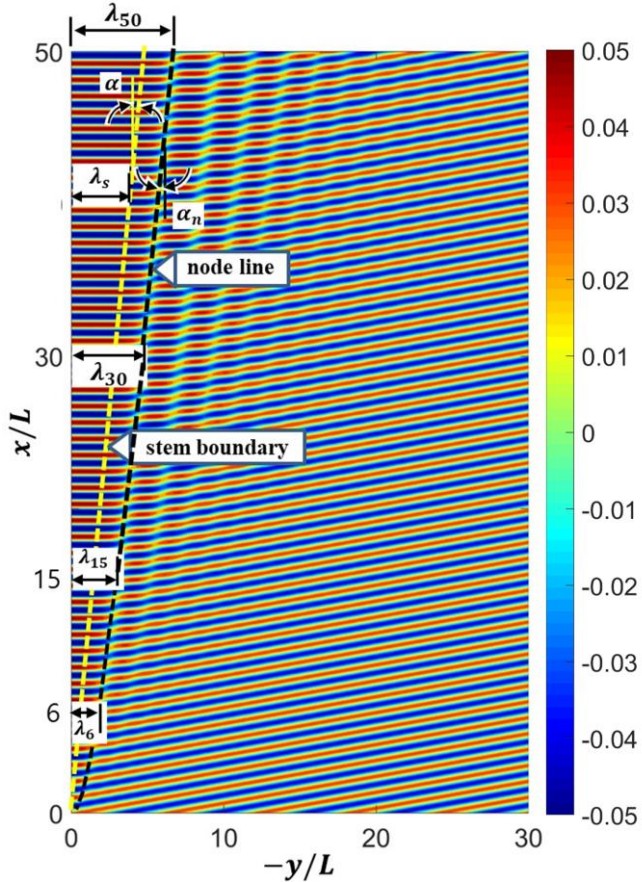

**Figure 5. Definition sketch for the stem angle and the stem boundary.**

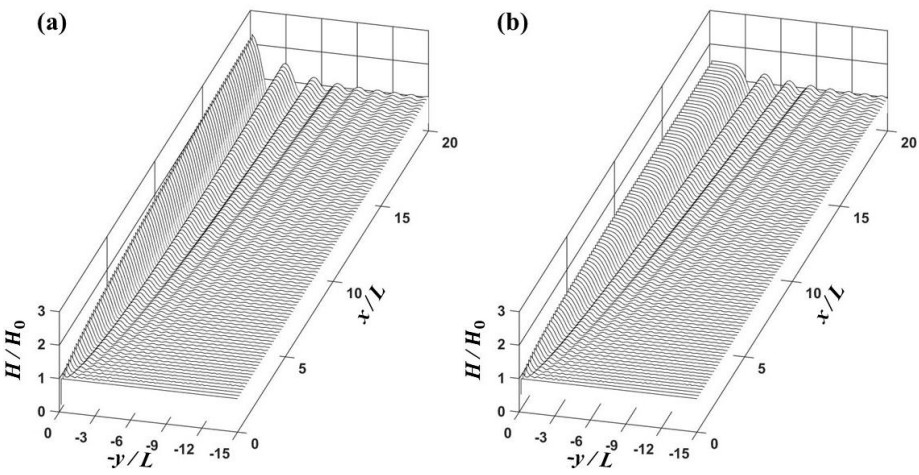

5        **Figure 6. Three-dimensional plots of normalized wave height for (a) MLS1 and (b) MLL1 cases from simulation.**

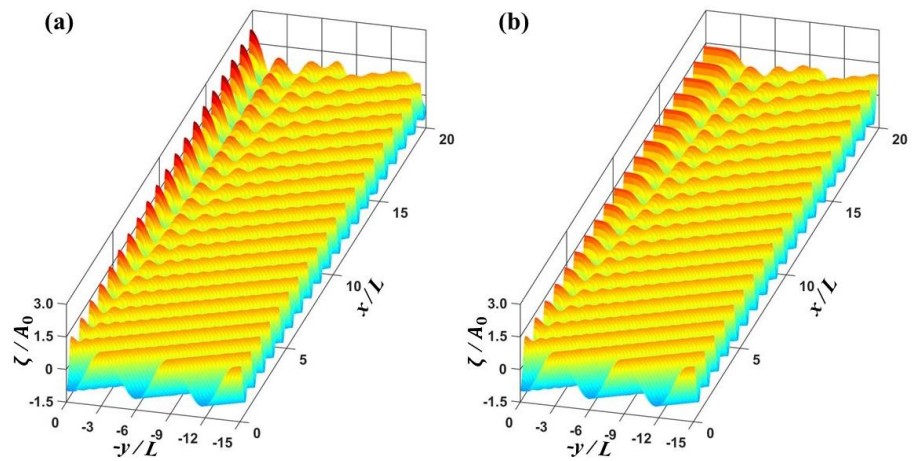

**Figure 7. Three-dimensional plots of normalized free surface displacements (a) MLS1 and (b) MLL1 cases from simulation.**

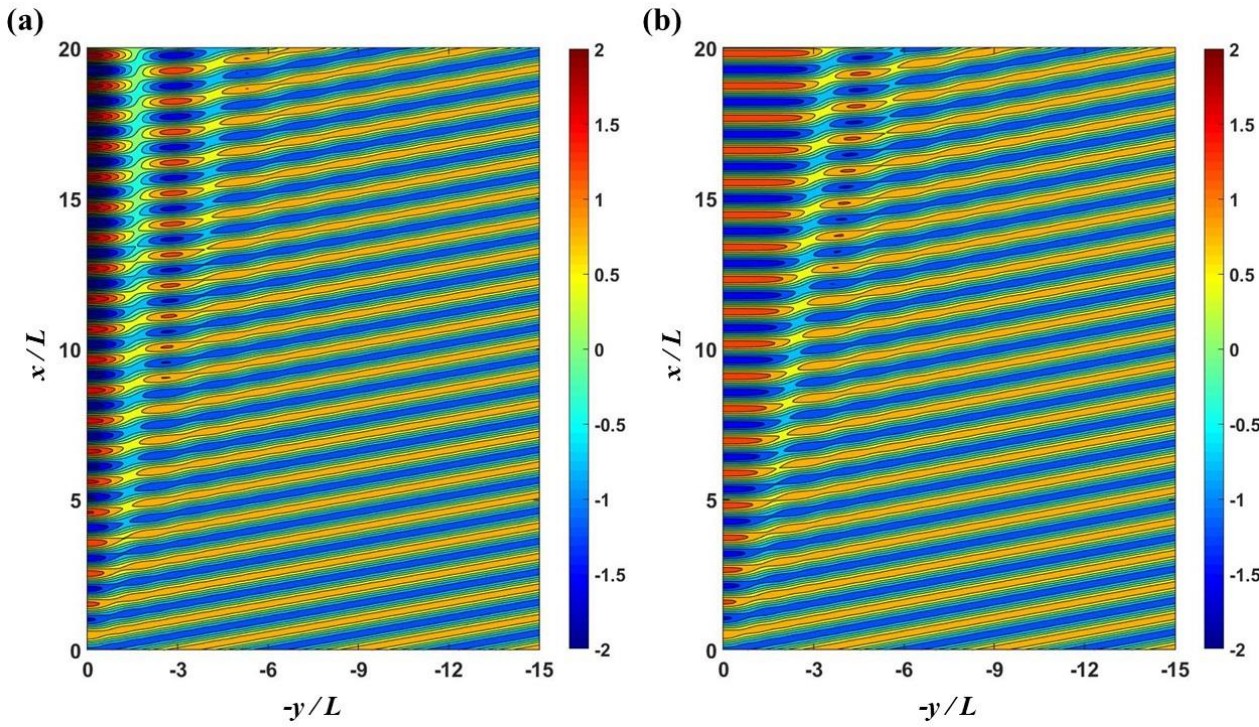

**Figure 8. Contour plots of the instantaneous normalized free surface for (a) MLS1 and (b) MLL1 cases from simulation.**

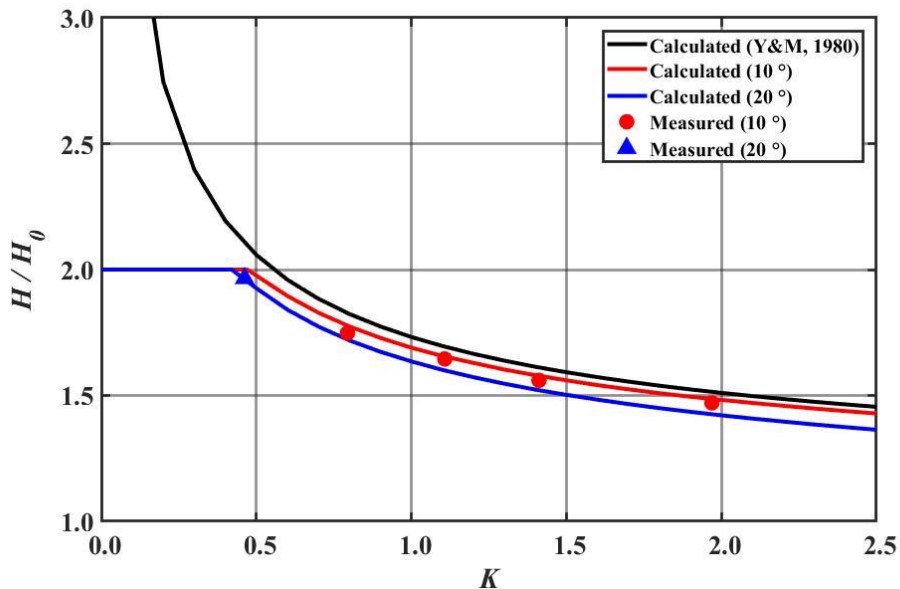

**Figure 9. Comparison of calculated and measured normalized wave heights along the wall as a function of nonlinear parameter $K$. Black solid curve represents the wave height predicted by shock theory of Yue and Mei (1980), red and blue solid curves denote the calculated wave heights for $\theta_0 = 10°$ and $20°$, respectively. Symbols are measured data.**

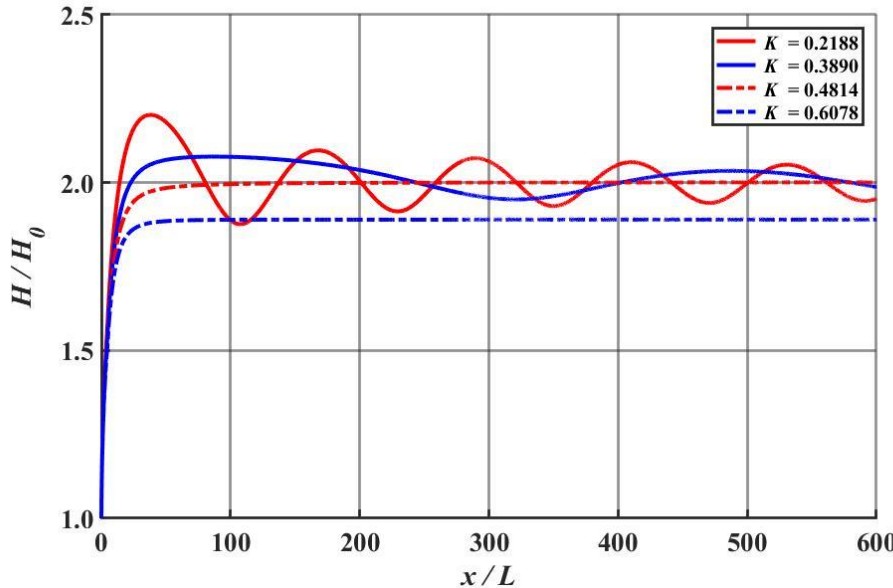

**Figure 10. Comparison of calculated normalized wave heights along the wall for various nonlinear parameter values of $K$ ($\theta_0 = 10°$).**

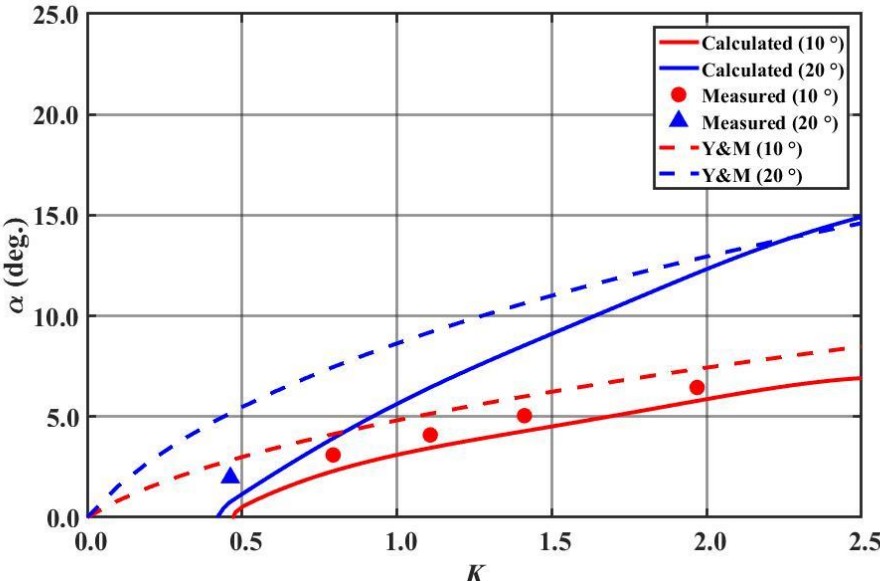

**Figure 11. Comparison of calculated and measured stem angle $\alpha$ as a function of nonlinear parameter $K$. Dashed curves represent the calculated values using Yue and Mei (1980), solid curves are the calculated values using Eq. (12), symbols are measured data. Red and blue colors are for $\theta_0 = 10°$ and $20°$, respectively.**

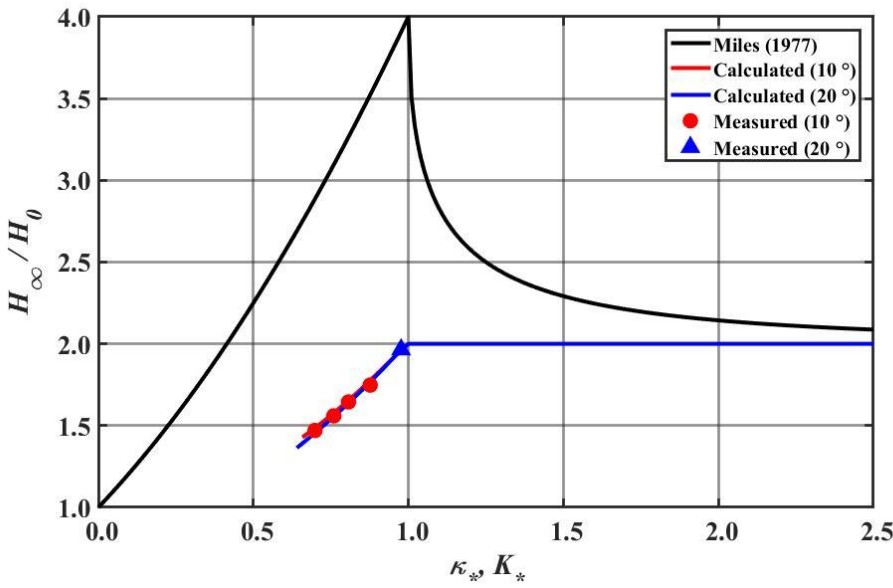

**Figure 12. Comparison of amplification ratios, $H_\infty/H_0$, as a function of nonlinear parameter $\kappa_*$ for solitary waves and $K_*$ for Stokes waves. Black solid curve represents the Miles' solution for solitary waves, red and blue solid curves denote the calculated values for Stokes waves for $\theta_0 = 10°$ and $20°$, respectively. Symbols are measured data for Stokes waves.**

**Appendix**

All the figures for the experimental and calculated data are presented in this Appendix.

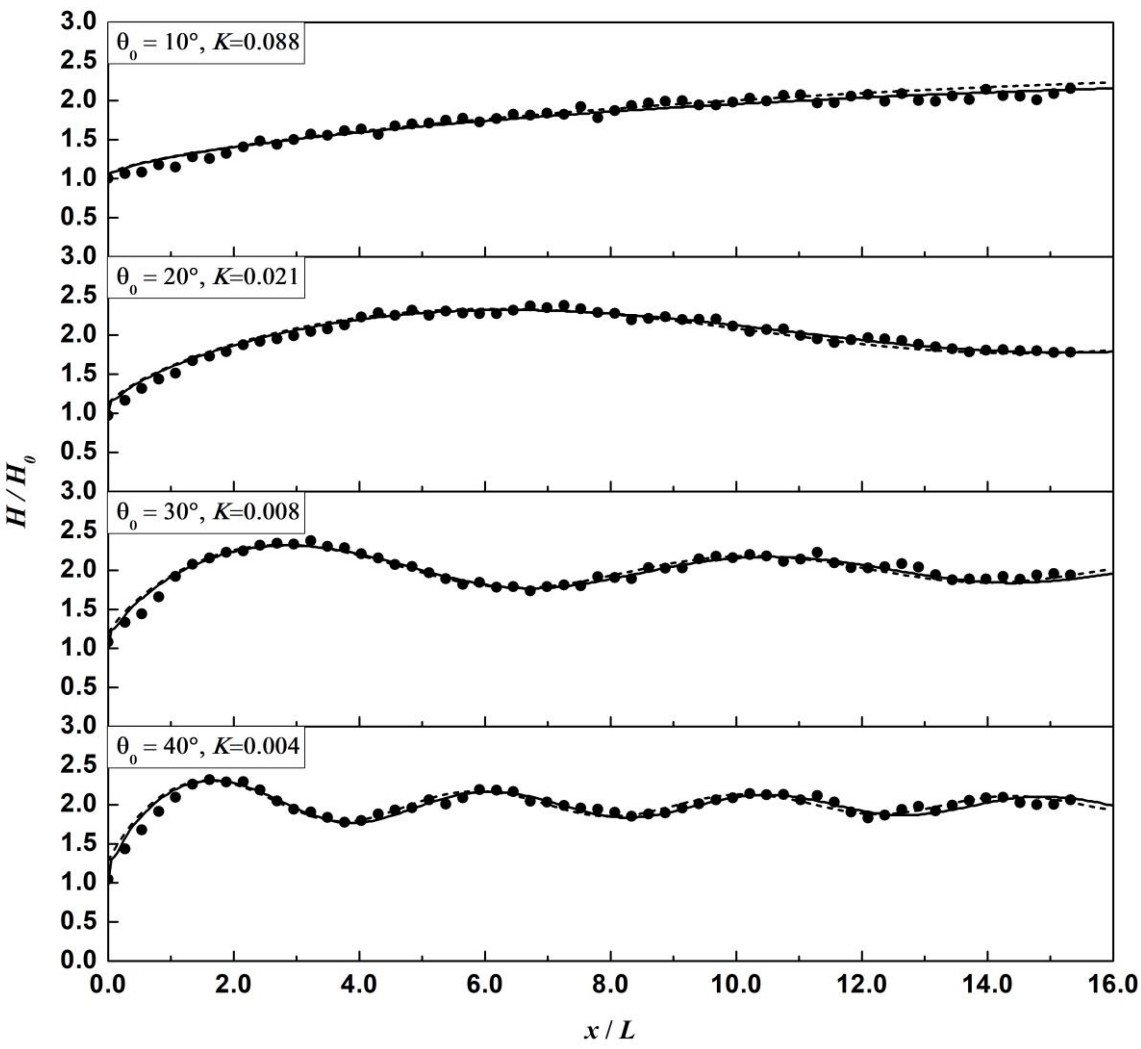

**Figure A1. Normalized wave heights along the wall for the cases of MSS1 ~ MSS4. Solid circle: measured, solid line: present numerical, dashed line: analytical (Chen, 1987).**

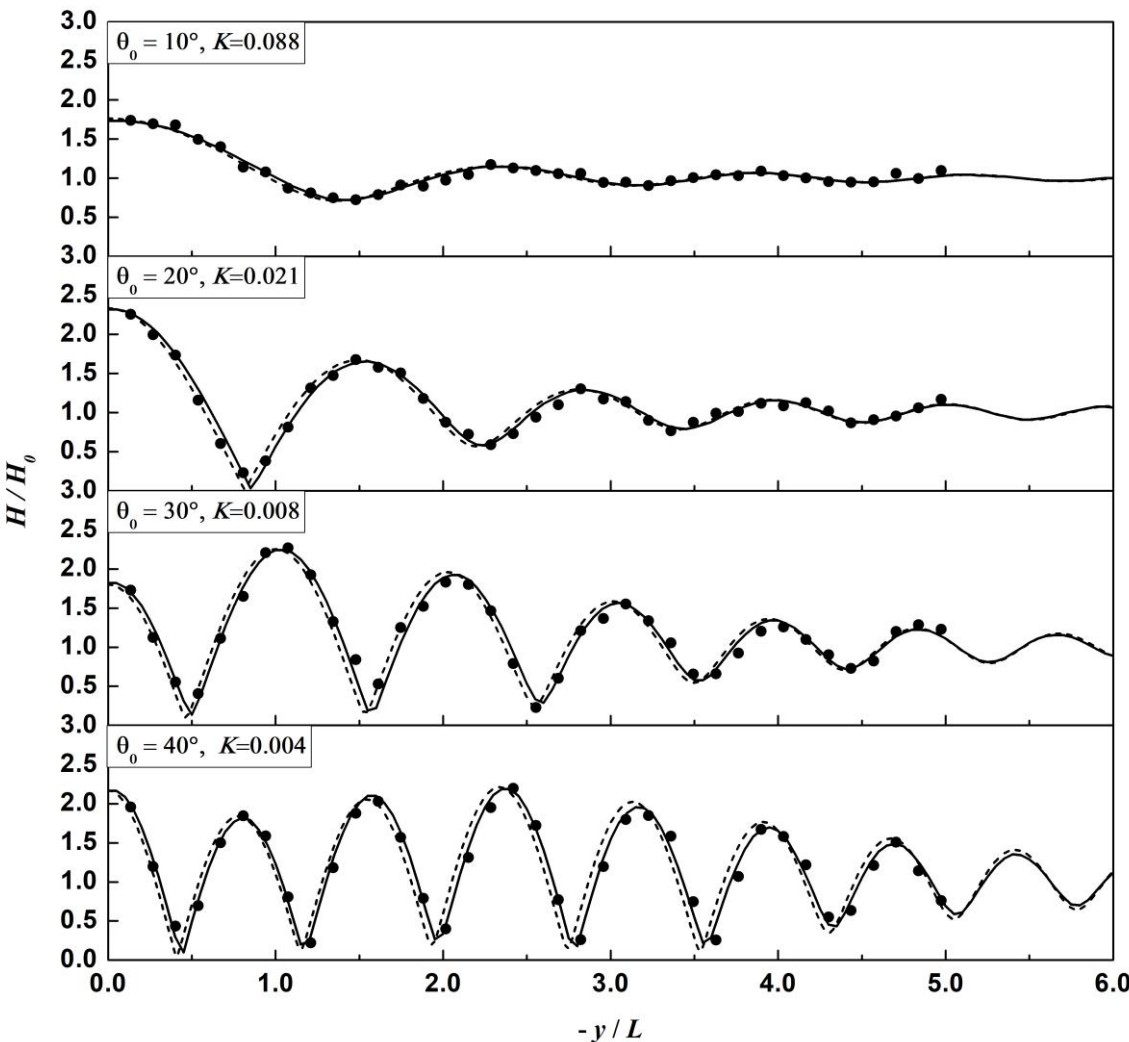

**Figure A2. Normalized wave heights normal to the wall at** *x* **= 6***L* **for the cases of MSS1 ~ MSS4. Solid circle: measured, solid line: present numerical, dashed line: analytical (Chen, 1987).**

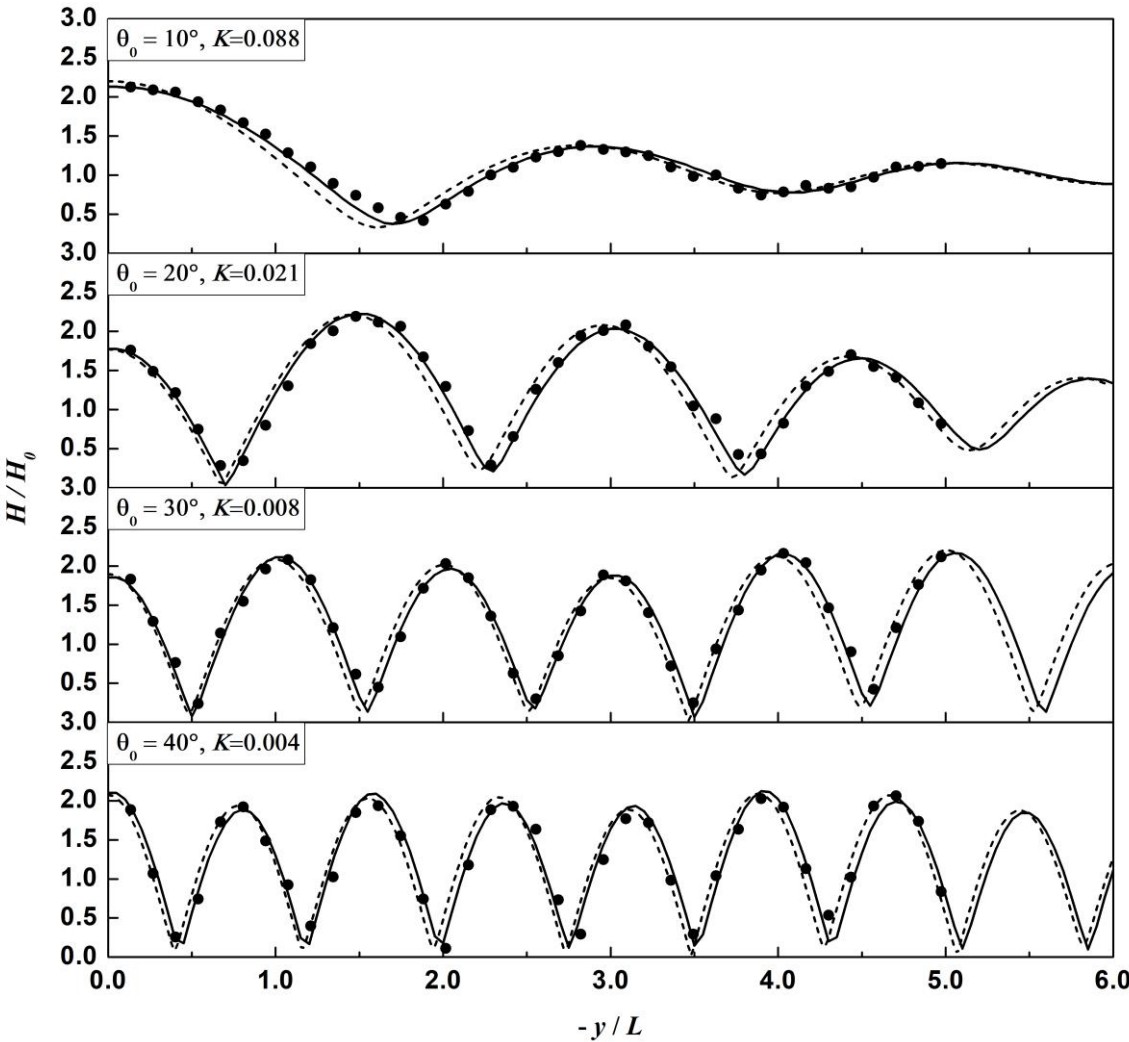

**Figure A3. Normalized wave heights normal to the wall at *x* = 15*L* for the cases of MSS1 ~ MSS4. Solid circle: measured, solid line: present numerical, dashed line: analytical (Chen, 1987).**

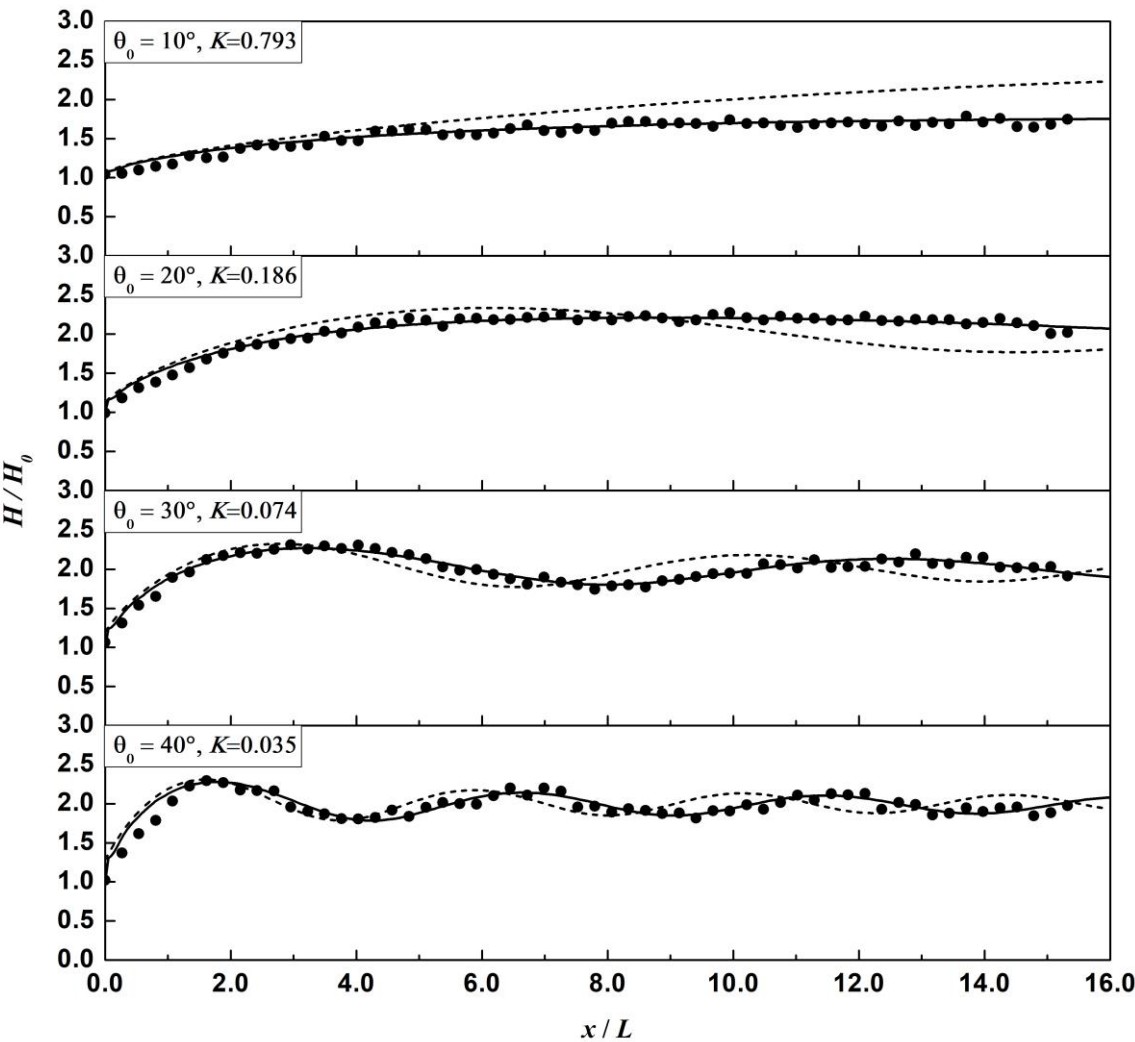

**Figure A4. Normalized wave heights along the wall for the cases of MSM1 ~ MSM4. Solid circle: measured, solid line: present numerical, dashed line: analytical (Chen, 1987).**

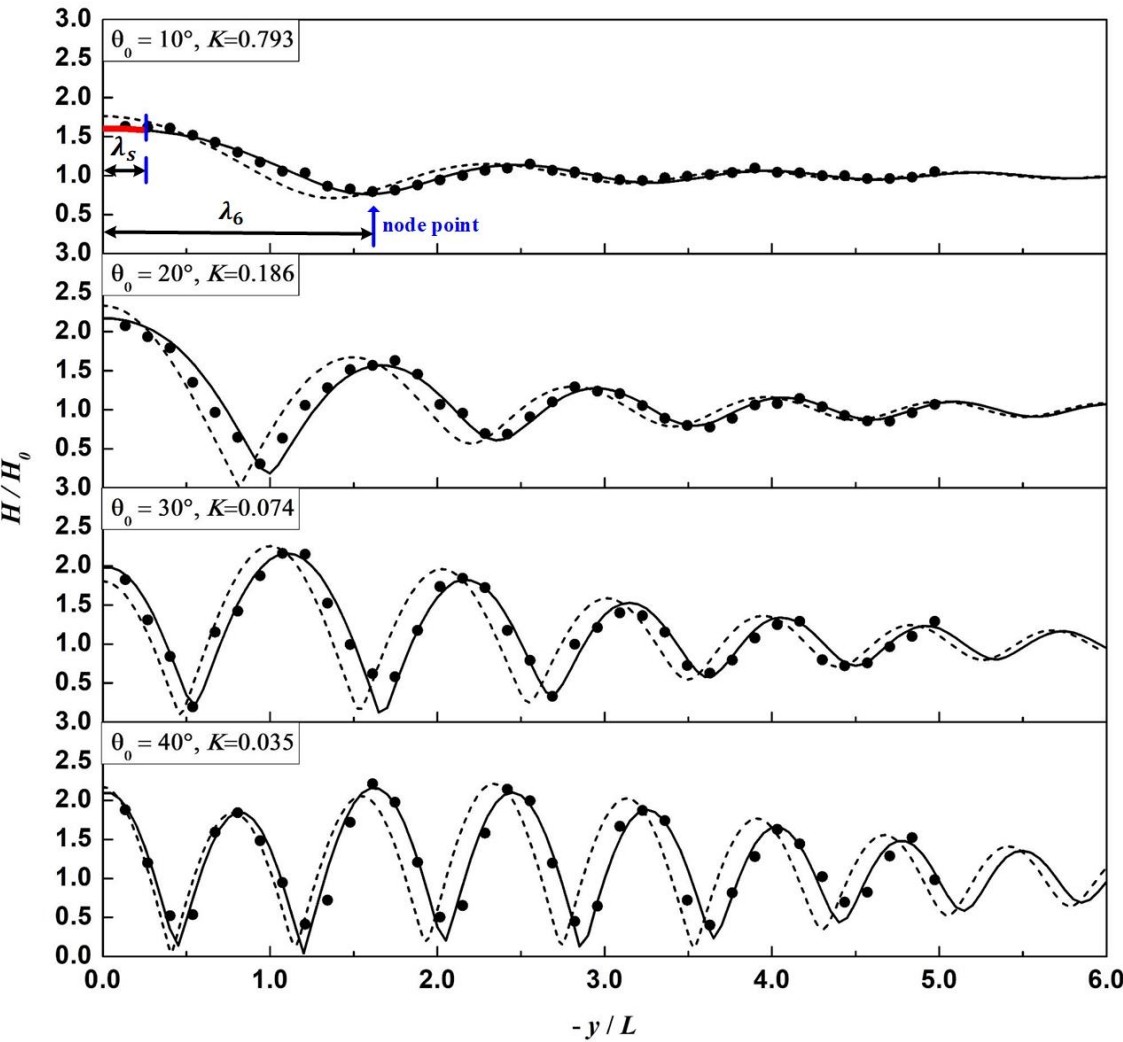

**Figure A5. Normalized wave heights normal to the wall at $x = 6L$ for the cases of MSM1 ~ MSM4. Solid circle: measured, solid line: present numerical, dashed line: analytical (Chen, 1987). The red line represents the stem waves. The stem width $\lambda_s$ is determined using Eq. (13).**

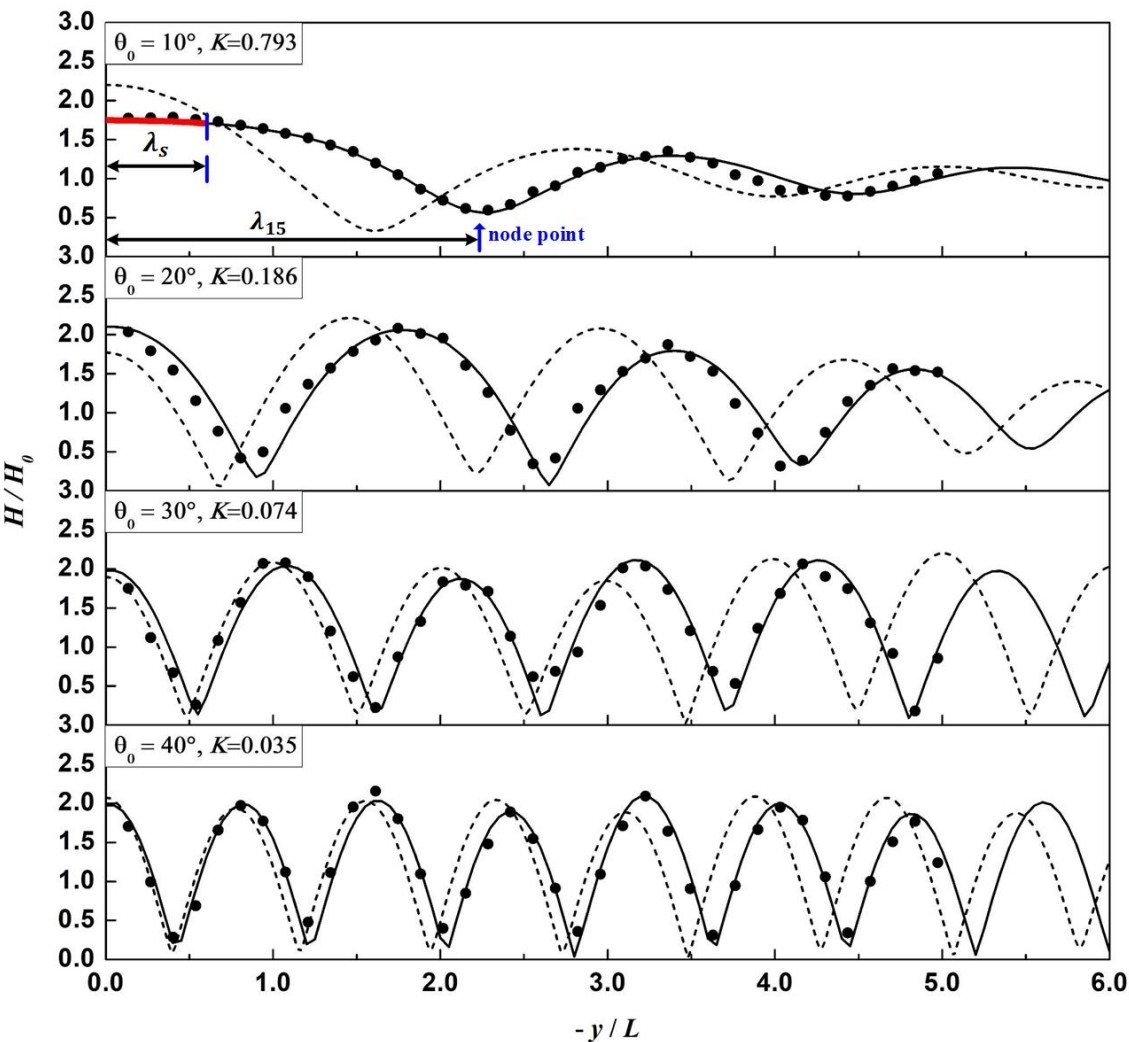

**Figure A6.** Normalized wave heights normal to the wall at $x = 15L$ for the cases of MSM1 ~ MSM4. Solid circle: measured, solid line: present numerical, dashed line: analytical (Chen, 1987). The red line represents the stem waves. The stem width $\lambda_s$ is determined using Eq. (13).

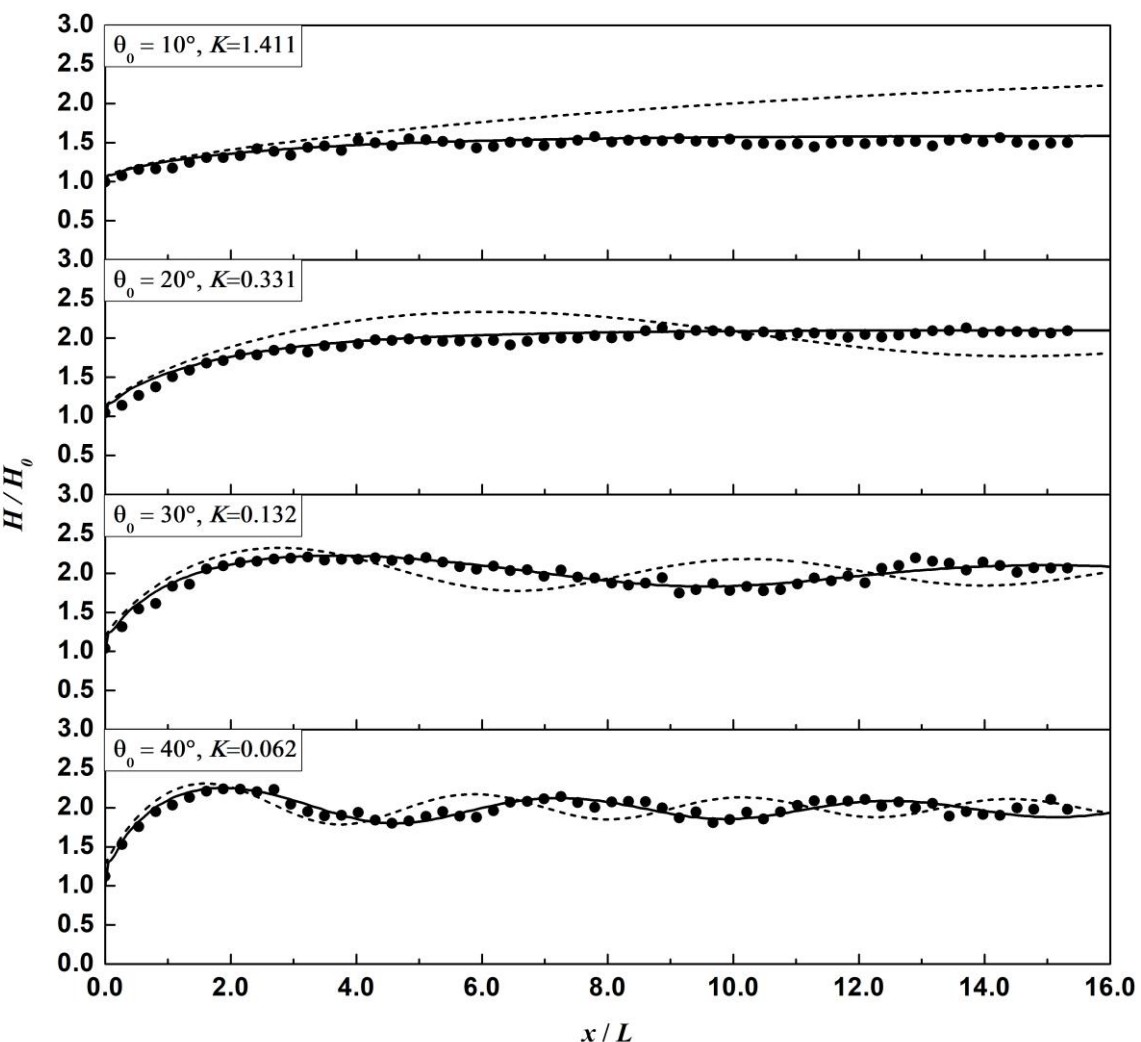

**Figure A7. Normalized wave heights along the wall for the cases of MSL1 ~ MSL4. Solid circle: measured, solid line: present numerical, dashed line: analytical (Chen, 1987).**

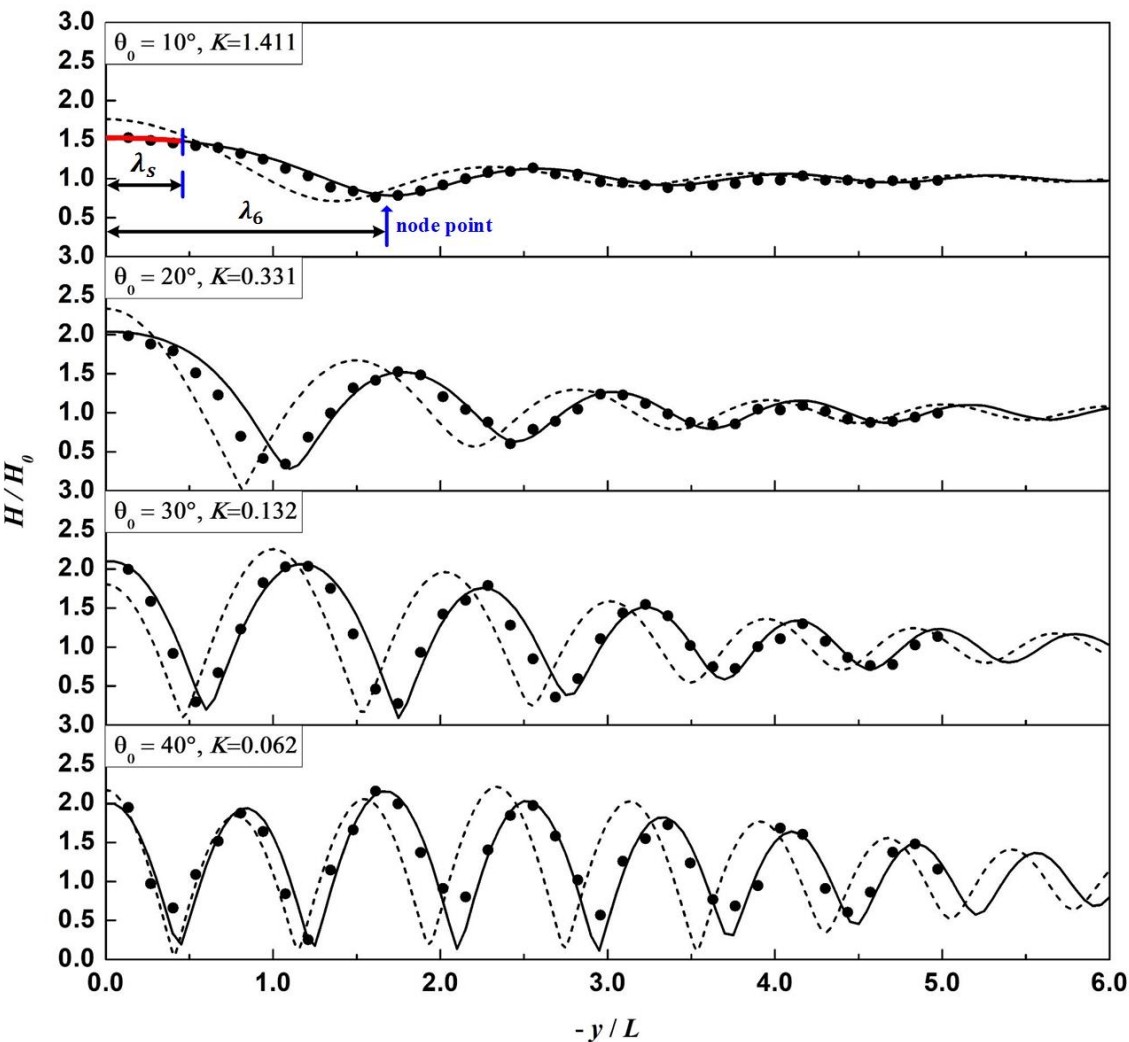

**Figure A8.** Normalized wave heights normal to the wall at $x = 6L$ for the cases of MSL1 ~ MSL4. Solid circle: measured, solid line: present numerical, dashed line: analytical (Chen, 1987). The red line represents the stem waves. The stem width $\lambda_s$ is determined using Eq. (13).

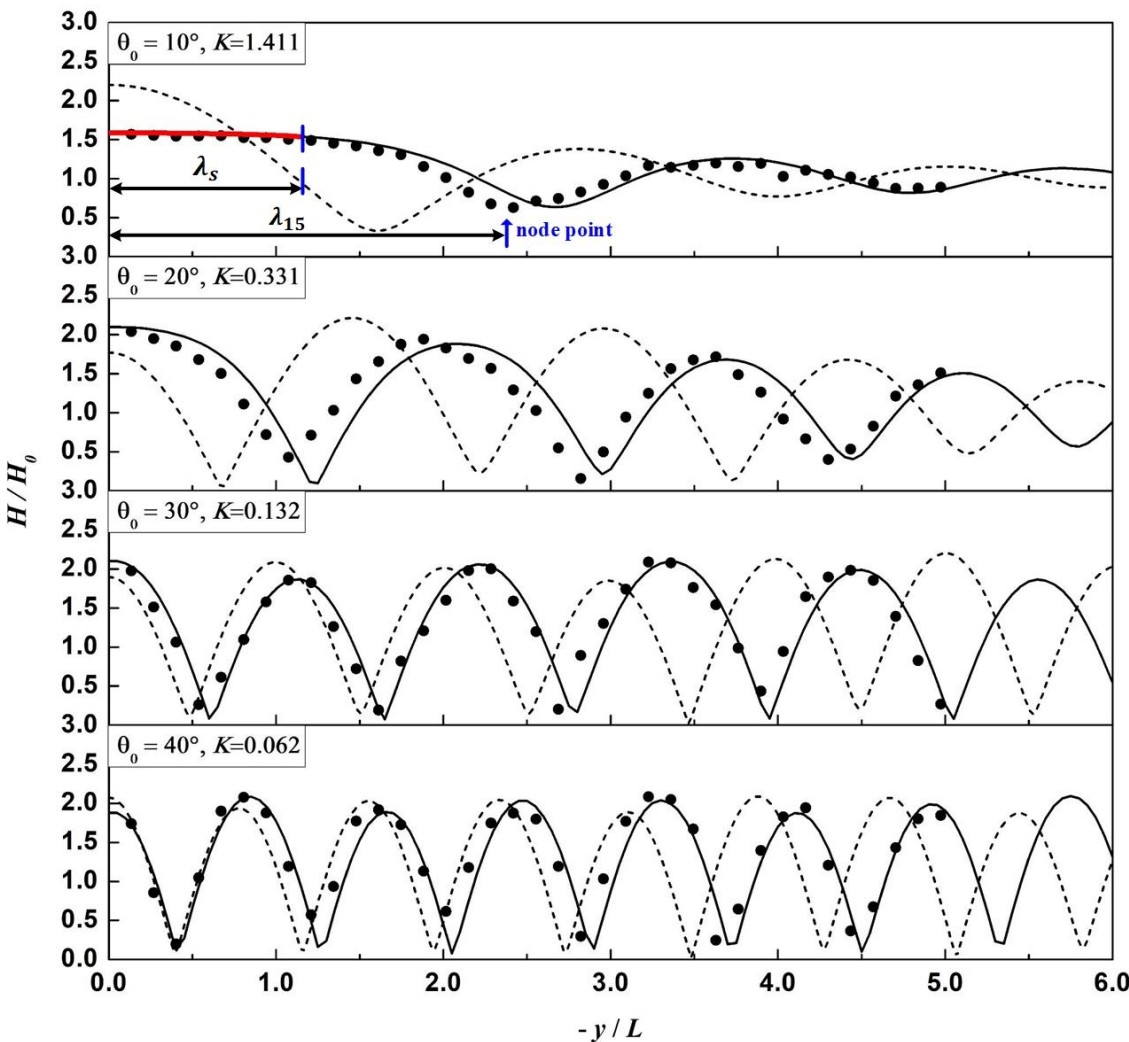

**Figure A9. Normalized wave heights normal to the wall at $x = 15L$ for the cases of MSL1 ~ MSL4. Solid circle: measured, solid line: present numerical, dashed line: analytical (Chen, 1987). The red line represents the stem waves. The stem width $\lambda_s$ is determined using Eq. (13).**

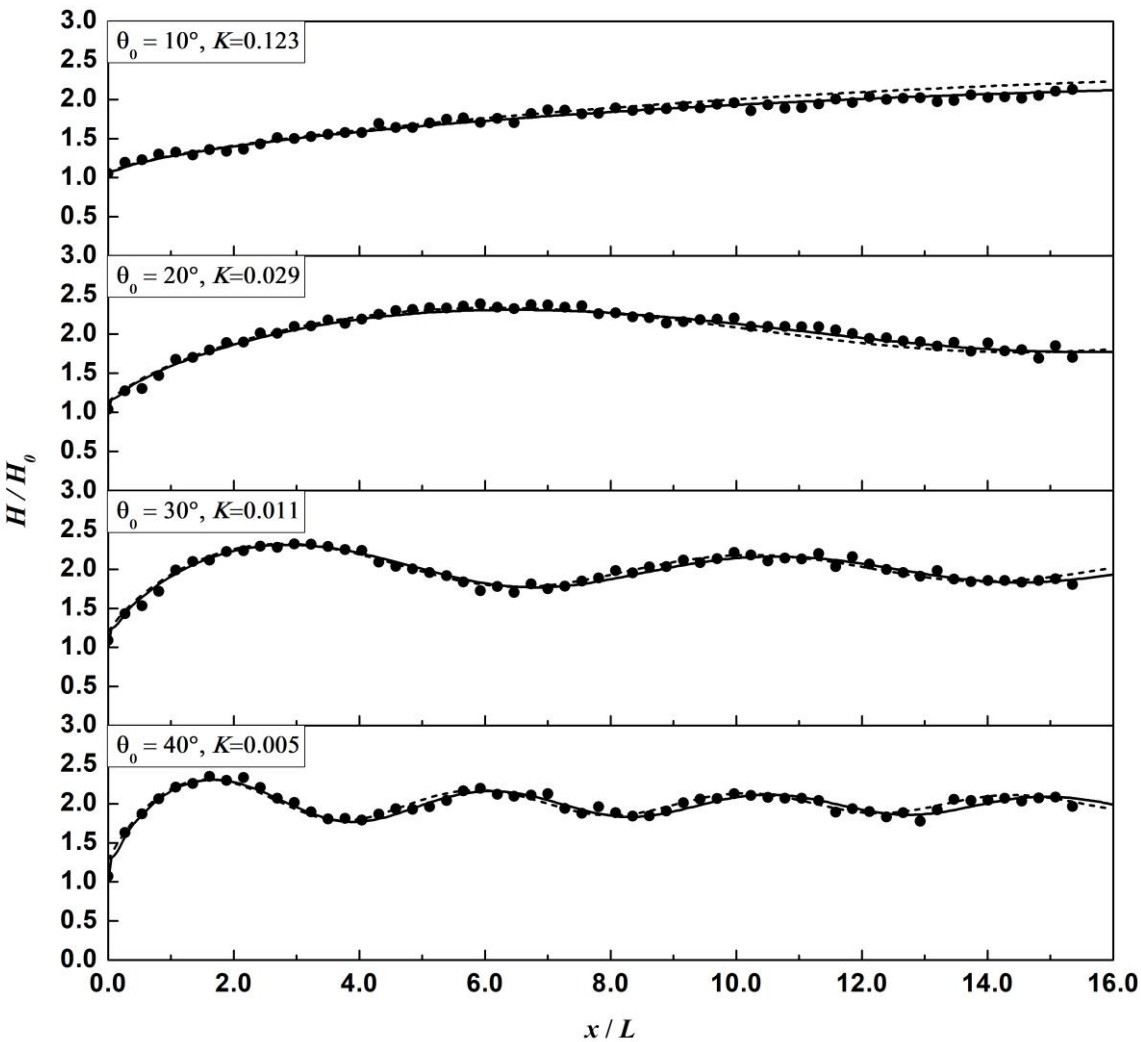

**Figure A10. Normalized wave heights along the wall for the cases of MLS1 ~ MLS4. Solid circle: measured, solid line: present numerical, dashed line: analytical (Chen, 1987).**

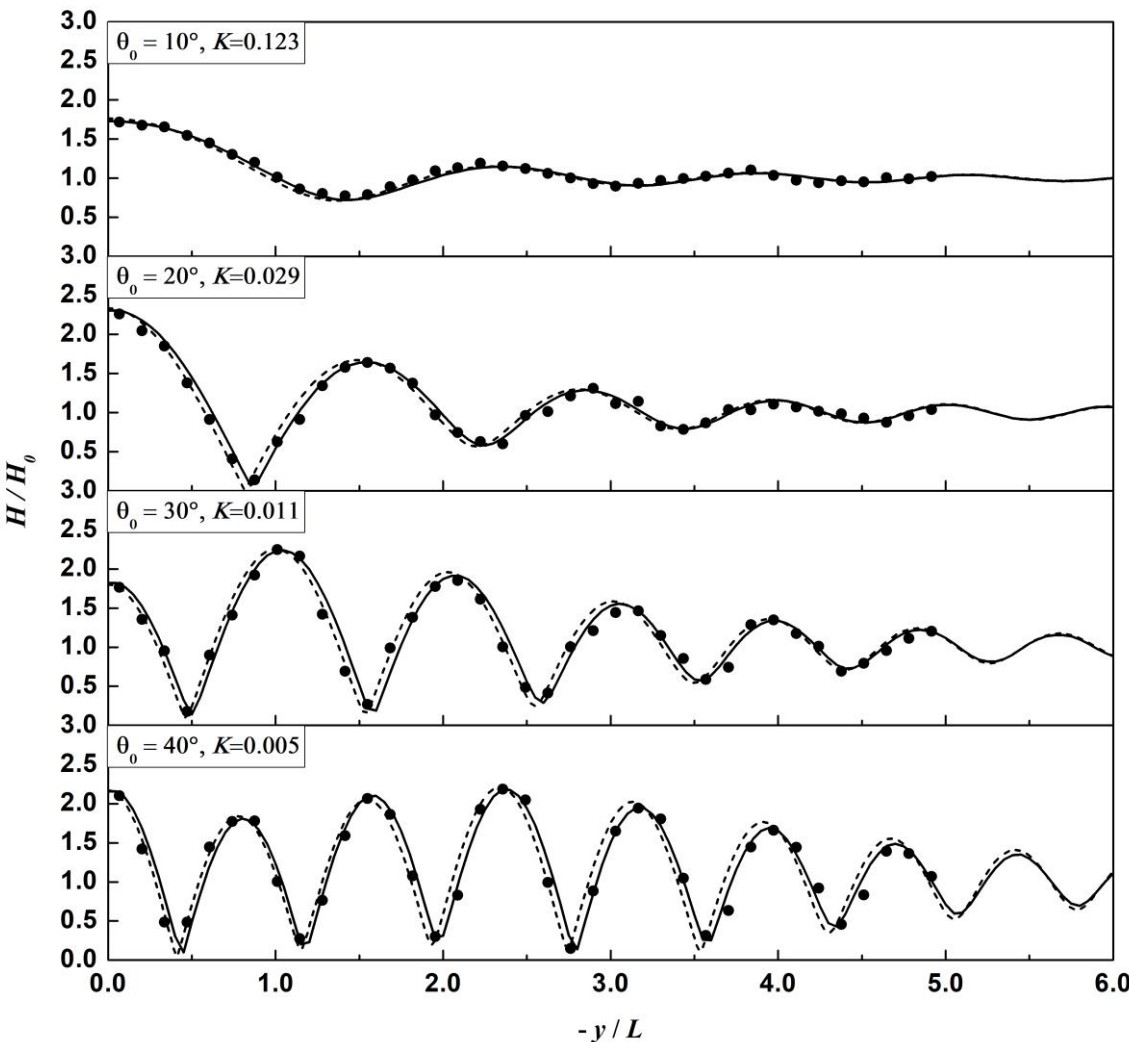

**Figure A11. Normalized wave heights normal to the wall at $x = 6L$ for the cases of MLS1 ~ MLS4. Solid circle: measured, solid line: present numerical, dashed line: analytical (Chen, 1987).**

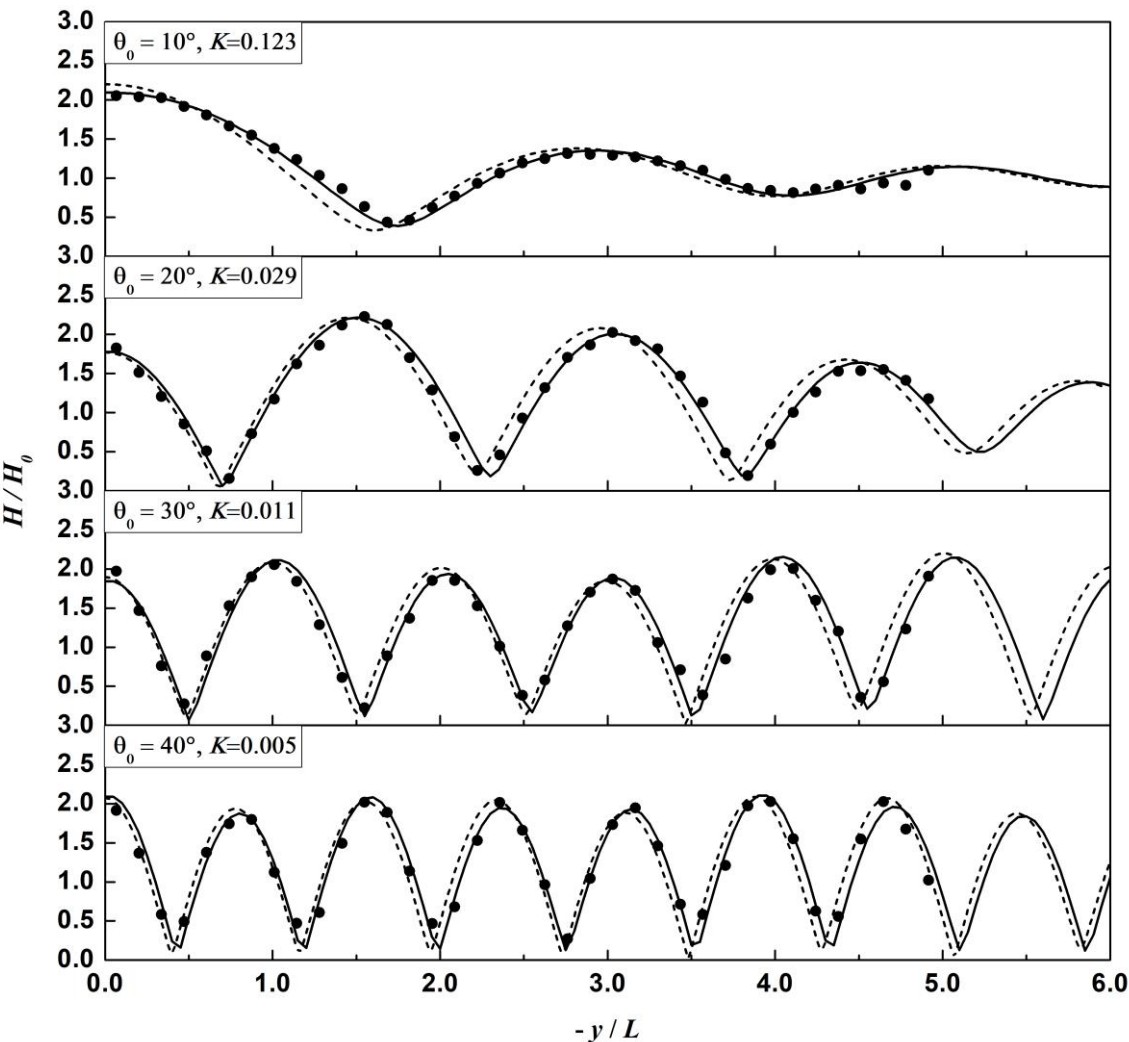

**Figure A12. Normalized wave heights normal to the wall at** $x = 15L$ **for the cases of MLS1 ~ MLS4. Solid circle: measured, solid line: present numerical, dashed line: analytical (Chen, 1987).**

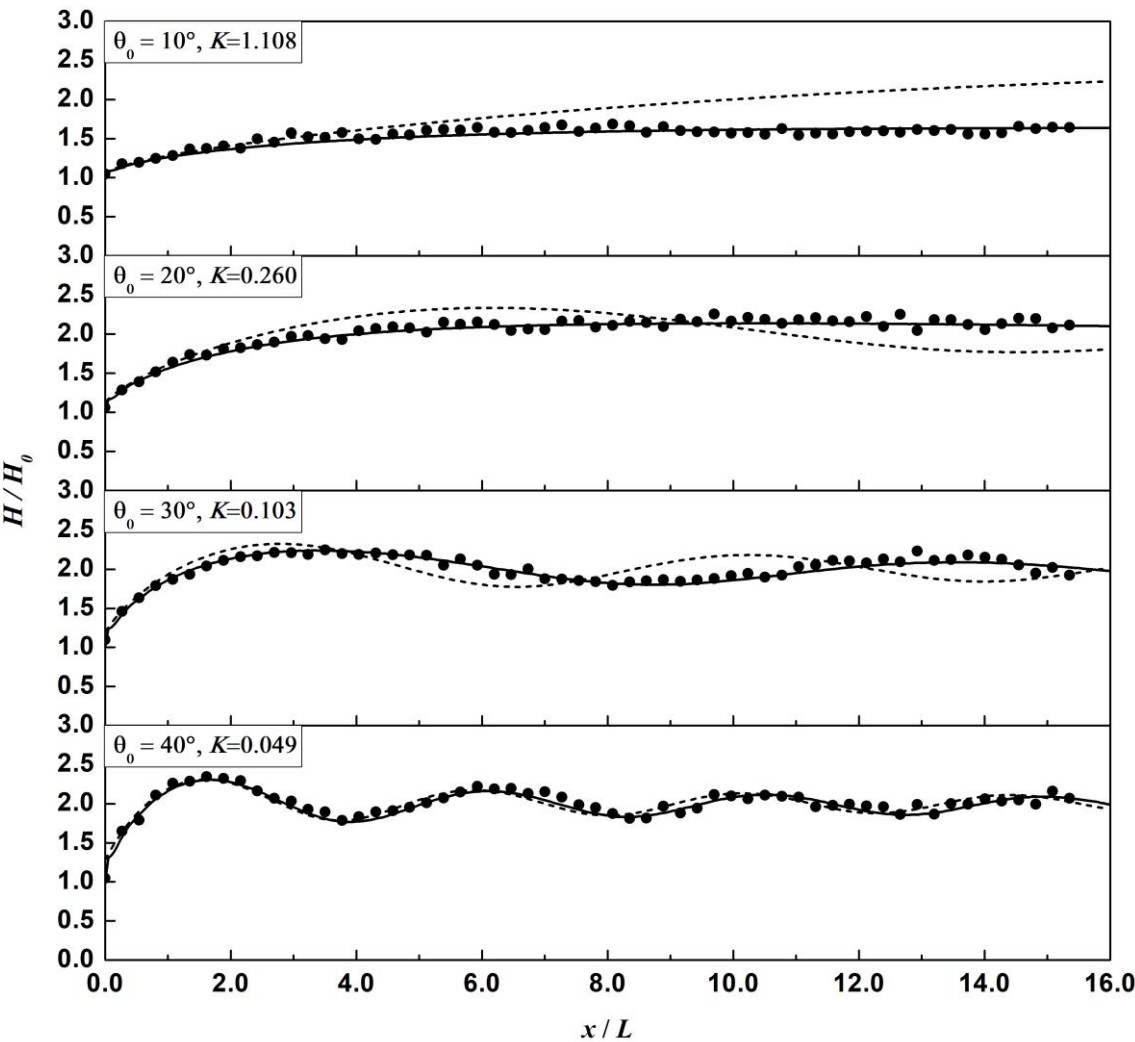

**Figure A13. Normalized wave heights along the wall for the cases of MLM1 ~ MLM4. Solid circle: measured, solid line: present numerical, dashed line: analytical (Chen, 1987).**

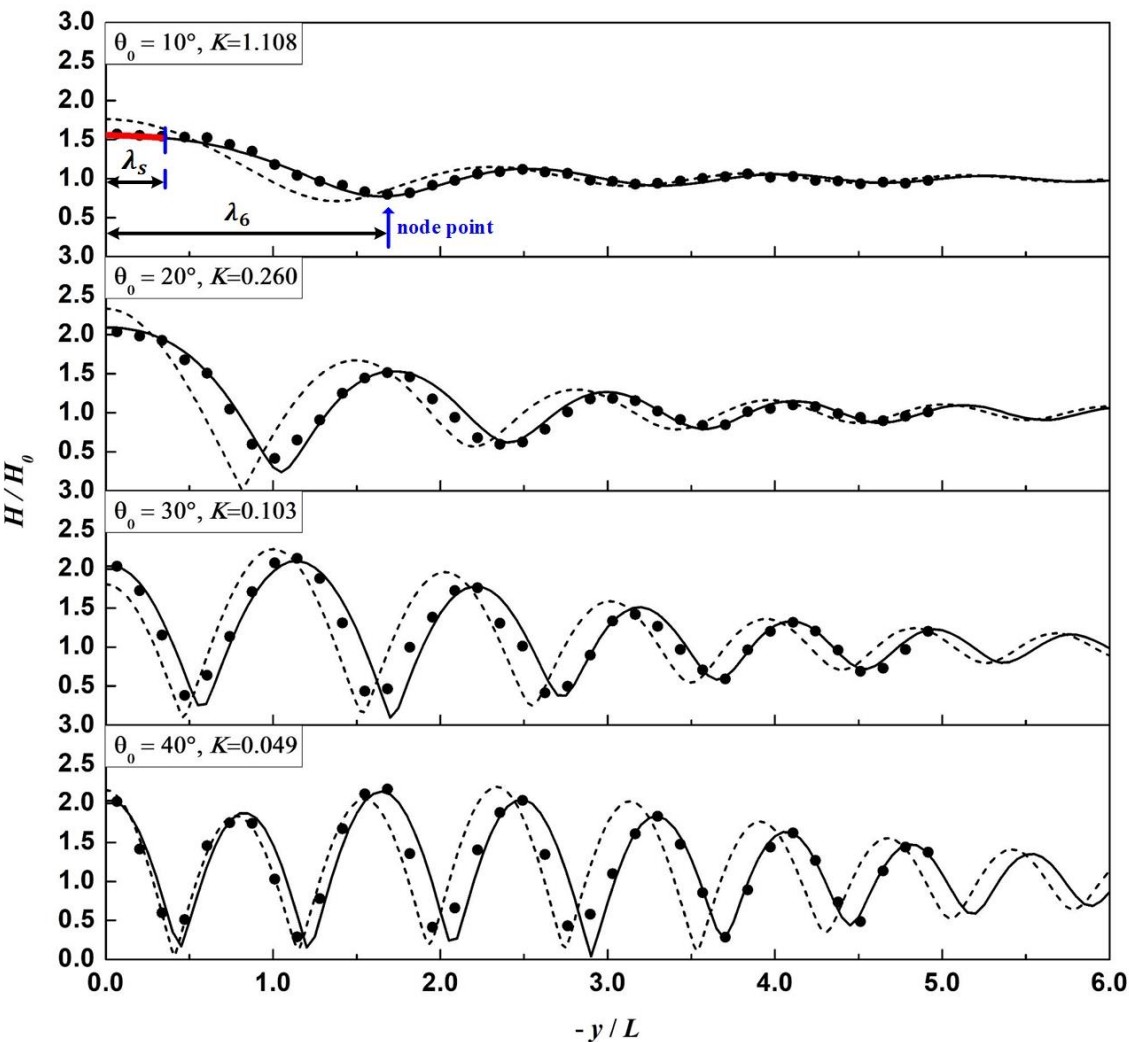

**Figure A14. Normalized wave heights normal to the wall at $x = 6L$ for the cases of MLM1 ~ MLM4. Solid circle: measured, solid line: present numerical, dashed line: analytical (Chen, 1987). The red line represents the stem waves. The stem width $\lambda_s$ is determined using Eq. (13).**

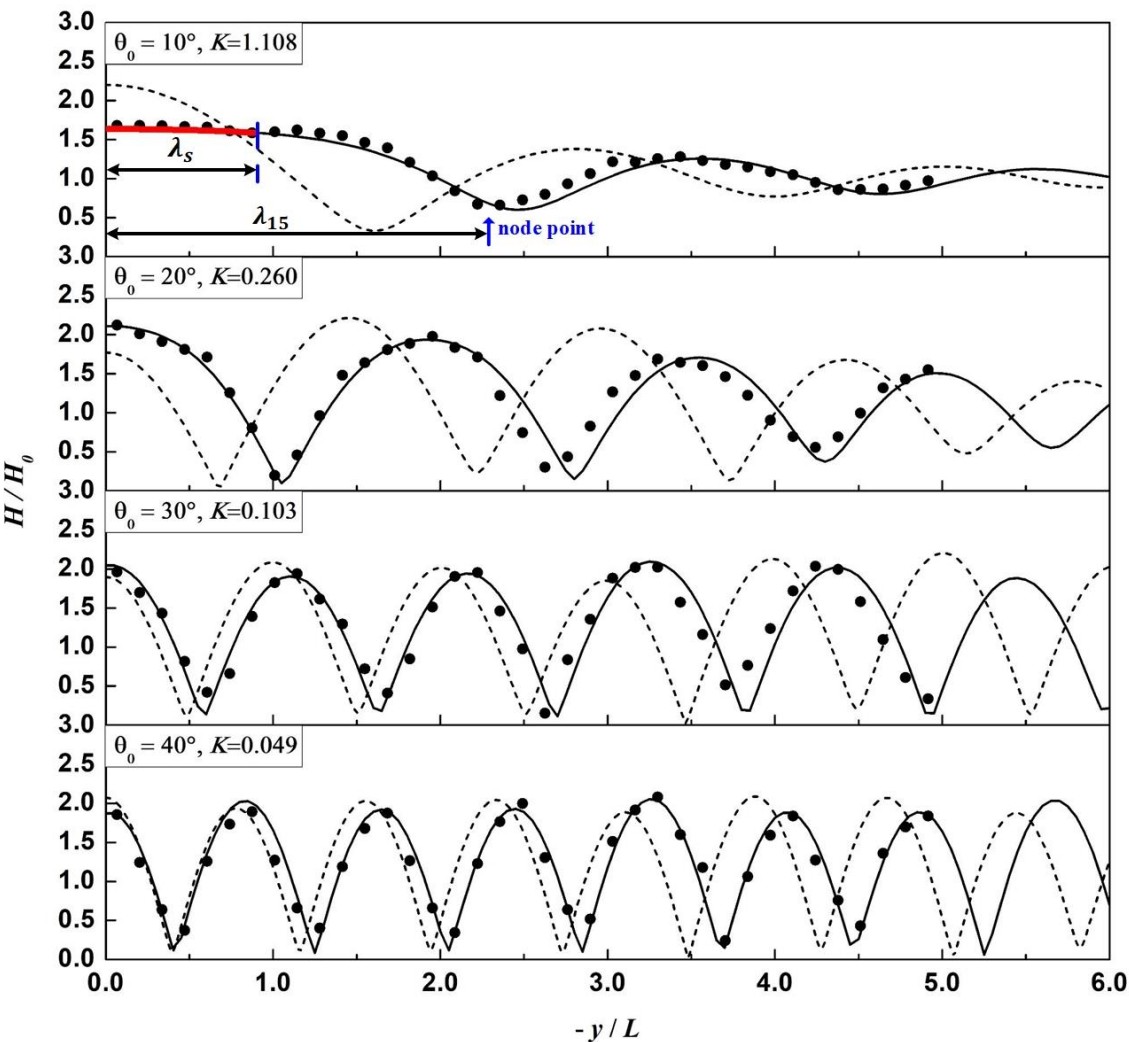

**Figure A15. Normalized wave heights normal to the wall at $x = 15L$ for the cases of MLM1 ~ MLM4. Solid circle: measured, solid line: present numerical, dashed line: analytical (Chen, 1987). The red line represents the stem waves. The stem width $\lambda_s$ is determined using Eq. (13).**

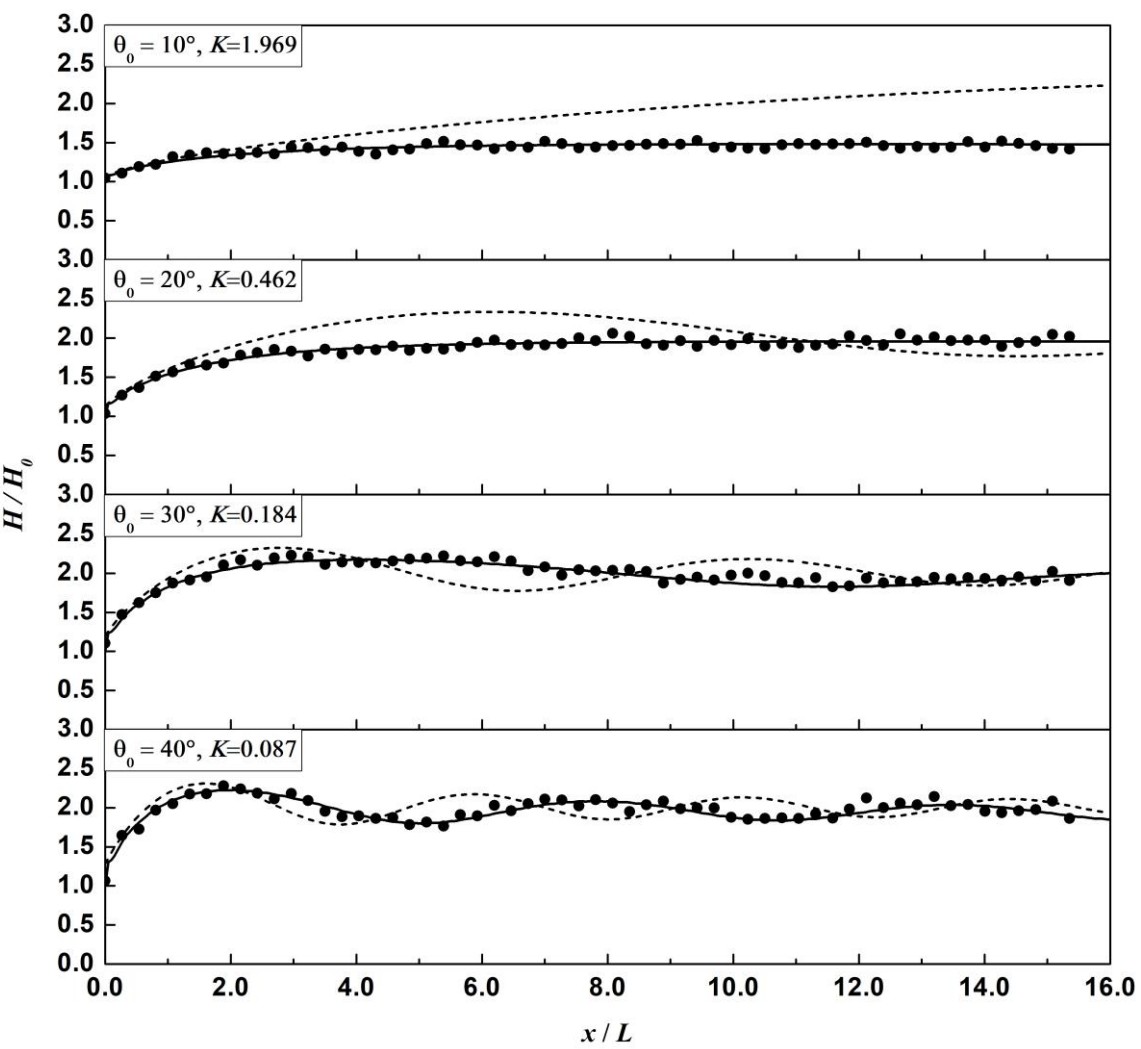

**Figure A16. Normalized wave heights along the wall for the cases of MLL1 ~ MLL4. Solid circle: measured, solid line: present numerical, dashed line: analytical (Chen, 1987).**

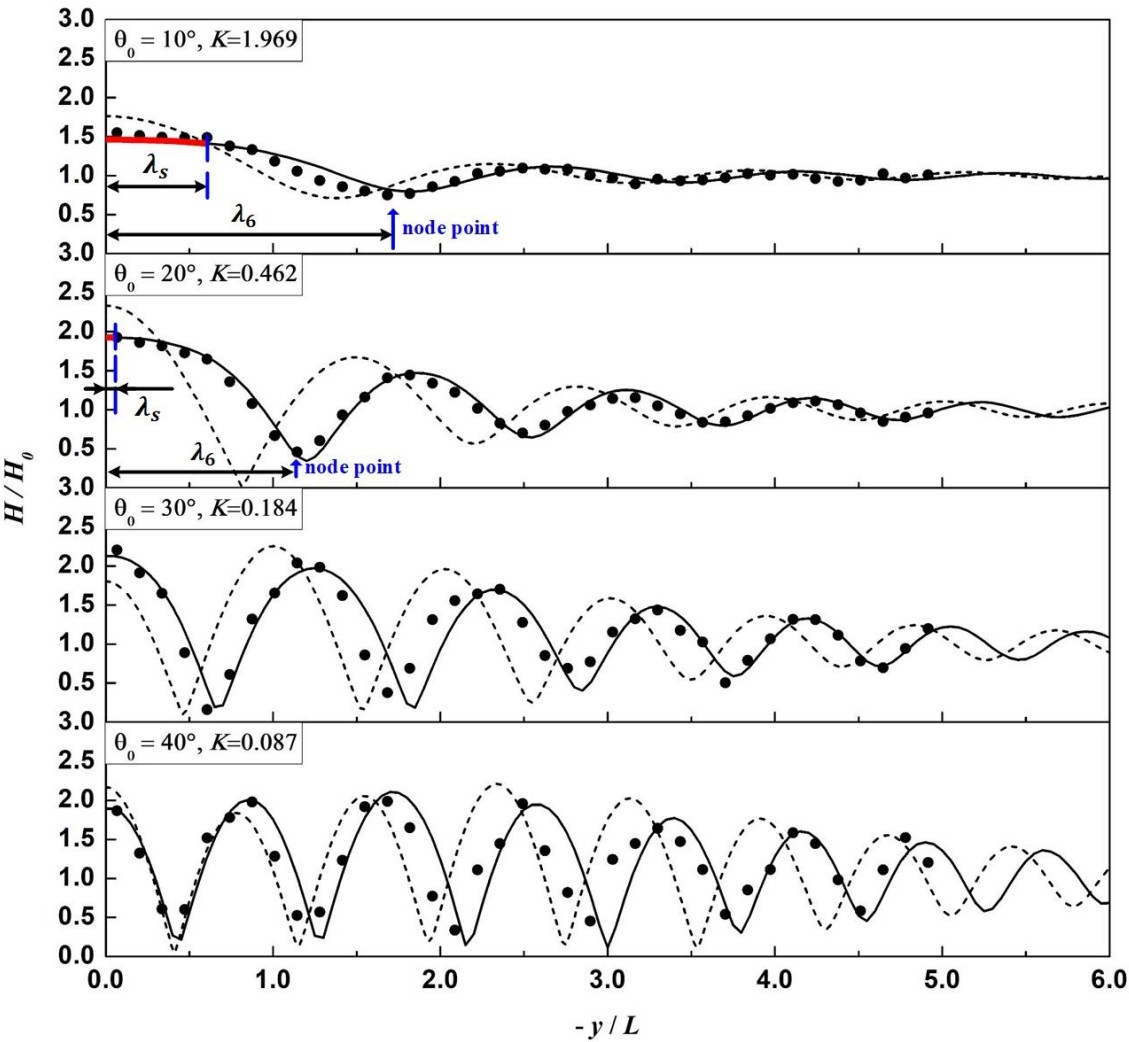

**Figure A17. Normalized wave heights normal to the wall at $x = 6L$ for the cases of MLL1 ~ MLL4. Solid symbol: measured, solid line: present numerical, dashed line: analytical (Chen, 1987). The red lines represent the stem waves. The stem width $\lambda_s$ is determined using Eq. (13).**

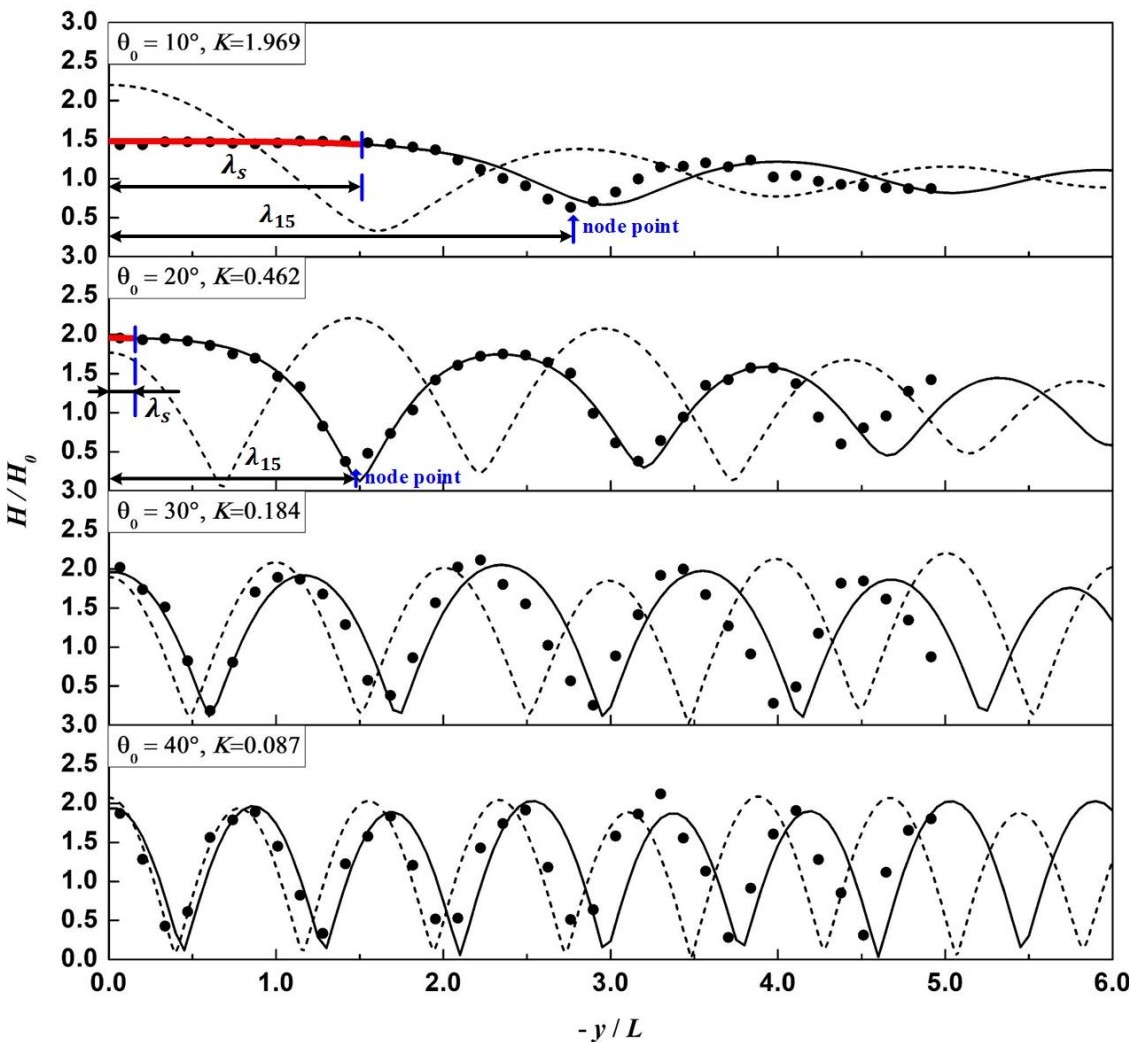

**Figure A18. Normalized wave heights normal to the wall at $x = 15L$ for the cases of MLL1 ~ MLL4. Solid circle (measured), solid line (present numerical), dashed line (analytical, Chen, 1987). The red lines represent the stem waves. The stem width $\lambda_s$ is determined using Eq. (13).**

