# Peer review of "Laboratory and numerical experiments on stem waves due to monochromatic waves along a vertical wall"

_Nonlinear Processes in Geophysics, 2017_

## Referee Comment (RC1) · Anonymous Referee #1 · 22 Aug 2017

Stem waves are an interesting topic and research on this field may help coastal engineers in properly design vertical defense structures. Experimental data are compared with numerical results and analytical solutions, with interesting findings on influence of wave nonlinearity on stem wave generation. The manuscript is fairly well written and merits to be presented to the scientific community, though some moderate additions and amendments are required.

[Figure]

**Major points**

In the results which illustrate the comparison between experiments, numerical simulations and analytical solutions, stem waves should be better highlighted. In particular, looking at the plane behavior of the waves depicted by Figs. 2 and 3, it would be interesting to present 3-dimensional results in addition to the existing 2-dimensional plots (Figs. 4 to 21). Since experimental measures were only collected along the x axis and at two specific y alignments, they do not cover the whole domain. However, numerical results from the REF/DIF model may be used to illustrate what happens in the whole domain for cases which clearly show existence of stem waves, e.g. using color maps to represent normalized wave heights $H/H_0$ in the x/L-y/L domain. Such 3d results may also be used to explain the wave reflection induced by the stem boundary. To this aim, the sentence at P11 L24-25 must be expanded.

With the purpose to properly identify stem waves in Figs. 4 to 21, these should be better highlighted, e.g. adding a further/overlapping colored line between the wall and the first nodal line. Such improvement will clarify the stem wave description (e.g., P8 L26-31).

Photo 2 suggests a "beehive" wave pattern. This is typical of the cross-sea, generated by two or more waves which interact as a consequence of, e.g., reflection, refraction. The authors are required to comment on that point referring to studies on propagation of plane waves (e.g., Le Mehauté, 1976; Mei, 1983) and cross-sea (Postacchini et al., 2014).

In the experiment description, the displacement of the measuring points should be clarified. In particular, two incident wave measuring points are illustrated in Fig.3, while three measuring points are recalled at P6 L18-19. Clarifications are needed about all used measuring/checking points (notice that five points are represented in Fig.3).

**Specific points**

- P1 L20-21: the last sentence of the abstract is awkward/unclear and should be rephrased.

- P2 L31-32: it should be "... the effects of both nonlinearity and angle of incidence. In the final section...".

- P3 L8-9: when talking of "recent version of REF/DIF", a significantly recent reference should be included (not only those of 1986 and 1994); otherwise, "latest version" is more appropriate.

- P5 L13-14: "each with dimensions of 0.5m ... in height and driven by".

- P5 L27 and P6 L2: "numeric number" should be replaced with "number" or "numeric digit".

- P6 L1-2: " 'shorter' or 'longer' waves in terms of period, respectively... or 'large' waves in terms of incident wave height, respectively".

- P7 L21: "of the incident wave is three times larger than the MSS-series waves".

- P7 L27: remove "downwave".

- P7 L31: "in good agreement"; check use of "agreement" throughout the text.

- P7 L32-33: "the measured data, probably because of nonlinear interactions between incident".

- P8 L7: "to reach a constant value".

- P8 L19-21: "The amplitude of the MLS incident waves is chosen to provide the same steepness, ... , as the MSS waves. Hence, the wave patterns observed in the MSS-series (Fig.4) are similar to the results of the MLS-series".

- P10 L24-26 – P11 L1-4: if $\beta$ is the slope ratio, $\beta\epsilon$ should be the slope of the stem boundary; if so, this must be clarified in the text.

- P11 L1 and L4: the wall angle is $\theta_0$, please amend.

- P11 L22 and Fig.23b: the term "l" must be added to Fig.23b.

- P12 L12: "The key results derived from this study are here illustrated".

- P12 L17: "agree".

- Fig.4 to 21: the y-axis label should be "$H/H_0$".

**References**

Le Méhauté, B. (1976). An introduction to hydrodynamics and water waves. Springer-Verlag.

Mei, C.C. (1983). The applied dynamics of ocean surface waves. Wiley-Interscience.

Postacchini, M., Brocchini, M., Soldini, L. (2014). Vorticity generation due to cross-sea. Journal of Fluid Mechanics, 744, 286-309.

---

## Referee Comment (RC2) · Anonymous Referee #2 · 9 Oct 2017

The authors study stem waves along a breakwater. Their motivation is stated:" Even though the existence and the properties of stem waves for sinusoidal waves are well known theoretically via numerical simulations (e.g., Yue and Mei, 1980; Yoon and Liu, 1989), they are not yet fully supported by physical experiments. Berger and Kohlhase (1976) and Mase et al. (2002) conducted hydraulic experiments to show the existence of stem waves for the cases of sinusoidal waves. Their experimental data, however, failed to produce clear stem waves, possibly due to partial reflection from the beach, diffraction from the ends of vertical wall, or insufficient space in the wave basin. Thus, there is still need to perform a precisely controlled experiment to investigate the existence and the properties of stem waves" The authors have published a similar paper

in English in 2007: 1) Lee, J & Kim, Young-Taek. (2007). Numerical Analysis of Stem Waves along a Vertical. Journal of Coastal Research, SI 50 (Proceedings of the 9th International Coastal Symposium), 1101 - 1105. Gold Coast, Australia, ISSN 0749.0208 They also published 2 papers (in Korean) in 2003 and in 2006: 2) Lee Jong-In, Kim Young-Teak, Cho Yong-Sik 2003, 2003.10, 4939-4943 (5 pages) Hydraulic Model Test for Stem Waves along Vertical Wall under Regular Wave Actions 3) Lee, Jong-In, and Sung-Bum Yoon. "Hydraulic and numerical experiments of stem waves along a vertical wall" Journal of The Korean Society of Civil Engineers 26.4B (2006): 405-412. the figures and the content of this paper seem to match very well the content of the current manuscript. They do not mention any of these 3 papers in the present manuscript. It seems that the current manuscript is lacking a proper discussion of previously published work, in English and in Korean, in particular by the authors themselves (from more than a decade ago). Such discussion is a prerequisite to evaluate the originality and merit of the present manuscript.

---

## Editor Comment (EC1) · V. Shrira (Editor) · 11 Oct 2017

Dear Dr Choong Hun Shin,

As you've already seen I've got two reports, both are not favourable and do not recommend the manuscript publication in its present form. However, the referee 1 sees a potential in your work, moreover, he provides detailed constructive suggestions on how the manuscript could be improved. Referee's 2 focus is on your earlier closely related publications you failed to mention and discuss. This is absolutely unacceptable and is against the basic principles of the journal.

Nevertheless I am inclined to give you a second chance and to allow you to submit a revised version. In the revised version you have to refer to all earlier versions of this work and to discuss the progress made since the last publication. I realize that one of these works is a conference proceedings type publication, nevertheless you have to discuss it.

Re the essence of the work, you have to make an effort and to present your observations of the Mach stem in a more convincing form. Perhaps the suggestions by Ref. 1 might help.

─────────────────────────

---

## Author Comment (AC1) · 23 Oct 2017

**Author Reply_1**

Authors: Sung Bum Yoon, Jong-In Lee, Young-Take Kim and Choong Hun Shin

Title: Laboratory and numerical experiments on stem waves due to monochromatic waves along a vertical wall

Revision date: 23 / 10/ 2017

Summary of responses:

We appreciate the interest and criticisms of the referee on our manuscript entitled "Laboratory and numerical experiments on stem waves due to monochromatic waves along a vertical wall". We hope that the revision we made could have well reflected the referee's comments.

**< Major points >**

| Comments and Suggestions | Response | Page Reference (Original) | Page Referred |
|---|---|---|---|
| In the results which illustrate the comparison between experiments, numerical simulations and analytical solutions, stem waves should be better highlighted. In particular, looking at the plane behavior of the waves depicted by Figs. 2 and 3, it would be interesting to present 3-dimensional results in addition to the existing 2-dimensional plots (Figs. 4 to 21). Since experimental measures were only collected along the x axis and at two specific y alignments, they do not cover the whole domain. However, numerical results from the REF/DIF model may be used to illustrate what happens in the whole domain for cases which clearly show existence of stem waves, e.g. using color maps to represent normalized wave heights H/H0 in the x/L-y/L domain. Such 3d results may also be used to explain the wave reflection induced by the stem boundary. To this aim, the sentence at P11 L24-25 must be expanded. | We corrected as referee suggested. "Fig. 22(a) and 22(b) show the comparison of the three-dimensional plots of normalized wave height for MLS1 and MLL1 cases, respectively, based on the numerical results of REF/DIF. For the nonlinear case, the overall amplitudes are much smaller and the stem waves are developed along the wall as shown in Fig. 22(b). The stem wave height is nearly constant and the width of the stem waves tended to increase along the wall. Fig. 23(a) and Fig. 23(b) present the comparison of the three-dimensional plots of normalized free surface displacements for MLS1 and MLL1 cases, respectively. From Fig. 23(b) it can be seen that the stem waves propagate along the wall. Fig. 24 shows the contour plots of the instantaneous free surface for MLS1 and MLL1 cases. The incident waves are reflected from the wall for the linear case. However, they are both refracted and partially reflected at the edge of stem region or the stem boundary as depicted also in Fig. 2." | | P9 L28-P10 L2 |

| | | | |
|---|---|---|---|
| With the purpose to properly identify stem waves in Figs. 4 to 21, these should be better highlighted, e.g. adding a further/overlapping colored line between the wall and the first nodal line. Such improvement will clarify the stem wave description (e.g., P8 L26-31). | We corrected as referee suggested. "The red lines shown in the figure represent the stem waves. The definition of stem width is rather controversial. Yue and Mei (1980) defined the stem width as the distance from the wall to the edge of the uniform wave amplitude region in the direction of incident wave crest lines. However, it is not an easy task to locate the edge of the flat region. On the other hand, Berger and Kohlhase (1976) defined the stem width as the distance along the stem crest lines from the wall to the first nodal line of standing wave pattern which is easier to identify from the measured data. In this study the stem edge was determined as a point which is apart from the first nodal point towards the wall by a distance λ between the first node and the second antinode (see Figs. 8 and 9). This new definition of stem width is easier to determine and is consistent with the definition of Yue and Mei (1980)." | | P8 L5-13 |
| Photo 2 suggests a "beehive" wave pattern. This is typical of the cross-sea, generated by two or more waves which interact as a consequence of, e.g., reflection, refraction. The authors are required to comment on that point referring to studies on propagation of plane waves (e.g., Le Mehauté, 1976; Mei, 1983) and cross-sea (Postacchini et al., 2014). | We corrected as referee suggested. "Photo 2 shows the hexagonal or beehive wave pattern captured during the experiment in front of a vertical wall for the case of $\theta_0 = 30°$. This is typical of the cross-sea generated by the oblique interaction of two or more traveling plane waves (see e.g., Le Mehauté, 1976; Mei, 1983; Nicholls, 2001). Postacchini et al. (2014) studied the generation and evolution of large-scale eddies of vertical axis generated by the breaking of two crossing wave trains." | P6 L15 | P6 L22-25 |
| In the experiment description, the displacement of the measuring points should be clarified. In particular, two incident wave measuring points are illustrated in Fig.3, while three measuring points are recalled at P6 L18-19. Clarifications are needed about all used measuring/checking points (notice that five points are represented in Fig.3). | We corrected as referee suggested. "Table 2 gives a summary of the wave height measurement positions." | | P6 L20-21 and Fig. 3. |

**Author Reply_1**

**< Specific points >**

| Comments and Suggestions | Response | Page Reference (Origin) | Page Referred |
|---|---|---|---|
| the last sentence of the abstract is awkward/unclear and should be rephrased. | We corrected as referee suggested. "The results of present experiments support favorably the existence and the properties of stem waves found by other researchers using numerical simulations." | P1 L20-21 | P1 L20-21 |
| it should be ". . . the effects of both nonlinearity and angle of incidence. In the final section. . .". | We corrected as referee suggested. "the effects of both nonlinearity and angle of incidence. In the final section," | P2 L31-32 | P3 L4-5 |
| when talking of "recent version of REF/DIF", a significantly recent reference should be included (not only those of 1986 and 1994); otherwise, "latest version" is more appropriate. | We corrected as referee suggested. "the latest version of REF/DIF, a wide-angle nonlinear parabolic approximation equation model developed by Kirby et al (2002)," | P3 L8-9 | P3 L13-14 |
| "each with dimensions of 0.5m . . . in height and driven by". | We corrected as referee suggested. | P5 L13-14 | P5 L19-20 |
| "numeric number" should be replaced with "number" or "numeric digit". | We corrected as referee suggested.. | P5 L27 and P6 L2 | P6 L6 and P6 L8 |
| " 'shorter' or 'longer' waves in terms of period, respectively. . . or 'large' waves in terms of incident wave height, respectively". | We corrected as referee suggested. | P6 L1-2 | P6 L6-8 |
| "of the incident wave is three times larger than the MSS-series waves". | We corrected as referee suggested. | P7 L21 | P7 L30 |
| remove "downwave". | We corrected as referee suggested. | P7 L27 | P8 L2 |
| "in good agreement"; check use of "agreement" throughout the text. | We corrected as referee suggested. | P7 L31 | P8 L14 P8 L33 P9 L12 P10 L3 P12 L20 P13 L8 |
| "the measured data, probably because of nonlinear interactions between incident". | We corrected as referee suggested. | P7 L32-33 | P8 L15-16 |
| "to reach a constant value". | We corrected as referee suggested. | P8 L7 | P8 L22 |
| "The amplitude of the MLS incident waves is chosen to provide the same steepness, … , as the MSS waves. Hence, the wave patterns observed in the MSS-series (Fig.4) are similar to the results of the MLS-series". | We corrected as referee suggested. | P8 L19-21 | P9 L1-3 |

**Author Reply_1**

| | | | |
|---|---|---|---|
| if $\beta$ is the slope ratio, $\beta_\epsilon$ should be the slope of the stem boundary; if so, this must be clarified in the text. | We corrected as referee suggested. "where $\beta\epsilon$ is the slope of the stem boundary as shown in Fig.26(a)." | P10 L24-26 P11 L1-4 | P11 L22 |
| the wall angle is $\theta_0$, please amend | We corrected as referee suggested. | P11 L1 and L4 | P11 L18 and L21 |
| the term " $l$ " must be added to Fig.23b. | We corrected as referee suggested. | P11 L22 and Fig.23b | Fig.23b. |
| "The key results derived from this study are here illustrated". | We corrected as referee suggested. | P12 L12 | P12 L27 |
| "agree". | We corrected as referee suggested. | P12 L17 | P13 L5 |
| the y-axis label should be "H/H0". | We corrected as referee suggested. | Fig.4 to 21 | Fig. 4 to Fig. 21 |

---

## Author Comment (AC2) · 23 Oct 2017

**Author Reply_2**

Authors: Sung Bum Yoon, Jong-In Lee, Young-Take Kim and Choong Hun Shin
Title: Laboratory and numerical experiments on stem waves due to monochromatic waves
    along a vertical wall
Revision date: 23 / 10/ 2017

Summary of responses:

We appreciate the referee's interest and criticisms on our manuscript entitled "Laboratory and numerical experiments on stem waves due to monochromatic waves along a vertical wall". We hope that the revision we made could have reflected the referee's comments.

The three papers mentioned by the referee show similar results to the present manuscript, but experimental conditions and numerical results are different. Their hydraulic experiments demonstrated stem waves for some cases with a relatively large incident wave. However, the stem waves were not clearly developed because of both the narrowness of wave basin and the reflected waves from the beach as shown in Figure 1. Only four cases of incident wave conditions were tested in their experiment. Thus, the experimental data were not sufficient to investigate the properties of stem waves. Moreover, the numerical results for the cases of large angle of incidence were not highly accurate because of the small-angle parabolic model employed for their numerical simulations. In addition, the previous papers did not analyze the effect of nonlinearity of incident waves on the development of stem waves.

[Figure]

(a) $\beta = 10°$ for CASE 1        (b) $\beta = 10°$ for CASE 2

Figure 1. Relative wave height along the front wall of CASE 1 and CASE 2 (Lee and Kim, 2007)

Thus, the present authors decide to conduct precisely-controlled and comprehensive hydraulic experiments to investigate the stem waves. In the present experiments the gravel beach is carefully designed to reduce the reflected waves at less than 3% for all the incident waves considered. To overcome the narrowness of the basin the water depth is reduced to 0.25 m to secure the length of vertical wall at least 40 wavelengths for the case of T = 0.7 s and 20 wavelengths for the case of T = 1.1 s. To obtain data for various wave conditions including nonlinearity and angle of incidence total of 24 cases are considered. The large-angle parabolic model is employed to get more accurate solutions for the waves with large angle of incidence. Based on the observation of the experimental data we

propose a mechanism for the generation of stem waves in a different point of view.

Corresponding to the referee's comment we added the following description in the manuscript.

| Response | Page Reference (Origin) | Page Referred |
|---|---|---|
| We add in text.
"Lee et al. (2003), Lee and Yoon (2006) and Lee and Kim (2007) performed laboratory experiments to investigate stem waves for sinusoidal waves, and compared the measured waves with the numerical results obtained using a nonlinear parabolic approximation equation model. Their hydraulic experiments demonstrated stem waves for some cases with a relatively large incident wave. However, the stem waves were not clearly developed because of both the narrowness of wave basin and the reflected waves from the beach. Only four cases of incident wave conditions were tested in their experiment. Thus, the experimental data were not sufficient to investigate the properties of stem waves. Moreover, the numerical results for the cases of large angle of incidence were not highly accurate because of the small-angle parabolic model employed for their numerical simulations." | | P2 L25-32 |

---

## Author Comment (AC3) · 23 Oct 2017

We appreciate the interest and criticisms of the editor on our manuscript entitled "Laboratory and numerical experiments on stem waves due to monochromatic waves along a vertical wall". We hope that the revision we made could have well reflected the referee's comments.

---

## Author Response (AR1)

**Author Reply\_1**

We appreciate the interest and criticisms of the referee on our manuscript entitled "Laboratory and numerical experiments on stem waves due to monochromatic waves along a vertical wall". We hope that the revision we made could have well reflected the referee's comments.

**< Major points >**

| Comments and Suggestions        | Response                                   | Page       | Page     |
|----------------------------------------|--------------------------------------------|------------|----------|
|                                        |                                            | Reference  | Referred |
|                                        |                                            | (Original) |          |
| In the results which illustrate the    | We corrected as referee suggested.         |            | P9 L28-  |
| comparison between experiments,        | "Fig. 22(a) and 22(b) show the             |            | P10 L2   |
| numerical simulations and analytical   | comparison of the three-dimensional        |            |          |
| solutions, stem waves should be        | plots of normalized wave height for        |            |          |
| better highlighted. In particular,     | MLS1 and MLL1 cases, respectively,         |            |          |
| looking at the plane behavior of the   | based on the numerical results of          |            |          |
| waves depicted by Figs. 2 and 3, it    | REF/DIF. For the nonlinear case, the       |            |          |
| would be interesting to present 3-     | overall amplitudes are much smaller and    |            |          |
| dimensional results in addition to the | the stem waves are developed along the     |            |          |
| existing 2-dimensional plots (Figs. 4  | wall as shown in Fig. 22(b). The stem      |            |          |
| to 21). Since experimental measures    | wave height is nearly constant and the     |            |          |
| were only collected along the x axis   | width of the stem waves tended to          |            |          |
| and at two specific y alignments, they | increase along the wall. Fig. 23(a) and    |            |          |
| do not cover the whole domain.         | Fig. 23(b) present the comparison of the   |            |          |
| However, numerical results from the    | three-dimensional plots of normalized      |            |          |
| REF/DIF model may be used to           | free surface displacements for MLS1 and    |            |          |
| illustrate what happens in the whole   | MLL1 cases, respectively. From Fig.        |            |          |
| domain for cases which clearly show    | 23(b) it can be seen that the stem waves   |            |          |
| existence of stem waves, e.g. using    | propagate along the wall. Fig. 24 shows    |            |          |
| color maps to represent normalized     | the contour plots of the instantaneous     |            |          |
| wave heights H/H0 in the x/L-y/L       | free surface for MLS1 and MLL1 cases.      |            |          |
| domain. Such 3d results may also be    | The incident waves are reflected from the  |            |          |
| used to explain the wave reflection    | wall for the linear case. However, they    |            |          |
| induced by the stem boundary. To       | are both refracted and partially reflected |            |          |
| this aim, the sentence at P11 L24-25   | at the edge of stem region or the stem     |            |          |
| must be expanded.                      | boundary as depicted also in Fig. 2."      |            |          |

| With the purpose to properly identify                                                                                                                                                                                                                                                                                                                                                                                                                                                                                                                                                                                                                                                                                                                    | We corrected as referee suggested.                                                                                                                                                                                                                                                                                                                                                                                                                                                                                                                                                                                                                                                                                                                                                                                                                                                                                                                                                                                                                                                                                                                                                                                                                                                                                                                                                                                                                                                                                                                                                                                                                                                                                                                                                                                                                                                 |         | P8 L5-13                                         |
|----------------------------------------------------------------------------------------------------------------------------------------------------------------------------------------------------------------------------------------------------------------------------------------------------------------------------------------------------------------------------------------------------------------------------------------------------------------------------------------------------------------------------------------------------------------------------------------------------------------------------------------------------------------------------------------------------------------------------------------------------------|------------------------------------------------------------------------------------------------------------------------------------------------------------------------------------------------------------------------------------------------------------------------------------------------------------------------------------------------------------------------------------------------------------------------------------------------------------------------------------------------------------------------------------------------------------------------------------------------------------------------------------------------------------------------------------------------------------------------------------------------------------------------------------------------------------------------------------------------------------------------------------------------------------------------------------------------------------------------------------------------------------------------------------------------------------------------------------------------------------------------------------------------------------------------------------------------------------------------------------------------------------------------------------------------------------------------------------------------------------------------------------------------------------------------------------------------------------------------------------------------------------------------------------------------------------------------------------------------------------------------------------------------------------------------------------------------------------------------------------------------------------------------------------------------------------------------------------------------------------------------------------|---------|--------------------------------------------------|
| stem waves in Figs. 4 to 21. these                                                                                                                                                                                                                                                                                                                                                                                                                                                                                                                                                                                                                                                                                                                       | "The red lines shown in the figure                                                                                                                                                                                                                                                                                                                                                                                                                                                                                                                                                                                                                                                                                                                                                                                                                                                                                                                                                                                                                                                                                                                                                                                                                                                                                                                                                                                                                                                                                                                                                                                                                                                                                                                                                                                                                                                 |         |                                                  |
| should be better highlighted, e.g.                                                                                                                                                                                                                                                                                                                                                                                                                                                                                                                                                                                                                                                                                                                       | represent the stem waves. The definition                                                                                                                                                                                                                                                                                                                                                                                                                                                                                                                                                                                                                                                                                                                                                                                                                                                                                                                                                                                                                                                                                                                                                                                                                                                                                                                                                                                                                                                                                                                                                                                                                                                                                                                                                                                                                                           |         |                                                  |
| adding a further/overlapping colored                                                                                                                                                                                                                                                                                                                                                                                                                                                                                                                                                                                                                                                                                                                     | of stem width is rather controversial. Yue                                                                                                                                                                                                                                                                                                                                                                                                                                                                                                                                                                                                                                                                                                                                                                                                                                                                                                                                                                                                                                                                                                                                                                                                                                                                                                                                                                                                                                                                                                                                                                                                                                                                                                                                                                                                                                         |         |                                                  |
| line between the wall and the first                                                                                                                                                                                                                                                                                                                                                                                                                                                                                                                                                                                                                                                                                                                      | and Mei (1980) defined the stem width                                                                                                                                                                                                                                                                                                                                                                                                                                                                                                                                                                                                                                                                                                                                                                                                                                                                                                                                                                                                                                                                                                                                                                                                                                                                                                                                                                                                                                                                                                                                                                                                                                                                                                                                                                                                                                              |         |                                                  |
| nodal line. Such improvement will                                                                                                                                                                                                                                                                                                                                                                                                                                                                                                                                                                                                                                                                                                                        | as the distance from the wall to the edge                                                                                                                                                                                                                                                                                                                                                                                                                                                                                                                                                                                                                                                                                                                                                                                                                                                                                                                                                                                                                                                                                                                                                                                                                                                                                                                                                                                                                                                                                                                                                                                                                                                                                                                                                                                                                                          |         |                                                  |
| clarify the stem wave description                                                                                                                                                                                                                                                                                                                                                                                                                                                                                                                                                                                                                                                                                                                        | of the uniform wave amplitude region in                                                                                                                                                                                                                                                                                                                                                                                                                                                                                                                                                                                                                                                                                                                                                                                                                                                                                                                                                                                                                                                                                                                                                                                                                                                                                                                                                                                                                                                                                                                                                                                                                                                                                                                                                                                                                                            |         |                                                  |
| (e.g., P8 L26-31).                                                                                                                                                                                                                                                                                                                                                                                                                                                                                                                                                                                                                                                                                                                                       | the direction of incident wave crest lines.                                                                                                                                                                                                                                                                                                                                                                                                                                                                                                                                                                                                                                                                                                                                                                                                                                                                                                                                                                                                                                                                                                                                                                                                                                                                                                                                                                                                                                                                                                                                                                                                                                                                                                                                                                                                                                        |         |                                                  |
| (0.9., 10 220 01).                                                                                                                                                                                                                                                                                                                                                                                                                                                                                                                                                                                                                                                                                                                                       | However it is not an easy task to locate                                                                                                                                                                                                                                                                                                                                                                                                                                                                                                                                                                                                                                                                                                                                                                                                                                                                                                                                                                                                                                                                                                                                                                                                                                                                                                                                                                                                                                                                                                                                                                                                                                                                                                                                                                                                                                           |         |                                                  |
|                                                                                                                                                                                                                                                                                                                                                                                                                                                                                                                                                                                                                                                                                                                                                          | the edge of the flat region On the other                                                                                                                                                                                                                                                                                                                                                                                                                                                                                                                                                                                                                                                                                                                                                                                                                                                                                                                                                                                                                                                                                                                                                                                                                                                                                                                                                                                                                                                                                                                                                                                                                                                                                                                                                                                                                                           |         |                                                  |
|                                                                                                                                                                                                                                                                                                                                                                                                                                                                                                                                                                                                                                                                                                                                                          | hand Berger and Kohlhase (1976)                                                                                                                                                                                                                                                                                                                                                                                                                                                                                                                                                                                                                                                                                                                                                                                                                                                                                                                                                                                                                                                                                                                                                                                                                                                                                                                                                                                                                                                                                                                                                                                                                                                                                                                                                                                                                                                    |         |                                                  |
|                                                                                                                                                                                                                                                                                                                                                                                                                                                                                                                                                                                                                                                                                                                                                          | defined the stem width as the distance                                                                                                                                                                                                                                                                                                                                                                                                                                                                                                                                                                                                                                                                                                                                                                                                                                                                                                                                                                                                                                                                                                                                                                                                                                                                                                                                                                                                                                                                                                                                                                                                                                                                                                                                                                                                                                             |         |                                                  |
|                                                                                                                                                                                                                                                                                                                                                                                                                                                                                                                                                                                                                                                                                                                                                          | along the stem crest lines from the wall                                                                                                                                                                                                                                                                                                                                                                                                                                                                                                                                                                                                                                                                                                                                                                                                                                                                                                                                                                                                                                                                                                                                                                                                                                                                                                                                                                                                                                                                                                                                                                                                                                                                                                                                                                                                                                           |         |                                                  |
|                                                                                                                                                                                                                                                                                                                                                                                                                                                                                                                                                                                                                                                                                                                                                          | to the first nodal line of standing wave                                                                                                                                                                                                                                                                                                                                                                                                                                                                                                                                                                                                                                                                                                                                                                                                                                                                                                                                                                                                                                                                                                                                                                                                                                                                                                                                                                                                                                                                                                                                                                                                                                                                                                                                                                                                                                           |         |                                                  |
|                                                                                                                                                                                                                                                                                                                                                                                                                                                                                                                                                                                                                                                                                                                                                          | pattern which is easier to identify from                                                                                                                                                                                                                                                                                                                                                                                                                                                                                                                                                                                                                                                                                                                                                                                                                                                                                                                                                                                                                                                                                                                                                                                                                                                                                                                                                                                                                                                                                                                                                                                                                                                                                                                                                                                                                                           |         |                                                  |
|                                                                                                                                                                                                                                                                                                                                                                                                                                                                                                                                                                                                                                                                                                                                                          | the measured data. In this study the stem                                                                                                                                                                                                                                                                                                                                                                                                                                                                                                                                                                                                                                                                                                                                                                                                                                                                                                                                                                                                                                                                                                                                                                                                                                                                                                                                                                                                                                                                                                                                                                                                                                                                                                                                                                                                                                          |         |                                                  |
|                                                                                                                                                                                                                                                                                                                                                                                                                                                                                                                                                                                                                                                                                                                                                          | edge was determined as a point which is                                                                                                                                                                                                                                                                                                                                                                                                                                                                                                                                                                                                                                                                                                                                                                                                                                                                                                                                                                                                                                                                                                                                                                                                                                                                                                                                                                                                                                                                                                                                                                                                                                                                                                                                                                                                                                            |         |                                                  |
|                                                                                                                                                                                                                                                                                                                                                                                                                                                                                                                                                                                                                                                                                                                                                          | apart from the first nodal point towards                                                                                                                                                                                                                                                                                                                                                                                                                                                                                                                                                                                                                                                                                                                                                                                                                                                                                                                                                                                                                                                                                                                                                                                                                                                                                                                                                                                                                                                                                                                                                                                                                                                                                                                                                                                                                                           |         |                                                  |
|                                                                                                                                                                                                                                                                                                                                                                                                                                                                                                                                                                                                                                                                                                                                                          | the well by a distance ) between the                                                                                                                                                                                                                                                                                                                                                                                                                                                                                                                                                                                                                                                                                                                                                                                                                                                                                                                                                                                                                                                                                                                                                                                                                                                                                                                                                                                                                                                                                                                                                                                                                                                                                                                                                                                                                                               |         |                                                  |
|                                                                                                                                                                                                                                                                                                                                                                                                                                                                                                                                                                                                                                                                                                                                                          | first node and the second entinede (see                                                                                                                                                                                                                                                                                                                                                                                                                                                                                                                                                                                                                                                                                                                                                                                                                                                                                                                                                                                                                                                                                                                                                                                                                                                                                                                                                                                                                                                                                                                                                                                                                                                                                                                                                                                                                                            |         |                                                  |
|                                                                                                                                                                                                                                                                                                                                                                                                                                                                                                                                                                                                                                                                                                                                                          | Figs 8 and 0) This new definition of                                                                                                                                                                                                                                                                                                                                                                                                                                                                                                                                                                                                                                                                                                                                                                                                                                                                                                                                                                                                                                                                                                                                                                                                                                                                                                                                                                                                                                                                                                                                                                                                                                                                                                                                                                                                                                               |         |                                                  |
|                                                                                                                                                                                                                                                                                                                                                                                                                                                                                                                                                                                                                                                                                                                                                          | rigs. 8 and 9). This new definition of                                                                                                                                                                                                                                                                                                                                                                                                                                                                                                                                                                                                                                                                                                                                                                                                                                                                                                                                                                                                                                                                                                                                                                                                                                                                                                                                                                                                                                                                                                                                                                                                                                                                                                                                                                                                                                             |         |                                                  |
|                                                                                                                                                                                                                                                                                                                                                                                                                                                                                                                                                                                                                                                                                                                                                          | stelli width is easier to determine and is                                                                                                                                                                                                                                                                                                                                                                                                                                                                                                                                                                                                                                                                                                                                                                                                                                                                                                                                                                                                                                                                                                                                                                                                                                                                                                                                                                                                                                                                                                                                                                                                                                                                                                                                                                                                                                         |         |                                                  |
|                                                                                                                                                                                                                                                                                                                                                                                                                                                                                                                                                                                                                                                                                                                                                          | Mai (1080) "                                                                                                                                                                                                                                                                                                                                                                                                                                                                                                                                                                                                                                                                                                                                                                                                                                                                                                                                                                                                                                                                                                                                                                                                                                                                                                                                                                                                                                                                                                                                                                                                                                                                                                                                                                                                                                                                       |         |                                                  |
|                                                                                                                                                                                                                                                                                                                                                                                                                                                                                                                                                                                                                                                                                                                                                          | Mei (1980).                                                                                                                                                                                                                                                                                                                                                                                                                                                                                                                                                                                                                                                                                                                                                                                                                                                                                                                                                                                                                                                                                                                                                                                                                                                                                                                                                                                                                                                                                                                                                                                                                                                                                                                                                                                                                                                                        |         |                                                  |
| Di                                                                                                                                                                                                                                                                                                                                                                                                                                                                                                                                                                                                                                                                                                                                                       | We consider the target of the second se | DC I 15 | DC 1.00                                          |
| Photo 2 suggests a "beehive" wave                                                                                                                                                                                                                                                                                                                                                                                                                                                                                                                                                                                                                                                                                                                        | We corrected as referee suggested.                                                                                                                                                                                                                                                                                                                                                                                                                                                                                                                                                                                                                                                                                                                                                                                                                                                                                                                                                                                                                                                                                                                                                                                                                                                                                                                                                                                                                                                                                                                                                                                                                                                                                                                                                                                                                                                 | P6 L15  | P6 L22-                                          |
| Photo 2 suggests a "beehive" wave
pattern. This is typical of the cross-                                                                                                                                                                                                                                                                                                                                                                                                                                                                                                                                                                                                                                                                              | We corrected as referee suggested.
"Photo 2 shows the hexagonal or beehive                                                                                                                                                                                                                                                                                                                                                                                                                                                                                                                                                                                                                                                                                                                                                                                                                                                                                                                                                                                                                                                                                                                                                                                                                                                                                                                                                                                                                                                                                                                                                                                                                                                                                                                                                                                                      | P6 L15  | P6 L22-
25                                    |
| Photo 2 suggests a "beehive" wave
pattern. This is typical of the cross-
sea, generated by two or more waves                                                                                                                                                                                                                                                                                                                                                                                                                                                                                                                                                                                                                                       | We corrected as referee suggested.
"Photo 2 shows the hexagonal or beehive
wave pattern captured during the                                                                                                                                                                                                                                                                                                                                                                                                                                                                                                                                                                                                                                                                                                                                                                                                                                                                                                                                                                                                                                                                                                                                                                                                                                                                                                                                                                                                                                                                                                                                                                                                                                                                                                                                                                  | P6 L15  | P6 L22-
25                                    |
| Photo 2 suggests a "beehive" wave
pattern. This is typical of the cross-
sea, generated by two or more waves
which interact as a consequence of,                                                                                                                                                                                                                                                                                                                                                                                                                                                                                                                                                                                                | We corrected as referee suggested.
"Photo 2 shows the hexagonal or beehive
wave pattern captured during the
experiment in front of a vertical wall for                                                                                                                                                                                                                                                                                                                                                                                                                                                                                                                                                                                                                                                                                                                                                                                                                                                                                                                                                                                                                                                                                                                                                                                                                                                                                                                                                                                                                                                                                                                                                                                                                                                                                                                    | P6 L15  | P6 L22-
25                                    |
| Photo 2 suggests a "beehive" wave
pattern. This is typical of the cross-
sea, generated by two or more waves
which interact as a consequence of,
e.g., reflection, refraction. The                                                                                                                                                                                                                                                                                                                                                                                                                                                                                                                                                           | We corrected as referee suggested.
"Photo 2 shows the hexagonal or beehive
wave pattern captured during the
experiment in front of a vertical wall for
the case of $\theta_0 = 30^\circ$ . This is typical of                                                                                                                                                                                                                                                                                                                                                                                                                                                                                                                                                                                                                                                                                                                                                                                                                                                                                                                                                                                                                                                                                                                                                                                                                                                                                                                                                                                                                                                                                                                                                                                                                                                          | P6 L15  | P6 L22-
25                                    |
| Photo 2 suggests a "beehive" wave
pattern. This is typical of the cross-
sea, generated by two or more waves
which interact as a consequence of,
e.g., reflection, refraction. The
authors are required to comment on                                                                                                                                                                                                                                                                                                                                                                                                                                                                                                                     | We corrected as referee suggested.
"Photo 2 shows the hexagonal or beehive
wave pattern captured during the
experiment in front of a vertical wall for
the case of $\theta_0 = 30^\circ$ . This is typical of
the cross-sea generated by the oblique                                                                                                                                                                                                                                                                                                                                                                                                                                                                                                                                                                                                                                                                                                                                                                                                                                                                                                                                                                                                                                                                                                                                                                                                                                                                                                                                                                                                                                                                                                                                                                                                                | P6 L15  | P6 L22-
25                                    |
| Photo 2 suggests a "beehive" wave
pattern. This is typical of the cross-
sea, generated by two or more waves
which interact as a consequence of,
e.g., reflection, refraction. The
authors are required to comment on
that point referring to studies on                                                                                                                                                                                                                                                                                                                                                                                                                                                                               | We corrected as referee suggested.
"Photo 2 shows the hexagonal or beehive
wave pattern captured during the
experiment in front of a vertical wall for
the case of $\theta_0 = 30^\circ$ . This is typical of
the cross-sea generated by the oblique
interaction of two or more traveling                                                                                                                                                                                                                                                                                                                                                                                                                                                                                                                                                                                                                                                                                                                                                                                                                                                                                                                                                                                                                                                                                                                                                                                                                                                                                                                                                                                                                                                                                                                                                                        | P6 L15  | P6 L22-
25                                    |
| Photo 2 suggests a "beehive" wave
pattern. This is typical of the cross-
sea, generated by two or more waves
which interact as a consequence of,
e.g., reflection, refraction. The
authors are required to comment on
that point referring to studies on
propagation of plane waves (e.g., Le                                                                                                                                                                                                                                                                                                                                                                                                                                       | We corrected as referee suggested.
"Photo 2 shows the hexagonal or beehive
wave pattern captured during the
experiment in front of a vertical wall for
the case of $\theta_0 = 30^\circ$ . This is typical of
the cross-sea generated by the oblique
interaction of two or more traveling
plane waves (see e.g., Le Mehauté, 1976;                                                                                                                                                                                                                                                                                                                                                                                                                                                                                                                                                                                                                                                                                                                                                                                                                                                                                                                                                                                                                                                                                                                                                                                                                                                                                                                                                                                                                                                                                                                            | P6 L15  | P6 L22-
25                                    |
| Photo 2 suggests a "beehive" wave
pattern. This is typical of the cross-
sea, generated by two or more waves
which interact as a consequence of,
e.g., reflection, refraction. The
authors are required to comment on
that point referring to studies on
propagation of plane waves (e.g., Le
Mehauté, 1976; Mei, 1983) and                                                                                                                                                                                                                                                                                                                                                                                                      | We corrected as referee suggested.
"Photo 2 shows the hexagonal or beehive
wave pattern captured during the
experiment in front of a vertical wall for
the case of $\theta_0 = 30^\circ$ . This is typical of
the cross-sea generated by the oblique
interaction of two or more traveling
plane waves (see e.g., Le Mehauté, 1976;
Mei, 1983; Nicholls, 2001). Postacchini                                                                                                                                                                                                                                                                                                                                                                                                                                                                                                                                                                                                                                                                                                                                                                                                                                                                                                                                                                                                                                                                                                                                                                                                                                                                                                                                                                                                                                                                                 | P6 L15  | P6 L22-
25                                    |
| Photo 2 suggests a "beehive" wave
pattern. This is typical of the cross-
sea, generated by two or more waves
which interact as a consequence of,
e.g., reflection, refraction. The
authors are required to comment on
that point referring to studies on
propagation of plane waves (e.g., Le
Mehauté, 1976; Mei, 1983) and
cross-sea (Postacchini et al., 2014).                                                                                                                                                                                                                                                                                                                                                             | We corrected as referee suggested.
"Photo 2 shows the hexagonal or beehive
wave pattern captured during the
experiment in front of a vertical wall for
the case of $\theta_0 = 30^\circ$ . This is typical of
the cross-sea generated by the oblique
interaction of two or more traveling
plane waves (see e.g., Le Mehauté, 1976;
Mei, 1983; Nicholls, 2001). Postacchini
et al. (2014) studied the generation and                                                                                                                                                                                                                                                                                                                                                                                                                                                                                                                                                                                                                                                                                                                                                                                                                                                                                                                                                                                                                                                                                                                                                                                                                                                                                                                                                                                                                                     | P6 L15  | P6 L22-
25                                    |
| Photo 2 suggests a "beehive" wave
pattern. This is typical of the cross-
sea, generated by two or more waves
which interact as a consequence of,
e.g., reflection, refraction. The
authors are required to comment on
that point referring to studies on
propagation of plane waves (e.g., Le
Mehauté, 1976; Mei, 1983) and
cross-sea (Postacchini et al., 2014).                                                                                                                                                                                                                                                                                                                                                             | We corrected as referee suggested.
"Photo 2 shows the hexagonal or beehive
wave pattern captured during the
experiment in front of a vertical wall for
the case of $\theta_0 = 30^\circ$ . This is typical of
the cross-sea generated by the oblique
interaction of two or more traveling
plane waves (see e.g., Le Mehauté, 1976;
Mei, 1983; Nicholls, 2001). Postacchini
et al. (2014) studied the generation and
evolution of large-scale eddies of vertical                                                                                                                                                                                                                                                                                                                                                                                                                                                                                                                                                                                                                                                                                                                                                                                                                                                                                                                                                                                                                                                                                                                                                                                                                                                                                                                                                                                      | P6 L15  | P6 L22-
25                                    |
| Photo 2 suggests a "beehive" wave
pattern. This is typical of the cross-
sea, generated by two or more waves
which interact as a consequence of,
e.g., reflection, refraction. The
authors are required to comment on
that point referring to studies on
propagation of plane waves (e.g., Le
Mehauté, 1976; Mei, 1983) and
cross-sea (Postacchini et al., 2014).                                                                                                                                                                                                                                                                                                                                                             | We corrected as referee suggested.
"Photo 2 shows the hexagonal or beehive
wave pattern captured during the
experiment in front of a vertical wall for
the case of $\theta_0 = 30^\circ$ . This is typical of
the cross-sea generated by the oblique
interaction of two or more traveling
plane waves (see e.g., Le Mehauté, 1976;
Mei, 1983; Nicholls, 2001). Postacchini
et al. (2014) studied the generation and
evolution of large-scale eddies of vertical
axis generated by the breaking of two                                                                                                                                                                                                                                                                                                                                                                                                                                                                                                                                                                                                                                                                                                                                                                                                                                                                                                                                                                                                                                                                                                                                                                                                                                                                                                                                             | P6 L15  | P6 L22-
25                                    |
| Photo 2 suggests a "beehive" wave
pattern. This is typical of the cross-
sea, generated by two or more waves
which interact as a consequence of,
e.g., reflection, refraction. The
authors are required to comment on
that point referring to studies on
propagation of plane waves (e.g., Le
Mehauté, 1976; Mei, 1983) and
cross-sea (Postacchini et al., 2014).                                                                                                                                                                                                                                                                                                                                                             | We corrected as referee suggested.
"Photo 2 shows the hexagonal or beehive
wave pattern captured during the
experiment in front of a vertical wall for
the case of $\theta_0 = 30^\circ$ . This is typical of
the cross-sea generated by the oblique
interaction of two or more traveling
plane waves (see e.g., Le Mehauté, 1976;
Mei, 1983; Nicholls, 2001). Postacchini
et al. (2014) studied the generation and
evolution of large-scale eddies of vertical
axis generated by the breaking of two
crossing wave trains."                                                                                                                                                                                                                                                                                                                                                                                                                                                                                                                                                                                                                                                                                                                                                                                                                                                                                                                                                                                                                                                                                                                                                                                                                                                                                                                   | P6 L15  | P6 L22-
25                                    |
| Photo 2 suggests a "beehive" wave
pattern. This is typical of the cross-
sea, generated by two or more waves
which interact as a consequence of,
e.g., reflection, refraction. The
authors are required to comment on
that point referring to studies on
propagation of plane waves (e.g., Le
Mehauté, 1976; Mei, 1983) and
cross-sea (Postacchini et al., 2014).                                                                                                                                                                                                                                                                                                                                                             | We corrected as referee suggested.
"Photo 2 shows the hexagonal or beehive
wave pattern captured during the
experiment in front of a vertical wall for
the case of $\theta_0 = 30^\circ$ . This is typical of
the cross-sea generated by the oblique
interaction of two or more traveling
plane waves (see e.g., Le Mehauté, 1976;
Mei, 1983; Nicholls, 2001). Postacchini
et al. (2014) studied the generation and
evolution of large-scale eddies of vertical
axis generated by the breaking of two
crossing wave trains."
We corrected as referee suggested.                                                                                                                                                                                                                                                                                                                                                                                                                                                                                                                                                                                                                                                                                                                                                                                                                                                                                                                                                                                                                                                                                                                                                                                                                                                                             | P6 L15  | P6 L22-
25
P6 L20-                         |
| Photo 2 suggests a "beehive" wave
pattern. This is typical of the cross-
sea, generated by two or more waves
which interact as a consequence of,
e.g., reflection, refraction. The
authors are required to comment on
that point referring to studies on
propagation of plane waves (e.g., Le
Mehauté, 1976; Mei, 1983) and
cross-sea (Postacchini et al., 2014).
In the experiment description, the
displacement of the measuring points                                                                                                                                                                                                                                                                               | We corrected as referee suggested.
"Photo 2 shows the hexagonal or beehive
wave pattern captured during the
experiment in front of a vertical wall for
the case of $\theta_0 = 30^\circ$ . This is typical of
the cross-sea generated by the oblique
interaction of two or more traveling
plane waves (see e.g., Le Mehauté, 1976;
Mei, 1983; Nicholls, 2001). Postacchini
et al. (2014) studied the generation and
evolution of large-scale eddies of vertical
axis generated by the breaking of two
crossing wave trains."
We corrected as referee suggested.
"Table 2 gives a summary of the wave                                                                                                                                                                                                                                                                                                                                                                                                                                                                                                                                                                                                                                                                                                                                                                                                                                                                                                                                                                                                                                                                                                                                                                                                                                     | P6 L15  | P6 L22-
25
P6 L20-
21                   |
| Photo 2 suggests a "beehive" wave
pattern. This is typical of the cross-
sea, generated by two or more waves
which interact as a consequence of,
e.g., reflection, refraction. The
authors are required to comment on
that point referring to studies on
propagation of plane waves (e.g., Le
Mehauté, 1976; Mei, 1983) and
cross-sea (Postacchini et al., 2014).
In the experiment description, the
displacement of the measuring points
should be clarified. In particular, two                                                                                                                                                                                                                                    | We corrected as referee suggested.
"Photo 2 shows the hexagonal or beehive
wave pattern captured during the
experiment in front of a vertical wall for
the case of $\theta_0 = 30^\circ$ . This is typical of
the cross-sea generated by the oblique
interaction of two or more traveling
plane waves (see e.g., Le Mehauté, 1976;
Mei, 1983; Nicholls, 2001). Postacchini
et al. (2014) studied the generation and
evolution of large-scale eddies of vertical
axis generated by the breaking of two
crossing wave trains."
We corrected as referee suggested.
"Table 2 gives a summary of the wave
height measurement positions."                                                                                                                                                                                                                                                                                                                                                                                                                                                                                                                                                                                                                                                                                                                                                                                                                                                                                                                                                                                                                                                                                                                                                                                                   | P6 L15  | P6 L22-
25 P6 L20-
21 and                  |
| Photo 2 suggests a "beehive" wave
pattern. This is typical of the cross-
sea, generated by two or more waves
which interact as a consequence of,
e.g., reflection, refraction. The
authors are required to comment on
that point referring to studies on
propagation of plane waves (e.g., Le
Mehauté, 1976; Mei, 1983) and
cross-sea (Postacchini et al., 2014).
In the experiment description, the
displacement of the measuring points
should be clarified. In particular, two
incident wave measuring points are                                                                                                                                                                                              | We corrected as referee suggested.
"Photo 2 shows the hexagonal or beehive
wave pattern captured during the
experiment in front of a vertical wall for
the case of $\theta_0 = 30^\circ$ . This is typical of
the cross-sea generated by the oblique
interaction of two or more traveling
plane waves (see e.g., Le Mehauté, 1976;
Mei, 1983; Nicholls, 2001). Postacchini
et al. (2014) studied the generation and
evolution of large-scale eddies of vertical
axis generated by the breaking of two
crossing wave trains."
We corrected as referee suggested.
"Table 2 gives a summary of the wave
height measurement positions."                                                                                                                                                                                                                                                                                                                                                                                                                                                                                                                                                                                                                                                                                                                                                                                                                                                                                                                                                                                                                                                                                                                                                                                                   | P6 L15  | P6 L22-
25
P6 L20-
21
and
Fig. 3. |
| Photo 2 suggests a "beehive" wave
pattern. This is typical of the cross-
sea, generated by two or more waves
which interact as a consequence of,
e.g., reflection, refraction. The
authors are required to comment on
that point referring to studies on
propagation of plane waves (e.g., Le
Mehauté, 1976; Mei, 1983) and
cross-sea (Postacchini et al., 2014).
In the experiment description, the
displacement of the measuring points
should be clarified. In particular, two
incident wave measuring points are
illustrated in Fig.3, while three                                                                                                                                                         | We corrected as referee suggested.
"Photo 2 shows the hexagonal or beehive
wave pattern captured during the
experiment in front of a vertical wall for
the case of $\theta_0 = 30^\circ$ . This is typical of
the cross-sea generated by the oblique
interaction of two or more traveling
plane waves (see e.g., Le Mehauté, 1976;
Mei, 1983; Nicholls, 2001). Postacchini
et al. (2014) studied the generation and
evolution of large-scale eddies of vertical
axis generated by the breaking of two
crossing wave trains."
We corrected as referee suggested.
"Table 2 gives a summary of the wave
height measurement positions."                                                                                                                                                                                                                                                                                                                                                                                                                                                                                                                                                                                                                                                                                                                                                                                                                                                                                                                                                                                                                                                                                                                                                                                                   | P6 L15  | P6 L22-
25
P6 L20-
21
and
Fig. 3. |
| Photo 2 suggests a "beehive" wave
pattern. This is typical of the cross-
sea, generated by two or more waves
which interact as a consequence of,
e.g., reflection, refraction. The
authors are required to comment on
that point referring to studies on
propagation of plane waves (e.g., Le
Mehauté, 1976; Mei, 1983) and
cross-sea (Postacchini et al., 2014).
In the experiment description, the
displacement of the measuring points
should be clarified. In particular, two
incident wave measuring points are
illustrated in Fig.3, while three
measuring points are recalled at P6                                                                                                                  | We corrected as referee suggested.
"Photo 2 shows the hexagonal or beehive
wave pattern captured during the
experiment in front of a vertical wall for
the case of $\theta_0 = 30^\circ$ . This is typical of
the cross-sea generated by the oblique
interaction of two or more traveling
plane waves (see e.g., Le Mehauté, 1976;
Mei, 1983; Nicholls, 2001). Postacchini
et al. (2014) studied the generation and
evolution of large-scale eddies of vertical
axis generated by the breaking of two
crossing wave trains."
We corrected as referee suggested.
"Table 2 gives a summary of the wave
height measurement positions."                                                                                                                                                                                                                                                                                                                                                                                                                                                                                                                                                                                                                                                                                                                                                                                                                                                                                                                                                                                                                                                                                                                                                                                                   | P6 L15  | P6 L22-
25
P6 L20-
21
and
Fig. 3. |
| Photo 2 suggests a "beehive" wave
pattern. This is typical of the cross-
sea, generated by two or more waves
which interact as a consequence of,
e.g., reflection, refraction. The
authors are required to comment on
that point referring to studies on
propagation of plane waves (e.g., Le
Mehauté, 1976; Mei, 1983) and
cross-sea (Postacchini et al., 2014).
In the experiment description, the
displacement of the measuring points
should be clarified. In particular, two
incident wave measuring points are
illustrated in Fig.3, while three
measuring points are recalled at P6
L18-19. Clarifications are needed                                                                             | We corrected as referee suggested.
"Photo 2 shows the hexagonal or beehive
wave pattern captured during the
experiment in front of a vertical wall for
the case of $\theta_0 = 30^\circ$ . This is typical of
the cross-sea generated by the oblique
interaction of two or more traveling
plane waves (see e.g., Le Mehauté, 1976;
Mei, 1983; Nicholls, 2001). Postacchini
et al. (2014) studied the generation and
evolution of large-scale eddies of vertical
axis generated by the breaking of two
crossing wave trains."
We corrected as referee suggested.
"Table 2 gives a summary of the wave
height measurement positions."                                                                                                                                                                                                                                                                                                                                                                                                                                                                                                                                                                                                                                                                                                                                                                                                                                                                                                                                                                                                                                                                                                                                                                                                   | P6 L15  | P6 L22-
25
P6 L20-
21
and
Fig. 3. |
| Photo 2 suggests a "beehive" wave
pattern. This is typical of the cross-
sea, generated by two or more waves
which interact as a consequence of,
e.g., reflection, refraction. The
authors are required to comment on
that point referring to studies on
propagation of plane waves (e.g., Le
Mehauté, 1976; Mei, 1983) and
cross-sea (Postacchini et al., 2014).
In the experiment description, the
displacement of the measuring points
should be clarified. In particular, two
incident wave measuring points are
illustrated in Fig.3, while three
measuring points are needed
about all used measuring/checking                                                                                     | We corrected as referee suggested.
"Photo 2 shows the hexagonal or beehive
wave pattern captured during the
experiment in front of a vertical wall for
the case of $\theta_0 = 30^\circ$ . This is typical of
the cross-sea generated by the oblique
interaction of two or more traveling
plane waves (see e.g., Le Mehauté, 1976;
Mei, 1983; Nicholls, 2001). Postacchini
et al. (2014) studied the generation and
evolution of large-scale eddies of vertical
axis generated by the breaking of two
crossing wave trains."
We corrected as referee suggested.
"Table 2 gives a summary of the wave
height measurement positions."                                                                                                                                                                                                                                                                                                                                                                                                                                                                                                                                                                                                                                                                                                                                                                                                                                                                                                                                                                                                                                                                                                                                                                                                   | P6 L15  | P6 L22-
25
P6 L20-
21
and
Fig. 3. |
| Photo 2 suggests a "beehive" wave
pattern. This is typical of the cross-
sea, generated by two or more waves
which interact as a consequence of,
e.g., reflection, refraction. The
authors are required to comment on
that point referring to studies on
propagation of plane waves (e.g., Le
Mehauté, 1976; Mei, 1983) and
cross-sea (Postacchini et al., 2014).
In the experiment description, the
displacement of the measuring points
should be clarified. In particular, two
incident wave measuring points are
illustrated in Fig.3, while three
measuring points are recalled at P6
L18-19. Clarifications are needed
about all used measuring/checking
points (notice that five points are | We corrected as referee suggested.
"Photo 2 shows the hexagonal or beehive
wave pattern captured during the
experiment in front of a vertical wall for
the case of $\theta_0 = 30^\circ$ . This is typical of
the cross-sea generated by the oblique
interaction of two or more traveling
plane waves (see e.g., Le Mehauté, 1976;
Mei, 1983; Nicholls, 2001). Postacchini
et al. (2014) studied the generation and
evolution of large-scale eddies of vertical
axis generated by the breaking of two
crossing wave trains."
We corrected as referee suggested.
"Table 2 gives a summary of the wave
height measurement positions."                                                                                                                                                                                                                                                                                                                                                                                                                                                                                                                                                                                                                                                                                                                                                                                                                                                                                                                                                                                                                                                                                                                                                                                                   | P6 L15  | P6 L22-
25
P6 L20-
21
and
Fig. 3. |

**< Specific points >**

| omments and Suggestions Response       |                                         | Page       | Page      |
|----------------------------------------|-----------------------------------------|------------|-----------|
|                                        | -                                       | Reference  | Referred  |
|                                        |                                         | (Origin)   |           |
| the last sentence of the abstract is   | We corrected as referee suggested.      | P1 L20-21  | P1 L20-21 |
| awkward/unclear and should be          | "The results of present experiments     |            |           |
| rephrased.                             | support favorably the existence and the |            |           |
| 1                                      | properties of stem waves found by       |            |           |
|                                        | other researchers using numerical       |            |           |
|                                        | simulations."                           |            |           |
| it should be " the effects of both     | We corrected as referee suggested.      | P2 L31-32  | P3 L4-5   |
| nonlinearity and angle of incidence.   | "the effects of both nonlinearity and   |            |           |
| In the final section".                 | angle of incidence. In the final        |            |           |
|                                        | section,"                               |            |           |
| when talking of "recent version of     | We corrected as referee suggested.      | P3 L8-9    | P3 L13-14 |
| REF/DIF", a significantly recent       | "the latest version of REF/DIF, a wide- |            |           |
| reference should be included (not      | angle nonlinear parabolic               |            |           |
| only those of 1986 and 1994);          | approximation equation model            |            |           |
| otherwise, "latest version" is more    | developed by Kirby et al (2002),"       |            |           |
| appropriate.                           |                                         |            |           |
| "each with dimensions of 0.5m in       | We corrected as referee suggested.      | P5 L13-14  | P5 L19-20 |
| height and driven by".                 |                                         |            |           |
| "numeric number" should be             | We corrected as referee suggested       | P5 L27     | P6 L6     |
| replaced with "number" or "numeric     |                                         | and        | and       |
| digit".                                |                                         | P6 L2      | P6 L8     |
| " 'shorter' or 'longer' waves in terms | We corrected as referee suggested.      | P6 L1-2    | P6 L6-8   |
| of period, respectively or 'large'     |                                         |            |           |
| waves in terms of incident wave        |                                         |            |           |
| height, respectively".                 |                                         |            |           |
| "of the incident wave is three times   | We corrected as referee suggested.      | P7 L21     | P7 L30    |
| larger than the MSS-series waves".     |                                         |            | DOLO      |
| remove "downwave".                     | We corrected as referee suggested.      | P7 L27     | P8 L2     |
| "in good agreement"; check use of      | We corrected as referee suggested.      | P7 L31     | P8 L14    |
| agreement' throughout the text.        |                                         |            | P8 L33    |
|                                        |                                         |            | P9 L12    |
|                                        |                                         |            | P10 L3    |
|                                        |                                         |            | P12 L20   |
| "the manurad data probably because     | We corrected as referee suggested       | D7 L 22 22 | PI3L8     |
| of nonlinear interactions between      | we confected as referee suggested.      | F / L32-33 | F8 L15-10 |
| incident"                              |                                         |            |           |
| "to reach a constant value"            | We corrected as referee suggested       | P817       | P81.22    |
| "The amplitude of the MLS incident     | We corrected as referee suggested.      | P8 L19-21  | P9 L1-3   |
| waves is chosen to provide the same    |                                         |            | - / 21 0  |
| steepness,, as the MSS waves           |                                         |            |           |
| Hence, the wave patterns observed in   |                                         |            |           |
| the MSS-series (Fig.4) are similar to  |                                         |            |           |
| the results of the MLS-series".        |                                         |            |           |

| if $\beta$ is the slope ratio, $\beta_{\epsilon}$ should be | We corrected as referee suggested.               | P10 L24-26       | P11 L22   |
|-------------------------------------------------------------|--------------------------------------------------|------------------|-----------|
| the slope of the stem boundary; if so,                      | "where $\beta \epsilon$ is the slope of the stem | P11 L1-4         |           |
| this must be clarified in the text.                         | boundary as shown in Fig.26(a)."                 |                  |           |
| the wall angle is $\theta_0$ , please amend                 | We corrected as referee suggested.               | P11 L1 and P11 I |           |
|                                                             |                                                  | L4               | and L21   |
| the term "l" must be added to                               | We corrected as referee suggested.               | P11 L22 and      | Fig.23b.  |
| Fig.23b.                                                    |                                                  | Fig.23b          |           |
| "The key results derived from this                          | We corrected as referee suggested.               | P12 L12          | P12 L27   |
| study are here illustrated".                                |                                                  |                  |           |
| "agree".                                                    | We corrected as referee suggested.               | P12 L17          | P13 L5    |
| the y-axis label should be "H/H0".                          | We corrected as referee suggested.               | Fig.4 to 21      | Fig. 4 to |
|                                                             |                                                  |                  | Fig. 21   |

**Author Reply\_2**

We appreciate the referee's interest and criticisms on our manuscript entitled "Laboratory and numerical experiments on stem waves due to monochromatic waves along a vertical wall". We hope that the revision we made could have reflected the referee's comments.

The three papers mentioned by the referee show similar results to the present manuscript, but experimental conditions and numerical results are different. Their hydraulic experiments demonstrated stem waves for some cases with a relatively large incident wave. However, the stem waves were not clearly developed because of both the narrowness of wave basin and the reflected waves from the beach as shown in Figure 1. Only four cases of incident wave conditions were tested in their experiment. Thus, the experimental data were not sufficient to investigate the properties of stem waves. Moreover, the numerical results for the cases of large angle of incidence were not highly accurate because of the small-angle parabolic model employed for their numerical simulations. In addition, the previous papers did not analyze the effect of nonlinearity of incident waves on the development of stem waves.

Figure 1. Relative wave height along the front wall of CASE 1 and CASE 2 (Lee and Kim, 2007)

Thus, the present authors decide to conduct precisely-controlled and comprehensive hydraulic experiments to investigate the stem waves. In the present experiments the gravel beach is carefully designed to reduce the reflected waves at less than 3% for all the incident waves considered. To overcome the narrowness of the basin the water depth is reduced to 0.25 m to secure the length of vertical wall at least 40 wavelengths for the case of T = 0.7 s and 20 wavelengths for the case of T = 1.1 s. To obtain data for various wave conditions including nonlinearity and angle of incidence total of 24 cases are considered. The large-angle parabolic model is employed to get more accurate solutions for the waves with large angle of incidence. Based on the observation of the experimental data we propose a mechanism for the generation of stem waves in a different point of view.

Corresponding to the referee's comment we added the following description in the manuscript.

[revised manuscript text omitted]

---

## Referee Report (RR1)

Report on "**Laboratory and numerical experiments on stem waves due to monochromatic waves along a vertical wall**", by Yoon, Lee, Kim & Shin

The manuscript at hand presents a detailed experimental analysis of the formation of stem waves, due to monochromatic waves interacting with a vertical wall. The experiments presented here are of excellent quality, and they are compared with a weakly nonlinear numerical code (REF/DIF) and a linear analytical solution.

The data are new, and truly interesting for the community. Besides, the comparisons with numerical and analytical data provide good insights of the phenomenon.

Furthermore, the manuscript is well written, and the presentation of the results is relatively clear. For all these reasons, I consider this manuscript should be published in the "Nonlinear Processes in Geophysics". In the meantime, I have the feeling the data are not fully analysed, and the discussion might support them better. I will detail these concerns in the following:

- Presentation of the two models is not sufficient.
  First, the simplification of equation (6) in Kirby and Dalrymple (2002) to equation (1) of the present manuscript is not straightforward. Extra precision should be given, especially focusing on the assumptions used (the order of nonlinearity, the use of parabolic formulation of mild slope equation, which forbids reflexion in the main direction of propagation, but not in the transverse direction, and the use of Padé approximants related to the kind of angles which might be reached in such conditions, …). Furthermore, the manuscript suffers an important lack of details about the numerical solution (numerical grid, boundary conditions used on two out of four boundaries, …)
  Secondly, the linear analytical solution is interesting, because it is linear, and, by definition, does not allow the formation of stem waves. This point is not clearly enough stated in the discussion. Besides, a few more details on the derivation might be welcome.

- The second point which needs clarification concerns the very definition of stem waves. It is not clearly stated in the manuscript, even if the doodle in figure 2 provides good indication. For this reason, the definition of the stem width and its computation is awkward, even if it probably constitutes a major finding of the manuscript (discussion in page 8, lines 5-15). I have the feeling this discussion should be significantly enlarged. For instance, a map of the wavenumbers can be computed from ref-dif data, providing the area were waves propagate parallelly to the wall. A comparison with these data, and the three definitions suggested here could be interesting, providing a benchmark of each of the three methods. Furthermore, the definition introduced by the authors is very interesting: given their definition of lambda, they provide the location of an imaginary wall, where idealized reflexion would appear. The distance between the wall, and this imaginary reflexion location corresponds to the stem width. This point is not explained in the text, and it would support the discussion. Finally, this new definition could be used to analyse the dependence of this width to the two parameters (nonlinearity and angle of the wall). Besides, it was not obvious to me why a single nonlinear parameter K would be sufficient to describe the phenomenon. Few words about it, and a plot of the stem width versus K could also be enlightening.

- The final point which could be improved concerns the interpretation provided by the authors about stem waves formation. Even if their observations are interesting, I was not convinced

by their interpretation. Since the phenomenon is nonlinear, it is probably connected to a resonant interaction among waves. This is rather classical (see for instance three waves interactions). Surely, it is connected to a shift in the wavelength of water waves, but this is probably not the main mechanism responsible for their formation.

---

## Referee Report (RR2)

Response to comments from anonymous referee #1

Title: **Laboratory and numerical experiments on stem waves due to monochromatic waves along a vertical wall**

MS No.: npg-2017-35

We appreciate the referee's interest and criticisms on our manuscript entitled "Laboratory and numerical experiments on stem waves due to monochromatic waves along a vertical wall". We hope that the revision we made could have well reflected the referee's comments.

Referee's comment:

(1) When the authors talk about the cross-sea condition, the nexus between references and the present manuscript should be better focused. Specifically, the present manuscript does not analyze the vorticity induced by crossing breaking waves, but the interaction between two angled wave trains (the incident and the reflected ones), which have clear connections with what described in the analytical theory of Postacchini et al. (2014) for the identification of the breaking location.

[Figure]

<Response from authors>

Figure 1, obtained and modified from Fig. 2 of Postacchini et al. (2014), presents the free surface pattern of the shoaling crossing waves. The free surface calculated using nonlinear model shows that the diamond pattern in the offshore changes to the honeycomb pattern in the surf zone because of the increase of wave length (or phase speed) due to nonlinear effect. The stem waves are growing as the waves approach the shore.

[Figure]

Figure 1. Free surface patterns of shoaling crossing waves calculated using linear (left) and nonlinear (right) models. The vertical scale is increased to fit the horizontal scale.

The authors revised the manuscript as the referee suggested as:

**Lines 11-16 of page 7**

[Figure]

Postacchini et al. (2014) studied the dynamics of crossing wave trains on a plane slope in shallow waters. The stem waves can be developed at the intersection of two crest lines of the crossing waves. The crossing waves propagating towards a shore experience the shoaling and break. Postacchini et al. (2014) proposed an analytical theory based on ray convergence to identify the position and the crest length of the breaker. The stem waves in the present study are developed by the oblique nonlinear interaction between the incident and the reflected waves. Thus, the generation mechanism is similar to each other.

(2) The term "l" does not seem to have been included into Fig.26b (i.e. Fig.23b of the original manuscript).

<Response from authors>
The idea to deal with stem waves as a refraction-reflection along the stem boundary is premature to propose. Thus, all of the sentences and figures related to it are removed from the manuscript. The authors provide a new definition of stem width in the revised manuscript as:

**Page 8**

Prior to presenting the experimental and numerical results, the definitions of the stem angle and the stem width are discussed. The definition of stem width is rather controversial. Yue and Mei (1980) defined the stem width as the distance from the wall to the edge of the uniform wave amplitude region. However, it is not an easy task to locate the edge of the flat region. Berger and Kohlhase (1976) defined the stem width for the periodic waves as the distance along the stem crest lines from the wall to the first node line of standing wave pattern which is easier to identify from the measured data. On the other hand, Peterson et al. (2003), Soomere (2004) and Soomere and Engelbrecht (2005) obtained the analytical stem length using the KP equation for the obliquely interacting two solitary waves. As pointed out by Li et al. (2011) the crest lines of the stem wave, the incident and the reflected solitons measured in their experiment are not straight, and they do not meet at a point. In reality, the analytical solutions of the KP equation deviate slightly from the pattern observed in the experiment. Thus, Li et al. (2011) proposed the edge of the Mach stem as the intersection of the linear extensions of the stem and the incident-wave crest lines.

For the periodic waves the wave pattern is more complicated because many wave components are superposed. Thus, the definitions of the stem boundary and the stem angle should be different from the case of solitary waves. As shown in Fig. 2(a) and Fig. 5, when the stem waves are fully developed, the stem boundary is nearly parallel to the first node line. Thus, as suggested by Berger and Kohlhase (1976), the experimental stem angle $\alpha$ is determined in this study as the angle of node line, $\alpha_n$. The node line is roughly determined using the node points from the wave height data measured along two lines of $x = 6L$ and $15L$. When the distances between the first node points and the wall are $\lambda_6$ and $\lambda_{15}$ for two sections of $x = 6L$ and $15L$, respectively, then the angle of the node line, $\alpha_n$, can be determined as

$$\alpha \approx \alpha_n = \tan^{-1}\left(\frac{\lambda_{15} - \lambda_6}{9L}\right). \tag{11}$$

[Figure]

This $\alpha_n$ decreases as the waves propagate along the wall. It reaches an asymptotic value after the waves propagate approximately 30 wave lengths. Thus, the experimental $\alpha_n$ determined by Eq. (11) is slightly overestimated for $x \leq 30L$.

In this study the stem angle, $\alpha$, is defined as the asymptotic angle of node line as shown in Fig. 5. To estimate the asymptotic $\alpha_n$ the numerical calculation is conducted using the domain extended up to $50L$ in the $x$-direction, and the instantaneous free surface displacements are calculated and plotted as shown in Fig. 5. Using two distances between the node points and the wall, $\lambda_{30}$ and $\lambda_{50}$ for two sections of $x = 30L$ and $50L$, respectively, the stem angle $\alpha$ is determined as

$$\alpha = \alpha_n = \tan^{-1}\left(\frac{\lambda_{50} - \lambda_{30}}{20L}\right). \tag{12}$$

The stem width $\lambda_s$ can be determined using the stem angle $\alpha$ as

$$\lambda_s = x \tan \alpha. \tag{13}$$

[Figure]

Figure 2. Coordinate system for numerical simulations: (a) present, (b) Yue & Mei (1980).

[Figure]

**Figure 5. Definition sketch for the stem angle and the stem boundary.**

Response to comments from referee #3 (Soomere, Tarmo)

**Title: Laboratory and numerical experiments on stem waves due to monochromatic waves along a vertical wall**

MS No.: npg-2017-35

We appreciate the referee's interest and criticisms on our manuscript entitled "Laboratory and numerical experiments on stem waves due to monochromatic waves along a vertical wall". We hope that the revision we made could have well reflected the referee's comments.

Referee's comment:

(1)  Having said that, I wonder whether the authors would consider possible to make a little bit of extra work. As the authors correctly discuss, the definition of stem width (=length of the high common crest of the incoming and reflected wave) is controversial and used in different meanings by different authors. The same problem becomes evident in the limiting case of stationary interactions of shallow-water Kadomtsev-Petviashvili solitons (e.g., Peterson et al. 2003. Soliton interaction as a possible model for extreme waves in shallow water, Nonlinear Processes in Geophysics, 10, 6, 503–510). In this specific case the height of the joint crest varies along the stem, except for the near-resonance case, and it takes time to form a stem of reasonable length (Li et al. 2011. On the Mach reflection of a solitary wave: revisited. Journal of Fluid Mechanics, 672, 326-357).

Even though the stem formation from wave trains considered by the authors is time-dependent and thus very much different from the process of the formation of stationary pattern of interaction of shallow-water solitons, the existence of simple expressions for the core quantities for solitons interactions (e.g. Soomere and Engelbrecht 2005. Extreme elevations and slopes of interacting solitons in shallow water, Wave Motion, 41, 2, 179–192) may put the results in a wider context and can possibly make the results applicable for Mach reflection of solitons as well.

Namely, a rough estimate of the critical angle for resonance of solitons of equivalent amplitude (that match the amplitudes of the incident and reflected waves), crossing angle of the two wave systems and water depth; see, e.g., Soomere 2004. Interaction of Kadomtsev-Petviashvili solitons with unequal amplitudes, Physics Letters A, 332, 1-2, 74–81) might provide some additional explanation why stem formation only occurs for quite a selected set of generated wave fields. I guess that the resonance angle varies considerably for different generated wave heights and thus its value has some potential to clarify why in some cases the stem exists and why it is not present in some other cases. Trains of longer and/or higher waves are in this sense closer to similar trains of shallow-water line solitons and thus the estimates for parameters of soliton interactions should better match the observed development of stem.

However, as this possible amendment would eventually involve references to my own papers, please consider this suggestion as a very gentle one, and in no way as a condition for the acceptance of the manuscript.

<Response from authors>

The authors agree with the reviewer in the fact that the generating mechanism of stem waves for the periodic waves is similar to that for the solitary wave. The authors provide some summary of the previous research works on the stem length by solitary wave as in the followings:

**Page 8**

[revised manuscript text omitted]

< Minor points >

| Comments and Suggestions | Response | Page Reference (original) | Page Referred (revised) |
|---|---|---|---|
| Abstract, line 19 and page 13, line 12: replace „the lengthening of wave length" by „the increase in the wave length" | The authors eliminated the sentence including "the lengthening of wave length" because the generation mechanism of stem waves is analyzed in a different way." | | |
| Page 7, line 22: it would be better to say that the same result „apparently" could be obtained. | The authors corrected "If the vertical wall is sufficiently long, the same result could apparently be obtained for $\theta_0=10°$." as suggested by the reviewer. | Page 7, line 22 | Page 9, line 16 |
| Page 8, line 8: I agree that „However, it is not an easy task to locate the edge of the flat region." Here, again, a reference (even though not 100% relevant) to the case of interacting line solitons (or solitons reflecting from the wall) would make this explanation clearer. | As suggested by the reviewer, the definition of stem angle and stem width are revised. The revision is already presented as a response to the major comment (1) above. | | |
| Table 1: move water depth (0.25 m) into the caption as otherwise it creates an empty column. | As suggested by the reviewer, the water depth (0.25 m) is moved into the caption of Table 1. | Table 1 | Table 1 |

Response to comments from referee #4 (Touboul, Julien)

Title: **Laboratory and numerical experiments on stem waves due to monochromatic waves along a vertical wall**

MS No.: npg-2017-35

We appreciate the referee's interest and criticisms on our manuscript entitled "Laboratory and numerical experiments on stem waves due to monochromatic waves along a vertical wall". We hope that the revision we made could have well reflected the referee's comments.

Referee's comment:

(1) Presentation of the two models is not sufficient.

First, the simplification of equation (6) in Kirby and Dalrymple (2002) to equation (1) of the present manuscript is not straightforward. Extra precision should be given, especially focusing on the assumptions used (the order of nonlinearity, the use of parabolic formulation of mild slope equation, which forbids reflexion in the main direction of propagation, but not in the transverse direction, and the use of Padé approximants related to the kind of angles which might be reached in such conditions, …). Furthermore, the manuscript suffers an important lack of details about the numerical solution (numerical grid, boundary conditions used on two out of four boundaries, …)

Secondly, the linear analytical solution is interesting, because it is linear, and, by definition, does not allow the formation of stem waves. This point is not clearly enough stated in the discussion. Besides, a few more details on the derivation might be welcome.

<Response from authors>

The simplification is relatively straightforward. The authors provide some details on the ref/dif model in the revised manuscript as the referee suggested as:

**Lines 22-26 of page 3**

The REF/DIF model can deal with the refraction-diffraction of Stokes waves of third order nonlinearity over a slowly varying depth and current. Due to the use of parabolic formulation the reflection in the main direction of propagation is forbidden, but not in the transverse direction. In this study, the water depth is uniform, and no ambient current is present. With no current and energy dissipation on a constant water depth and by selecting (1, 1) Padé approximant in the model, the governing equation of the REF/DIF model is simplified as

**Lines 8-11 of page 4**

The third term of Eq. (1) is the correction term obtained by selecting (1, 1) Padé approximant for the wide angle parabolic approximation. According to Fig. 2 of Kirby (1986) the accuracy of the waves propagating obliquely to the main direction of propagation, i.e., $x$-direction, can be maintained up to $\pm 45°$. In this study the range of the incidence angles of both incident and reflected waves lies from $\pm 10°$ to $\pm 40°$. Thus, the considerable accuracy

of the numerical solution is expected.

**Lines 1-8 of page 5**
If the side boundary opposite to the vertical wall is located far from the wall, no flux boundary condition, Eq. (6), can also be used. However, to save the computational resources the obliquely-incident plane wave condition is prescribed along the side boundary at $y = -y_{max}$ as

$$A = A_0 e^{i(k_n\ x \cos\theta_0\ -\ k_n\ y_{max} \sin\theta_0)}. \tag{7}$$

Along the down-wave side no boundary condition is necessary, because Eq. (1) is a parabolic type differential equation. The grid size, $\Delta x$ and $\Delta y$, is $L/80$ where $L$ is the wave length of incident wave. The size of computational domain is $50L$ in the $x$-direction, and $400L$ in the $y$-direction.

**Line 26 of page 5 and Lines 1-2 of page 6**
The analytical solution of Chen (1987) is linear. Thus, this analytical solution does not allow the formation of stem waves. The details of the derivation of the analytical solution can be found in Chen (1987).

(2) The second point which needs clarification concerns the very definition of stem waves. It is not clearly stated in the manuscript, even if the doodle in figure 2 provides good indication. For this reason, the definition of the stem width and its computation is awkward, even if it probably constitutes a major finding of the manuscript (discussion in page 8, lines 5-15). I have the feeling this discussion should be significantly enlarged. For instance, a map of the wavenumbers can be computed from ref-dif data, providing the area where waves propagate parallelly to the wall. A comparison with these data, and the three definitions suggested here could be interesting, providing a benchmark of each of the three methods. Furthermore, the definition introduced by the authors is very interesting: given their definition of lambda, they provide the location of an imaginary wall, where idealized reflexion would appear. The distance between the wall, and this imaginary reflexion location corresponds to the stem width. This point is not explained in the text, and it would support the discussion. Finally, this new definition could be used to analyse the dependence of this width to the two parameters (nonlinearity and angle of the wall). Besides, it was not obvious to me why a single nonlinear parameter K would be sufficient to describe the phenomenon. Few words about it, and a plot of the stem width versus K could also be enlightening.

<Response from authors>
**Page 8**
Prior to presenting the experimental and numerical results, the definitions of the stem angle and the stem width are discussed. The definition of stem width is rather controversial. Yue and Mei (1980) defined the stem width as the distance from the wall to the edge of the uniform wave amplitude region. However, it is not an easy task to locate the edge of the flat region. Berger and Kohlhase (1976) defined the stem width for the periodic waves as the distance along the stem crest lines from the wall to the first node line of standing wave pattern which is easier

to identify from the measured data. On the other hand, Peterson et al. (2003), Soomere (2004) and Soomere and Engelbrecht (2005) obtained the analytical stem length using the KP equation for the obliquely interacting two solitary waves. As pointed out by Li et al. (2011) the crest lines of the stem wave, the incident and the reflected solitons measured in their experiment are not straight, and they do not meet at a point. In reality, the analytical solutions of the KP equation deviate slightly from the pattern observed in the experiment. Thus, Li et al. (2011) proposed the edge of the Mach stem as the intersection of the linear extensions of the stem and the incident-wave crest lines.

For the periodic waves the wave pattern is more complicated because many wave components are superposed. Thus, the definitions of the stem boundary and the stem angle should be different from the case of solitary waves. As shown in Fig. 2(a) and Fig. 5, when the stem waves are fully developed, the stem boundary is nearly parallel to the first node line. Thus, as suggested by Berger and Kohlhase (1976), the experimental stem angle $\alpha$ is determined in this study as the angle of node line, $\alpha_n$. The node line is roughly determined using the node points from the wave height data measured along two lines of $x = 6L$ and $15L$. When the distances between the first node points and the wall are $\lambda_6$ and $\lambda_{15}$ for two sections of $x = 6L$ and $15L$, respectively, then the angle of the node line, $\alpha_n$, can be determined as

$$\alpha \approx \alpha_n = \tan^{-1}\left(\frac{\lambda_{15} - \lambda_6}{9L}\right). \tag{11}$$

This $\alpha_n$ decreases as the waves propagate along the wall. It reaches an asymptotic value after the waves propagate approximately 30 wave lengths. Thus, the experimental $\alpha_n$ determined by Eq. (11) is slightly overestimated for $x \leq 30L$.

In this study the stem angle, $\alpha$, is defined as the asymptotic angle of node line as shown in Fig. 5. To estimate the asymptotic $\alpha_n$ the numerical calculation is conducted using the domain extended up to $50L$ in the $x$-direction, and the instantaneous free surface displacements are calculated and plotted as shown in Fig. 5. Using two distances between the node points and the wall, $\lambda_{30}$ and $\lambda_{50}$ for two sections of $x = 30L$ and $50L$, respectively, the stem angle $\alpha$ is determined as

$$\alpha = \alpha_n = \tan^{-1}\left(\frac{\lambda_{50} - \lambda_{30}}{20L}\right). \tag{12}$$

The stem width $\lambda_s$ can be determined using the stem angle $\alpha$ as

$$\lambda_s = x \tan \alpha. \tag{13}$$

[Figure]

**Figure 2. Coordinate system for numerical simulations: (a) present, (b) Yue & Mei (1980).**

[Figure]

**Figure 5. Definition sketch for the stem angle and the stem boundary.**

As the referee suggested the amplitude, wave number, and incidence angle are calculated using ref/dif for the case of MLL1, and are given in the following figures. The free surface distribution is already given in Fig. 5 above. This analysis was made based on the old definition of stem boundary before the authors switch to the new one. According to the new definition (Fig. 5 above) which uses the node angle far downwave area of $30L < x <$

50*L*, the stem angle is reduced in comparison with that of old version (*x=25L*) shown in the followings:

[Figure]

Figure: Amplitude distribution in the domain (left) and along *x=25L* (right)

[Figure]

magnitude of wave number distribution in the domain (left) and along *x=25L* (right)

[Figure]

incidence angle distribution in the domain (left) and along *x=25L* (right)

In the figures the definition (but it is old definition) of stem boundary used in the present manuscript is shown. There is no clear cut to divide the stem region, because the amplitude, wave number, and incidence angle change slowly near the stem boundary defined in this manuscript. As the referee pointed out the wave number and the incidence angle can give a slightly better way to judge. As shown in figures the definition of stem width used in this manuscript covers effectively the stem area being defined using the wave number or the incidence angle. These discussions are not presented in the revised manuscript because the definition is switched to a new one. However, the suggestion from the referee gave insight for better understanding.

**Lines 12-14 of page 5**

*K* is the single parameter representing both the nonlinearity of incident wave and the angle of incidence on the formation of stem waves along the vertical wall. This nonlinear parameter was obtained by Yue and Mei (1980) from the dimensionless form of the small angle version of Eq. (1). The details of the derivation of *K* can be found in Yue and Mei (1980).

(3) The final point which could be improved concerns the interpretation provided by the authors about stem waves formation. Even if their observations are interesting, I was not convinced by their interpretation. Since the phenomenon is nonlinear, it is probably connected to a resonant interaction among waves. This is rather classical (see for instance three waves interactions). Surely, it is connected to a shift in the wavelength of water waves, but this is probably not the main mechanism responsible for their formation.

<Response from author>
The authors express their sincere apology to the referee for confusing about the generation mechanism of stem waves. The authors revised the manuscript as:

**Lines 23-29 of page 12**

It is well-known that the stem waves are generated by the nonlinear interaction between the incident and the reflected waves. When the angle between the incident and the reflected waves is small and the amplitude of two waves is small-but-finite, two waves attract each other and form a new wave with a single crest so-called the stem wave. The amplitude of the stem wave is larger than the incident wave, and that of reflected wave is smaller. Three waves meet at a point due to both the continuous growth of the crest length of stem wave and the phase-shift of reflected wave. All the mechanism observed in the formation of Mach stem wave for the solitary waves applies also for the monochromatic Stokes waves, but the intensity of nonlinear interaction is weaker than that of solitary waves.

Response to comments from anonymous referee #5

Title: **Laboratory and numerical experiments on stem waves due to monochromatic waves along a vertical wall**

MS No.: npg-2017-35

We appreciate the referee's interest and criticisms on our manuscript entitled "Laboratory and numerical experiments on stem waves due to monochromatic waves along a vertical wall". We hope that the revision we made could have well reflected the referee's comments.

Referee's comment:
(1) The theory could be presented much more clearly even though these are published, fairly old results. Please clarify and expand.

<Response from author>
We have provided some more details on the background of the theory presented in the manuscript.

(2) What is missing is a comparison with recent work (including references) on nonlinear stem waves in KP and higher-order water wave approximations than KP; KP and these other equations also allow monochromatic/harmonic standing wave solutions, maybe as solitary waves, which become harmonic waves in the small-amplitude limit. In these cases the amplification is a lot larger (up till 4x) and I miss a discussion of the relevance of these equations and solutions, the single soliton but also harmonic, solitary-wave solutions to these equations, which must somehow be connected with the work presented. Please update and clarify.

<Response from author>
The authors provide some summary of the previous research works on the topic of Mach stem generated by solitary waves in Section 1 as in the followings:

[revised manuscript text omitted]

(3) There are a lot of figures; are these all required? The nonlinear results with stem waves are the most interesting but I miss in these figures an indication what the extent of the stem wave is, as in Fig. 2. What are the observed stem-wave angles of the wall? There should be some reordering here, with perhaps some results relegated to an appendix or online-only appendix. It would also be useful to mention the values of K in the relevant captions. Please clarify.

<Response from author>
The authors agree with the referee's suggestion and have moved the experimental results to the appendix. The definition of the stem wave is clarified in section 4 as in the followings:

**Page 8**
Prior to presenting the experimental and numerical results, the definitions of the stem angle and the stem width are discussed. The definition of stem width is rather controversial. Yue and Mei (1980) defined the stem width as the distance from the wall to the edge of the uniform wave amplitude region. However, it is not an easy task to locate the edge of the flat region. Berger and Kohlhase (1976) defined the stem width for the periodic waves as the distance along the stem crest lines from the wall to the first node line of standing wave pattern which is easier to identify from the measured data. On the other hand, Peterson et al. (2003), Soomere (2004) and Soomere and Engelbrecht (2005) obtained the analytical stem length using the KP equation for the obliquely interacting two solitary waves. As pointed out by Li et al. (2011) the crest lines of the stem wave, the incident and the reflected solitons measured in their experiment are not straight, and they do not meet at a point. In reality, the analytical solutions of the KP equation deviate slightly from the pattern observed in the experiment. Thus, Li et al. (2011) proposed the edge of the Mach stem as the intersection of the linear extensions of the stem and the incident-wave crest lines.

For the periodic waves the wave pattern is more complicated because many wave components are superposed. Thus, the definitions of the stem boundary and the stem angle should be different from the case of solitary waves. As shown in Fig. 2(a) and Fig. 5, when the stem waves are fully developed, the stem boundary is nearly parallel to the first node line. Thus, as suggested by Berger and Kohlhase (1976), the experimental stem angle $\alpha$ is determined in this study as the angle of node line, $\alpha_n$. The node line is roughly determined using the node points from the wave height data measured along two lines of $x = 6L$ and $15L$. When the distances between the first node points and the wall are $\lambda_6$ and $\lambda_{15}$ for two sections of $x = 6L$ and $15L$, respectively, then the angle of the

node line, $\alpha_n$, can be determined as

$$\alpha \approx \alpha_n = \tan^{-1}\left(\frac{\lambda_{15} - \lambda_6}{9L}\right). \tag{11}$$

This $\alpha_n$ decreases as the waves propagate along the wall. It reaches an asymptotic value after the waves propagate approximately 30 wave lengths. Thus, the experimental $\alpha_n$ determined by Eq. (11) is slightly overestimated for $x \le 30L$.

In this study the stem angle, $\alpha$, is defined as the asymptotic angle of node line as shown in Fig. 5. To estimate the asymptotic $\alpha_n$ the numerical calculation is conducted using the domain extended up to $50L$ in the $x$-direction, and the instantaneous free surface displacements are calculated and plotted as shown in Fig. 5. Using two distances between the node points and the wall, $\lambda_{30}$ and $\lambda_{50}$ for two sections of $x = 30L$ and $50L$, respectively, the stem angle $\alpha$ is determined as

$$\alpha = \alpha_n = \tan^{-1}\left(\frac{\lambda_{50} - \lambda_{30}}{20L}\right). \tag{12}$$

The stem width $\lambda_s$ can be determined using the stem angle $\alpha$ as

$$\lambda_s = x \tan \alpha. \tag{13}$$

[Figure]

**Figure 2. Coordinate system for numerical simulations: (a) present, (b) Yue & Mei (1980).**

[Figure]

**Figure 5. Definition sketch for the stem angle and the stem boundary.**

The values of *K* are provided to each figure as in the followings:

[Figure]

**Figure A1. Normalized wave heights along the wall for the cases of MSS1 ~ MSS4. Solid circle: measured, solid line: present numerical, dashed line: analytical (Chen, 1987).**

(4)  What is the relevance to real-world situations? What range of nonlinearities do we expect in these real-world cases? Are the experiments lying in this range? Are the solitary waves lying in this range? Please clarify.

<Response from author>
   In the real world, we can assume the situation where the swell is incident on a breakwater. Swell waves are the regular longer period waves created by storms far away from the beach. Swell waves tend to have longer periods than wind waves. The wave period of swell lies between 10 s to 15 s. Breakwaters are generally constructed at a depth of about 10 m to 20 m. If the wave height is 1 m to 3m, the swell wave conditions can be within the range of Stokes wave as shown in the following figure.

[Figure]

Figure: Wave conditions frequently met in the real world.

We have added a statement to further illustrate the wave conditions tested in the experiment as in the followings:

**Lines 29-31 of page 6**
As shown in Fig. 4 the incident waves tested in this study belong to the Stokes range. The dispersion effect of the Stokes waves is much stronger than that of the solitary waves. Thus, the characteristics of stem waves in this study should be much different from those of the solitary waves.

[Figure]

**Figure 4. Wave conditions of the incident waves used in the present experiment.**

< Minor points >

| Comments and Suggestions | Response | Page Reference (original) | Page Referred (revised) |
|---|---|---|---|
| Abstract Line 45: is the word "decrease" correct? Should it not be "increase"? Counterintuitive. | The wave heights along the wall itself increase as the amplitude of the incident waves increase. However, the normalized wave heights decrease. | Page 1, line 16 | |
| Line 51: Mention relevance to harbours and such. | The relevance to real-world situations is presented in the response to the referee's major comment (4). | | |
| Line 109 page 4: Overview of reflection of solitary wave-wall-interactions are missing, with the maximum stem wave amplification being 4 for a critical angle in KP -see works of, e.g., Kodama, Yeh and Kodama, Ablowitz and Curtis, Gidel et al., etc., also with respect to the amplification in other water-wave model-approximations of potential flow. This should include the comparison between KP and other models and experiments. | The overview on the reflection of solitary waves is provided as the response to the referee's major comment (2). | | |
| Line 136: equations should be in italics. | The authors corrected as the referee suggested. | | |

| | | | |
|---|---|---|---|
| Line 143: singular dispersion. | To the best of the authors' knowledge the terminology 'singular dispersion' is not familiar. If it means 'dispersion derived from linear theory', the authors are happy to replace in the final manuscript. | Page 3, line 22 | |
| Page 5, section: clarification would be useful, 20 years after these old publications. The REF/DIF manual and reference is also not particularly clear. Everywhere: formulas need punctuation, also in NPG. | The publications referred in this manuscript are old, but they can be easily accessible on internet site.
The commas and punctuations are provided to each formula where they are appropriate. | Page 3, line 17 | |
| Line 188: explain/define the linear equation set which this linear solution solves. | As the referee suggested, the equation for the analytical solution is provided as:
"Chen (1987) developed an analytical solution for the Helmholtz equation in polar coordinates to solve the combined reflection and diffraction of monochromatic waves due to a vertical wedge." | Page 5, line 5 | Page 5, line 16 |
| Line 244: zero-crossings method: please explain. Which zero? What crossing? | Following the referee's suggestion, some statement explaining the zero-upcrossing method is added as:

"The wave heights are extracted from the measured free surface displacements using the zero-upcrossing method. In this method a wave is defined when the surface elevation crosses the zero-line or the mean water level upward and continues until the next crossing point. This method is a widely accepted method for extracting representative statistics from raw wave data." | Page 6, line 24 | Page 7, lines 6-9 |
| Line 293: remove first "as". | The first "as" is removed. | Page 7, line 27 | Page 9, line 19 |

| | | | |
|---|---|---|---|
| Page 9: Over the top amount of detail in these figures. Is there a more compact way to convey this? | The authors have moved all figures presenting the experimental data to the Appendix. | | |
| Line 319/320: "strong indication of stem wave development": please indicate why this statement holds: arrow in figure, etc; it was not very clear to me;
What is the stem-wave line, i.e. the measured dashed line of Fig. 2; per position plot along the wall and normal to the wall indicate where this dashed line is for this position. I.e. indicate where the stem wave end and what its angle is. Somehow, this stem-wave angle should be available from 2D-horizontal measurements or photographs? | In all of the relevant figures the portion of the stem waves is marked with a red solid line, and the end point of the stem wave corresponding to the dashed line of Fig. 2 is marked with a vertical line. In the revised manuscript the definitions of the stem angle and the stem width are revised (see the response to the referee's major comment (3)). Fig. 5 of the revised manuscript provides more detailed definition of stem waves and how the observed and the calculated stem angle and stem width are determined. | Page 8, line 25 | |
| Line 322: I would agree with this statement but which stem-wave angle and position of the end of the stem-wave do we measure or expect? Please add. | The response from the authors to the referee's comment for Line 319/320, applies also to this comment. | | |
| Line 354: "stem wave appear clearly"; please indicate where and in which figure (i.e. it is not very clear); add an arrow and symbol to indicate where the stem wave ends. What is the stem-wave angle (measured) for these cases? | The response from the authors to the referee's comment for Line 319/320, applies also to this comment. | | |
| Line 355: remove the 2nd "the". | The authors corrected as the referee suggested. | | |
| Line 372: explain why; say "because it is linear". [the analytical solution] | The authors corrected as the referee suggested as:
"while the analytical solution gives no stem wave, because it is linear." | Page 13, line 15 | Page 14, line 18 |

| | | | |
|---|---|---|---|
| Line 373: where does the conclusion come from; which figures? is it true? I can't really see it also because per figure it is not clearly indicated which one contains a stem wave. Mark this more properly and add reference to the relevant figures or subfigures backing up this statement. What about solutions to KP? Would they be better? Or is Benney-Luke or potential flow required? Please comment. Formula (12): can this be explained/derived quickly; why are shocks expected and is this representation relevant? | The response from the authors to the referee's comment for Line 319/320, applies also to the first part of this comment.

As far as the authors know, the KP and the Benney-Luke equations are valid for weakly-nonlinear and weakly-dispersive waves. As shown in Fig. 4 of the revised manuscript, the waves presented in this study are in the range of Stokes wave. Thus, the frequency dispersion is stronger than the shallow water waves.

Formula (12) was derived by Yue and Mei (1980) as an approximation to stem waves in analogous to a discontinuous shock. This is the only analytical formula to give the asymptotic stem height. Even though the authors do not understand how to derive it, it can be used for comparison purpose. | | |
| Line 413: remove first comma. | The authors corrected as the referee suggested. | | |
| Line 432 and Fig 24: I find the multiple lines displayed confusing and the figure caption unclear; there also seems to be only one theta=20 measurement; please clarify the figure and text. | The authors removed the lines that show the relation between the crest lines of the incident, reflected and stem waves. | Figure 24 | Figure 8 |
| Line 482: rewrite this sentence. Grammar. | As the referee suggested, the authors corrected the sentence as: "The results obtained from this study are summarized:" | Page 13, line 6 | Page 14, line 9 |
| Line 483: undulations. | It is corrected. | Page 13, line 7 | Page 14, line 10 |
| Line 484: an undulation. | It is corrected. | Page 13, line 8 | Page 14, line 11 |

| | | | |
|---|---|---|---|
| Line 487: I don't understand this statement; please clarify. | The statement is corrected as:
 "In particular, the wave height distributions for these small amplitude waves show no sign of stem wave." | Page 13, line | Page 14, line 13 |
| Line 488: this statement is not true as for larger waves the linear solution does not hold very well. Please amend. | This paragraph (numbered by 1) describes only the results obtained for small amplitude waves. Thus, the statement applies only for small amplitude waves. | | |
| Line 494: indicate in the figures what the values of $K$ are so this is more easy to judge. | The corresponding value of $K$ is supplied for each figure in Appendix. | | |
| Fig. 3: what is the signal imposed on the wavemaker; in order for the results to be reproducible? | The water depth where the wave paddles are placed is deeper than that of test area, and is connected with a gentle slope. The signal imposed on the wave generator was the monochromatic small amplitude waves. The waves experience shoaling before they enter the test area. The free surface displacements were measured at three incident wave measuring points shown in Fig. 3. The signal was adjusted until the target wave was produced. The generated wave showed a permanent form in the test area. The generation test was repeated three times to check the reproducibility. After the target wave was consistently obtained, the signal is stored. The signals for six target waves listed in Table 1 were obtained before the main experiments started. | | |
| Fig. 12: How does this match the sketch in Fig. 2; if the measurement is normal to the wall, where is the dashed line supposed to be, e.g. indicate with a vertical dashed line or cross? Please indicate. | The authors indicate the stem boundary with a vertical line. The stem width $\lambda_s$ is also marked in the relevant figures in Appendix. | | |

| | | | |
|---|---|---|---|
| Figures 15 & 18: Again, indicate the stem-wave end-point expected/ measured; cf. the dashed line in Fig. 2 at the appropriate x-location. | The authors indicate the stem boundary with a vertical line. The stem width $\lambda_s$ is also marked in the relevant figures in Appendix. | 💬 | |
| Figure 22 for K<0.5: What happens here? Please explain. | The authors corrected Fig. 22 (Fig. 9 in the revised manuscript) and added the following statement and a new figure (Fig. 10) to explain what happens for K<0.5.

[revised manuscript text omitted]

---

## Author Response (AR3)

Response to comments from referee #5 (Bokhove, Onno)

Title: **Laboratory and numerical experiments on stem waves due to monochromatic waves along a vertical wall**

MS No.: npg-2017-35

We appreciate the referee's interest and criticisms on our manuscript entitled "Laboratory and numerical experiments on stem waves due to monochromatic waves along a vertical wall". We hope that the revision we made could have well reflected the referee's comments.

Referee's comment:

(1) The text throughout requires some grammatical corrections pertaining to (better) understanding of statements. I have supplied a pdf with all my comments in confidence (not meant for online publication).

<Response from authors>

The authors corrected as the referee suggested.

(2) It is still unclear to me how lambda_s is determined at x=6L and 15L; is alpha determined and then knowing x, lambda_s or is lambda_s visually determined as in the appendix figures seems possible and is indicated? It is the chicken-and-the-egg question.

<Response from authors>

The measured $\alpha$ is determined as:

First, $\lambda_6$ and $\lambda_{15}$ are determined for given $\theta_0$ based on the figures in Appendix (e.g., Figs. A5 and A6). Then, $\lambda_6$ and $\lambda_{15}$ are substituted into Eq. (11) to determine the measured stem angle $\alpha$. This measured $\alpha$'s are presented by symbols in Fig. 11.

The calculated $\alpha$ and $\lambda_s$ are determined as:

First, $\lambda_{30}$ and $\lambda_{50}$ are determined for given $\theta_0$ based on the numerical simulation using extended domain. Then, $\lambda_{30}$ and $\lambda_{50}$ are substituted into Eq. (12) to determine the stem angle $\alpha$. This $\alpha$ is presented by solid curves in Fig. 11. Finally, the stem width $\lambda_s$ is calculated by substituting this $\alpha$ into Eq. (13) for given $x$, i.e. $x=30L$ and $50L$. This $\lambda_s$ is presented in the figures in Appendix.

(3) The question on the relevance to real-world cases should be answered in the paper, by combining the two figures in the response (figure 4 and the one before that).

<Response from authors>

Following the referee's suggestion, some statement explaining the relevance to real-world cases is added as:

In the real world, we can assume the situation where the swell is incident on a breakwater. Swell waves are the regular longer period waves created by storms far away from the beach. Swell waves tend to have longer periods than wind waves. The wave period of swell lies between 10 s to 15 s. Breakwaters are generally constructed at a depth of about 10 m to 20 m. If the wave height is 1 m to 3m, the swell wave conditions can be within the range of Stokes wave as shown in Fig. 4. In the figure the empty blue circles represent the swell wave conditions and the red triangles represent the incident waves tested in this study. It can be seen that the incident waves tested in this study belong to the Stokes range. The dispersion effect of the Stokes waves is much stronger than that of the solitary waves. Thus, the characteristics of stem waves in this study should be much different from those of the solitary waves. In Fig. 4, the *x*-axis represents the relative water depth (ratio of water depth to deep water wave length, i.e., the measure of wave dispersion). On the other hand, the *y*-axis represents the wave steepness (ratio of wave height to deep water wave length, i.e., the measure of wave nonlinearity).

[Figure]

Figure 4. The present experiment and wave conditions of the real-world cases (after Le Méhauté, 1976). The solid red triangles represent the incident waves tested in this study and empty blue circles represent the swell wave conditions. The *x*-axis represents the relative water depth (ratio of water depth to deep water wave length, i.e., the measure of wave dispersion). The *y*-axis represents the wave steepness (ratio of wave height to deep water wave length, i.e., the measure of wave nonlinearity).

(4) How is *K* calculated from Eq. (8) for the measurements and clarify how it is used to find the results in Figs. 11 and 12 a bit better.

<Response from authors>
In the experiment, the nonlinear parameter, $K$, is determined by Eq. (8) using the wave characteristics, $k$, $C$, and $C_g$ based on the linear wave theory using the measured $h$, $T$, $a_0$ and $\theta_0$. This statement is not included in the revision because it is well established in the coastal community.

(5) What is H(x,y,t) as follows from the simulations in terms of the amplitude A calculated from (1)?
<Response from authors>
The authors provide the relationship between $F$, $A$, and $H$ and revised the manuscript as:

**Lines 17 - 26 of page 5**
The analytical solution is given in a polar coordinate as shown in Fig. 1 as

$$\Phi(r,\theta^*,z,t) = -\frac{iga_0}{\omega}\frac{\cosh\{k(z+h)\}}{\cosh kh}F(r,\theta^*)e^{i\omega t}, \tag{9}$$

where $\Phi(r,\theta^*,z,t)$ is the velocity potential, and $F(r, \theta^*)$ is a diffraction factor (i.e., $A/a_0$) given as:

$$F(r,\theta^*) = \frac{2}{\nu}\left[J_0(kr) + 2\sum_{n=1}^{\infty} e^{in\pi/2\nu}J_{n/\nu}(kr)\cos\frac{n\alpha^*}{\nu}\cos\frac{n\theta^*}{\nu}\right], \tag{10}$$

where $\theta^* = \theta - 2\theta_0$, $\alpha^* = \pi - \theta_0$, $\nu = 2(\pi - \theta_0)/\pi$, and $\theta_0$ is the angle of incidence. $J_0(kr)$ is the Bessel function of the first kind of order 0. The absolute value of the diffraction factor $|F(r,\theta^*)|$ represents the normalized wave amplitude $|A|/a_0$, or the normalized wave height $H/H_0$ where $H_0$ $(= 2a_0)$ is the wave height of the incident wave.

(6) I still don't quite understand the zero crossing method. Please clarify what "next crossing" signifies?

<Response from authors>

The zero-upcrossing method is widely accepted method to get wave characteristics such as the wave height $H$ and the wave period $T$ from the measured time history of the free surface displacement at a given point of wave gauge. As shown in the following figure each wave is separated from a train of waves, using the zero-upcrossing points denoted by a red solid circle. This statement is not included in the revision.

[Figure]

(7) Figure 5 4: define axes and state what they indicate; add the real-world swell symbols used in the response as well and say in text and caption that the results seem thus relevant to the real world.

<Response from authors>
The definition and the physical meaning of axes are presented in the text and the caption of Fig. 4. The relevance to the real world is also included as in the response to the referee's major comment (3).

(8) What is a node line (define please)?

<Response from authors>

When the periodic waves are incident obliquely to the vertical wall, the normal component of incident wave is reflected. The superposition of incident and reflected waves foam the standing wave in the normal direction. The node point represents the location where the wave amplitude vanishes. When the amplitude of reflected wave is smaller than that of incident wave, the amplitude of node point is not zero but smaller than the neighbor points. The node line is a line which connects these node points. This statement is not included in the revision because this is widely accepted in the coastal community. However, the node points nearest to the vertical wall used to determine $\lambda_6$ and $\lambda_{15}$ are presented in the figures in the Appendix.

< Minor points >

| Comments and Suggestions | Response | Page Reference (original) | Page Referred (revised) |
|---|---|---|---|
| A lot of grammatical glitches have been indicated in the pdf-file. | The authors corrected as the referee suggested. | | |
| Formula (8): How is k determined in experiment? Are C and Cg then calculated using linear wave theory to find the K's used in figs 9 and 10? If so, say so. h is measured and known, A0 measured and theta0 as well so that K follows. | The details of how to determine $K$ in the experiment are presented in the response to the referee's major comment (4). | | |
| Line 14 on page 4 above section 2.2: What are h(x,y,t) and \phi(x,y,t) in terms of A? That would be useful as h (and perhaps phi) are used later on. | The relationship between $F$, $A$, and $H$ are provided as the response to the referee's major comment (5). | | |
| Line 7 page 7: I still don't understand it: what is meant by "next crossing point"? Please clarify. | The zero crossing method is provided as the response to the referee's major comment (6). | | |
| Lines 11-16: I don't think this addition is needed. | An anonymous referee suggested to include it in the text. | | |
| Lines 19: phrase "with a sufficiently long time" not understood | The authors added some statement as follows. "The first part of data with a sufficiently long time is discarded in evaluating the wave height to avoid the start-up transients, and the wave height and period are obtained using the zero-upcrossing method." | Page 7, line 19-20 | Page 7, line 26-28 |
| Line 31 page 7: All? I meant to choose a selective subset of figures for the main text and maybe relegate the rest to the appendix, provided they are useful. But the current setting does work although the subfigures with the stem waves indicated could be part of the main text. Either way would work. | Thank you for your comment. | | |

| | | | |
|---|---|---|---|
| Page 8 line 7: That is not a fair comment as it is much easier in the case of a solitary wave to find the Mach stem. | The authors are trying to understand that comment. However, it is not clear what it really means. Please be kind to give more details. | | |
| Formula (13): I am confused here: the figures in appendix indicate that lambda_s is determined by inspection and drawing (as the more flattened region), as now indicated in the appropriate cases, which is fine, and subsequently (13) is used, given x, to find alpha. But somewhere in captions it says lambda_s is determined using (13), in which case my question is how alpha has been measured from the data? Please clarify. | The descriptions of $\alpha$ and $\lambda_s$ are provided as in the response to the referee's major comment (2). | | |
| Page 9 line 31: I am lost here: given x, is lambda_s measured and then alpha calculated or the other way around: alpha is measured in which case please clarify how alpha is measured please clarify issue? | The authors revised the statements as follows to clarify as suggested by the referee. "The red lines shown in the figures represent the stem waves. For the stem width $\lambda_s$, the stem angle $\alpha$ is first determined by Eq. (12) using the numerical simulation result with the using extended domain. The stem width $\lambda_s$ is then calculated using Eq. (13) for given $x$." | Page 9 line 31 | Page 10, line 5-7 |
| Page 11 line 29: slightly? Can you indicate where this conclusion (that REF/DIF simulation angles are slightly different than experimental ones) comes from as the figures in the appendix indicate to me that experiments and REF/DIF are in agreement and any differences are within measurement error? | The authors express their sincere apology to the referee for confusing. This part should have been deleted from the previous manuscript. The authors eliminated the sentence including "The widths of stem waves in the REF/DIF model are shown to be slightly broader than those of the results from laboratory experiments. This may be due to the fact that the REF/DIF model overestimates the nonlinearity of the waves." because the definition of stem angle and stem width are revised. | | |

| | | | |
|---|---|---|---|
| Line 9 on page 12: provide the link between A and H as said/asked earlier. | The relationship between $A$ and $H$ is provided as the response to the referee's major comment (5). | | |
| Line 16 page 12: part of $K$ (namely finding C and Cg) seems to be based on linear theory? Is that fair? | The small angle version of the parabolic approximation equation, Eq. (1), was derived for the Stokes waves using the perturbation expansions based on the wave slope, $\epsilon = kA \ll O(1)$. The wave quantities of $k$, $C$ and $C_g$ are obtained from $O(kA)$. Thus, they are determined from linear dispersion relationships. The evaluation equation, Eq. (1), is obtained at $O(kA)^3$. Thus, there is no problem. | | |

| Lines 20-21, formulas (17) and (18): I am lost here? Please clarify the comparison. E.g., is gamma supposed to be gamma=tan theta_0/(sqrt(3)*cos theta0)? If so then say so. How was gamma determined? | For the solitary waves $\kappa_* = 1$ represents the border between two regions of stem development. When $\kappa_* < 1$, Mach stem is developed along the wall, while $\kappa_* > 1$, no stem wave is present. For the periodic Stokes waves, as shown in Fig. 9, the border lies at $K = 0.47$ for $\theta_0 = 10°$, and $K = 0.42$ for $\theta_0 = 20°$. To set the borders of stem region at a same point, i.e., $K_* = 1$, the scale adjustments are made using $\gamma$ as : $$(K_*)_{\theta_0=10°} = (K_*)_{\theta_0=20°} = 1$$ $$\left(\frac{\gamma}{K^{1/4}}\right)_{\theta_0=10°} = \left(\frac{\gamma}{K^{1/4}}\right)_{\theta_0=20°} = 1$$ $$\gamma_{\theta_0=10°} = \left(K_{\theta_0=10°}\right)^{1/4} = (0.47)^{1/4}$$ $$= 0.828$$ $$\gamma_{\theta_0=20°} = \left(K_{\theta_0=20°}\right)^{1/4} = (0.42)^{1/4}$$ $$= 0.805$$ This statement is not included in the revision. | | |

[revised manuscript text omitted]